# Delayed Algorithms for Distributed Stochastic Weakly Convex Optimization

**Wenzhi Gao**[*]
Stanford University
gwz@stanford.edu

**Qi Deng**[†]
Shanghai University of Finance and Economics
qideng@sufe.edu.cn

## Abstract

This paper studies delayed stochastic algorithms for weakly convex optimization in a distributed network with workers connected to a master node. Recently, Xu et al. 2022 showed that an inertial stochastic subgradient method converges at a rate of $\mathcal{O}(\tau_{\max}/\sqrt{K})$ which depends on the maximum information delay $\tau_{\max}$. In this work, we show that the delayed stochastic subgradient method (DSGD) obtains a tighter convergence rate which depends on the expected delay $\bar{\tau}$. Furthermore, for an important class of composition weakly convex problems, we develop a new delayed stochastic prox-linear (DSPL) method in which the delays only affect the high-order term in the complexity rate and hence, are negligible after a certain number of DSPL iterations. In addition, we demonstrate the robustness of our proposed algorithms against arbitrary delays. By incorporating a simple safeguarding step in both methods, we achieve convergence rates that depend solely on the number of workers, eliminating the effect of the delay. Our numerical experiments further confirm the empirical superiority of our proposed methods.

## 1 Introduction

In this paper, we consider the following stochastic optimization problem

$$\min_{x \in \mathbb{R}^n} \psi(x) := \mathbb{E}_{\xi \sim \Xi}[f(x,\xi)] + \omega(x), \tag{1}$$

where $f(x,\xi)$ is a nonconvex continuous function over $x$ and $\xi$ is a random variable sampled from some distribution $\Xi$; $\omega(x)$ is lower-semicontinuous and proximal-friendly. We assume that both $f(x,\xi)$ and $\omega(x)$ belong to a general class of nonsmooth nonconvex functions that exhibit weak convexity. Here, we say that a function $g(x)$ is $\kappa$-weakly convex if $g(x) + \frac{\kappa}{2}\|x\|^2$ is convex for some $\kappa \geq 0$. Weakly convex optimization has attracted growing interest in machine learning in recent years, and we are particularly interested in a general type of weakly convex problems with the following composition structure [12]

$$f(x,\xi) = h(c(x,\xi)), \tag{2}$$

where $h$ is a convex Lipschitz function and $c(x,\xi)$ is smooth. Optimization in the above composition form is pervasive in applications arising from machine learning and data science, including robust phase retrieval [14], blind deconvolution [8], robust PCA and matrix completion [5], among others.

Stochastic (sub)gradient method and its proximal variants [39, 10, 30, 28, 31] (all referred as SGD in our paper) are arguably the most popular approaches for solving problem (1). Typically, SGD iteratively solves $x^{k+1} = \arg\min_x \left\{ \langle f'(x^k, \xi^k), x - x^k \rangle + \omega(x) + \frac{\gamma_k}{2}\|x - x^k\|^2 \right\}$, where $\xi^k$ is

---

[*]Work done while at SHUFE.

[†]The corresponding author.

37th Conference on Neural Information Processing Systems (NeurIPS 2023).

a random sample and $f'(x^k, \xi^k)$ denotes a subgradient of $f(x^k, \xi^k)$. However, despite its wide popularity in machine learning, the sequential and synchronous nature of SGD is not suitable for modern applications that require parallel processing in multi-cores or over multiple machines.

To further improve SGD in parallel and distributed environments, recent work considers a more practical asynchronous setting where the parameter updates allow outdated gradient information. See [1, 32, 38, 29, 21]. In the asynchronous setting, it is crucial to know how the stale updates based on delayed information affects convergence. For smooth convex optimization, Agarwal and Duchi [1] show that delayed stochastic gradient method (DSGD) obtains a rate of $\mathcal{O}(\sigma/\sqrt{K} + \tau_{\sigma^2}/(\sigma^2 K))$, where $\tau_{\sigma^2}$ bounds of the second moment of random delays. DSGD with adaptive stepsize has also been studied by [25, 32, 38] to achieve better empirical performance and the latest work [2, 33] further improves the rate to $\mathcal{O}(\sigma/\sqrt{K} + \tau_{\max}/K)$. For general smooth (nonconvex) problems, the work [33] shows that DSGD converges to stationarity at a rate of $\mathcal{O}(\sigma/\sqrt{K} + \tau_{\max}/K)$. In a follow-up study [7], the authors propose a more robust DSGD whose rate depends on the average delay $\tau_{\mathrm{avg}}$, rather than the maximum delay. Recently, Koloskova et al. [18] develop a sharp analysis for asynchronous SGD and then a simple delay-adaptive stepsize to achieve the best rate $\mathcal{O}(\sigma/\sqrt{K} + \tau_{\mathrm{avg}}/K)$. Based on novel delay-adaptive stepsizes and virtual iterate-based analysis, a concurrent work [26] has established new convergence rates that depend on the number of workers rather than the delay of the gradient.

Despite much progress in distributed smooth optimization, it remains unclear how to develop efficient asynchronous algorithms for nonsmooth and even nonconvex problems. For general nonsmooth convex problems, the pioneering work [27] has shown that the asynchronous incremental subgradient method obtains an $\mathcal{O}(\sqrt{\tau_{\max}/K})$ convergence rate. The aforementioned work [26] shows that DSGD achieves a delay-independent rate of $\mathcal{O}(\sqrt{m/K})$, where $m$ is the number of workers. However, it is still unknown whether their key technique by telescoping the virtual iterates can be extended to composite nonconvex optimization.

For distributed weakly convex optimization, Xu et al. [34] shows that Delayed Stochastic Extrapolated Subgradient (DSEGD), an inertial version of DSGD, exhibits a convergence rate of $\mathcal{O}((1 + \tau_{\max})/\sqrt{K} + \tau_{\max}^2/K)$, which has an undesired dependence on the maximum delay $\tau_{\max}$. This issue is further exacerbated in real heterogeneous environments where the maximum delay is dictated by the slowest, or the "straggler" nodes. It remains unclear

*1) whether such delay dependence in DSGD is still improvable or not for weakly convex problems?*

Nevertheless, even if the rate is improvable in terms of $\tau$, there yet remains a fundamental challenge, as the delay significantly influences the leading term $\mathcal{O}(1/\sqrt{K})$ of the convergence rate. This contrasts with smooth distributed optimization, where the delay only affects a higher order term $\mathcal{O}(1/K)$ and hence is negligible in the long run. Such a performance gap highlights a substantial limitation of the prior study when nonsmoothness is present. Hence, it is natural to ask: For distributed weakly convex optimization,

*2) can we design algorithms with a negligible penalty caused by delays?*
*3) Moreover, can we make convergence rates delay independent?*

The goal of this paper is to address the above three questions.

**Contributions** In this paper, we answer the above questions positively (Table 1). Our contributions are as follows.

**1)** For distributed weakly convex optimization, we provide a sharp convergence analysis of DSGD under statistical assumptions on the delays and obtain a rate of $\mathcal{O}(\bar{\tau}/\sqrt{K} + \tau_{\sigma^2}/K)$, where $\bar{\tau}, \tau_{\sigma^2}$ are respectively first and second moments of stochastic delays. Our result significantly improves upon the previous $\mathcal{O}(\tau_{\max}/\sqrt{K} + \tau_{\max}^2/K)$ rate for DSGD [34].

**2)** For weakly convex problems with composition structure (2), we propose a new delayed stochastic prox-linear method (DSPL) which can exploit the structure (2) more effectively. Unlike SGD, the stochastic prox-linear algorithm ([9, 11]) partially linearizes the inner function $c(\cdot, \xi)$ while retaining the outer function $h(\cdot)$. Then it iteratively solves: $x^{k+1} = \arg\min_x \{h(c(x^k, \xi) + $

$\langle \nabla c(x^k, \xi), x - x^k \rangle) + \omega(x) + \frac{\gamma_k}{2} \|x - x^k\|^2 \}$. To the best of our knowledge, this is the first study of SPL in the asynchronous distributed setting. We show that the new DSPL method achieves a rate of $\mathcal{O}(1/\sqrt{K} + \tau_{\sigma^2}/K)$, which is consistently better than that of DSGD in terms of the dependence on delay. Interestingly, our result implies that the delay is negligible when $K$ is sufficiently large.

3) We propose a simple yet effective safeguarding step that skips the iteration when the delay is significantly large. This enhancement ensures that the rate depends only on the number of workers rather than on the delay explicitly. Specifically, in an environment of $m$ workers, we obtain an $\mathcal{O}(\sqrt{m}/\sqrt{K} + m/K)$ rate for DSGD and $\mathcal{O}(1/\sqrt{K} + m^2/K)$ for DSPL, making both methods robust to arbitrary delays. As per our knowledge, these are the first delay-independent rates for distributed nonsmooth nonconvex optimization. Prior to our work, delay-independent rates were only known for smooth or convex optimization [26], which were derived through a distinctly different delay-adaptive stepsize strategy.

Table 1: Rates of delayed algorithms for nonconvex optimization. $m$: the agent number; $\tau_{\max}$: the maximum delay; $\bar{\tau} = \mathbb{E}[\tau_k]$; $\tau_{\sigma^2} = \mathbb{E}[\tau_k^2]$; $\tau_{\text{avg}}$ average over arbitrary delays.

| Delay | Work | Setting | Problems / algorithms | Convergence rate |
|---|---|---|---|---|
| Non-robust | [34] | weakly convex | $f + \omega$ / DSGD | $\mathcal{O}\left(\frac{\tau_{\max}}{\sqrt{K}} + \frac{\tau_{\max}^2}{K}\right)$ |
| | Ours | | $f + \omega$ / DSGD | $\mathcal{O}\left(\frac{\bar{\tau}}{\sqrt{K}} + \frac{\tau_{\sigma^2}}{K}\right)$ |
| | | | $h \circ c + \omega$ / DSPL | $\mathcal{O}\left(\frac{1}{\sqrt{K}} + \frac{\tau_{\sigma^2}}{K}\right)$ |
| Robust | [26, 18, 7] | smooth nonconvex | $f$ / DSGD | $\mathcal{O}\left(\frac{1}{\sqrt{K}} + \frac{m}{K}\right)$ or $\mathcal{O}\left(\frac{1}{\sqrt{K}} + \frac{\tau_{\text{avg}}}{K}\right)$ |
| | Ours | weakly convex | $f + \omega$ / DSGD | $\mathcal{O}\left(\frac{\sqrt{m}}{\sqrt{K}} + \frac{m}{K}\right)$ |
| | | | $h \circ c + \omega$ / DSPL | $\mathcal{O}\left(\frac{1}{\sqrt{K}} + \frac{m^2}{K}\right)$ |

**Structure of the paper** Section 2 introduces the notations and problem setup. Section 3 analyzes the delayed stochastic (proximal) subgradient method (DSGD). Section 4 develops the delayed stochastic prox-linear method (DSPL). Section 5 proposes a simple strategy to make the asynchronous algorithms robust to arbitrary delays. Section 6 conducts experiments to verify our theoretical results. We draw conclusions in Section 7 and leave all the proofs and further discussions in the appendix.

## 2 Preliminaries

**Notations** We use $\|\cdot\|$ and $\langle \cdot, \cdot \rangle$ to denote the Euclidean norm and inner product. The subdifferential of a function $f$ is defined by $\partial f(x) := \{v : f(y) \geq f(x) + \langle v, y - x \rangle + o(\|x - y\|), y \to x\}$ and $f'(x) \in \partial f(x)$ denotes a subgradient. If $0 \in \partial f(x)$, then $x$ is called a stationary point of $f$. The domain of $f$ is defined by $\text{dom } f := \{x : f(x) < \infty\}$. At iteration $k$, we use $\mathbb{E}_k[\cdot]$ to denote the expectation conditioned on past iterations $\{x^1, \ldots, x^{k-1}\}$. $\mathbb{I}\{\cdot\}$ denotes the 0-1 indicator function of an event. Given a delay sequence $\{\tau_k\}$, we write $\Delta_1 := \frac{1}{K} \sum_{k=1}^K \tau_k$ and $\Delta_2 := \frac{1}{K} \sum_{k=1}^K \tau_k^2$.

Our analysis will adopt the Moreau envelope as the potential function, a technique initially identified in the work [8]. Let $f$ be a $\kappa$-weakly convex function. Given $\rho > \kappa$, the Moreau envelope and the associated proximal mapping of $f$ are respectively defined by

$$f_{1/\rho}(x) := \min_y \left\{ f(y) + \frac{\rho}{2} \|x - y\|^2 \right\} \quad \text{and} \quad \text{prox}_{f/\rho}(x) := \arg \min_y \left\{ f(y) + \frac{\rho}{2} \|x - y\|^2 \right\}.$$

By the optimality condition and convexity of $f(y) + \frac{\rho}{2} \|x - y\|^2$, $0 \in \partial f(\text{prox}_{f/\rho}(x)) + \rho(\text{prox}_{f/\rho}(x) - x)$. The Moreau envelope can be nicely interpreted as a smooth approximation of the original function. Specifically, it can be shown that $f_{1/\rho}(x)$ is differentiable and its gradient is $\nabla f_{1/\rho}(x) = \rho(x - \text{prox}_{f/\rho}(x))$ (See [8]). Combining the above two relations, we obtain $\nabla f_{1/\rho}(x) \in \partial f(\text{prox}_{f/\rho}(x))$. Therefore, the Moreau envelope can be used as a measure of approximate stationarity: if $\|\nabla f_{1/\rho}(x)\| \leq \varepsilon$, then $x$ is in the proximity (i.e. $\|x - \text{prox}_{f/\rho}(x)\| \leq \rho^{-1}\varepsilon$) of a near stationary point $\text{prox}_{f/\rho}(x)$ (i.e. $\text{dist}(\partial f(\text{prox}_{f/\rho}(x)), 0) \leq \varepsilon$).

## 2.1 Assumptions

Throughout the paper, we make the following assumptions.

**A1:** (i.i.d. sample) We draw i.i.d. samples $\{\xi^k\}$ from $\Xi$.

**A2:** (Lipschitz continuity) $\omega(x)$ is $L_\omega$-Lipschitz continuous over its domain.

**A3:** (Weak convexity) $\omega$ is $\kappa$-weakly convex.

**A4:** (Bounded moments) The distribution of the independent stochastic delays $\{\tau_k\}$ has bounded first and second moments. i.e., $\mathbb{E}[\tau_k] \leq \bar{\tau} < \infty, \mathbb{E}[\tau_k^2] \leq \tau_{\sigma^2} < \infty, \forall k$.

*Remark* 1. Assumptions **A1** to **A3** are typical in stochastic weakly convex optimization [8], while **A4** is common in distributed optimization [3, 32]. Moreover, throughout our analysis, we regard $\{\tau_k\}$ as a pre-defined random sequence independent of data sampling and the algorithm [32, 34].

## 3 Delayed proximal subgradient method

In this section, we analyze the convergence rate of the delayed stochastic proximal subgradient method (DSGD) for weakly convex optimization. DSGD is the workhorse of most applications and is frequently analyzed in the literature on centralized distributed optimization.

---

**Algorithm 1:** Delayed stochastic proximal subgradient method

**Input:** $x^1$;
**for** $k = 1, 2,...$ **do**
  Let $g^{k-\tau_k} \in \partial f(x^{k-\tau_k}, \xi^{k-\tau_k})$ be computed by a worker with delay $\tau_k$;
  In the master node update
$$x^{k+1} = \arg\min_{x \in \mathbb{R}^n} \{\langle g^{k-\tau_k}, x - x^k \rangle + \omega(x) + \tfrac{\gamma_k}{2}\|x - x^k\|^2\}. \qquad (3)$$

**end**

---

We restrict $f$ to have a bounded subgradient and make the following extra assumption.

**B1:** $f(x, \xi)$ is $\lambda$-weakly convex, $\mathbb{E}_\xi[\|g\|^2] \leq L_f^2, g \in \partial f(x, \xi)$ for all $x \in \text{dom } \omega \subseteq \text{int}(\text{dom } f)$ and all $\xi \sim \Xi$.

Now we are ready to present the convergence analysis of DSGD. Our analysis relies on Moreau envelope, denoted as $\psi_{1/\rho}(x^k)$, which serves as the potential function. After assessing the descent of the potential function with delays separated as an error term, we derive the convergence result by bounding the stochastic delays. The following lemma provides the descent property.

**Lemma 1.** *Let* $\hat{x}^k := \text{prox}_{\psi/\rho}(x^k)$. *Suppose* **A1**, **A2**, **A3** *and* **B1** *hold. If* $\rho > 2\lambda + \kappa, \gamma_k \geq \rho$, *then*

$$\frac{\rho(\rho - 2\lambda - \kappa)}{2(\gamma_k - 2\lambda - \kappa)}\|\hat{x}^k - x^k\|^2 \leq \psi_{1/\rho}(x^k) - \mathbb{E}_k[\psi_{1/\rho}(x^{k+1})] - \frac{\rho(\gamma_k - \rho)}{2(\gamma_k - 2\lambda - \kappa)}\mathbb{E}_k[\|x^{k+1} - x^k\|^2]$$
$$+ \frac{\rho\lambda}{\gamma_k - 2\lambda - \kappa}\mathbb{E}_k[\|x^{k+1} - x^{k-\tau_k}\|^2] + \frac{2\rho L_f}{\gamma_k - 2\lambda - \kappa}\mathbb{E}_k[\|x^{k+1} - x^{k-\tau_k}\|]$$

*Remark* 2. **Lemma 1** shows that aside from the noise and delay-related terms, unless we are close to approximate stationarity characterized by $\|\hat{x}^k - x^k\|^2$, there is always sufficient decrease in the potential function. Intuitively, if we take sufficiently large regularization $\gamma$ and bound the delays using **A4**, convergence is almost immediate. Now we show convergence of DSGD in **Theorem 1**.

**Theorem 1.** *Under the same conditions as* **Lemma 1** *as well as* **A4**, *taking* $\gamma_k \equiv \gamma = \rho + 4\lambda + \kappa + \sqrt{K}/\alpha$ *for some* $\alpha > 0$, *letting* $k^*$ *be chosen between 1 and K uniformly at random, then*

$$\mathbb{E}\big[\|\nabla\psi_{1/\rho}(x^{k^*})\|^2\big] \leq \frac{2\rho}{\rho - 2\lambda - \kappa}\bigg[\frac{(\rho + 2\lambda)D}{K} + \frac{D}{\sqrt{K}\alpha} + \frac{4\rho L_f(L_f + L_\omega)\alpha}{\sqrt{K}}(\Delta_1 + 1) + \frac{8\rho\lambda(L_f + L_\omega)^2\alpha^2}{K}\Delta_2\bigg],$$

*where* $D = \psi_{1/\rho}(x^1) - \inf_x \psi(x)$ *and recall our notation* $\Delta_1 = \frac{1}{K}\sum_{k=1}^{K}\tau_k, \Delta_2 = \frac{1}{K}\sum_{k=1}^{K}\tau_k^2$.

*Remark* 3. Note that $\alpha$ controls the trade-off between noise, delay and optimization. In practice we can set $\alpha$ as a hyper-parameter and tune it to improve performance.

The bound for DSGD states that

$$\mathbb{E}[\|\nabla\psi_{1/\rho}(x^{k^*})\|^2] = \mathcal{O}(\tfrac{1}{\sqrt{K}} + \tfrac{\Delta_1}{\sqrt{K}} + \tfrac{\lambda\Delta_2}{K}) \text{ or } \mathbb{E}[\|\nabla\psi_{1/\rho}(x^{k^*})\|^2] = \mathcal{O}(\tfrac{1}{\sqrt{K}} + \tfrac{\bar{\tau}}{\sqrt{K}} + \tfrac{\lambda\tau_{\sigma^2}}{K}),$$

if we take $\mathbb{E}[\Delta_1] = \bar{\tau}$ and $\mathbb{E}[\Delta_2] = \tau_{\sigma^2}$. Here $\Delta_1$ is the average delay and is considered "robust" in the distributed setting. However, there exists a second term $\lambda\Delta_2$ arising from $\lambda$-weak convexity. As we will show later, this term, in the worst case, can not be trivially bounded. Therefore, we now resort to our statistical assumption **A4** and use the first and second moments to give a bound.

## 4 Delayed stochastic prox-linear method

In this section, we present a delayed stochastic prox-linear method (DSPL) for the composition optimization problem with $f(x,\xi) = h(c(x,\xi))$, where $h(\cdot) : \mathbb{R} \to \mathbb{R}$ is a convex nonsmooth function and $c(\cdot,\xi) : \mathbb{R}^n \to \mathbb{R}$ is a smooth, potentially nonconvex function. For brevity, we denote $f_z(x,\xi) = h(c(z,\xi) + \langle\nabla c(z,\xi), x - z\rangle)$, a partial linearization of the objective, and we use these two notations interchangeably to describe DSPL. We will show that this linearization scheme improves the accuracy of approximating the objective, thus giving improved convergence rates in the presence of delay. It should be noted that while we assume $c(\cdot)$ to be a smooth function, our analysis can seamlessly extend to the case where $c$ is a smooth mapping with a bounded Jacobi matrix.

---

**Algorithm 2:** Delayed stochastic prox-linear method

---

**Input:** $x^1$;

**for** $k$ = 1, 2,... **do**

    Let $c(x^{k-\tau_k}, \xi^{k-\tau_k})$ and $\nabla c(x^{k-\tau_k}, \xi^{k-\tau_k})$ be computed by a worker with delay $\tau_k$;

    In the master node update

$$x^{k+1} = \underset{x\in\mathbb{R}^n}{\arg\min} \left\{ f_{x^{k-\tau_k}}(x, \xi^{k-\tau_k}) + \omega(x) + \tfrac{\gamma_k}{2}\|x - x^k\|^2 \right\} \qquad (4)$$

**end**

---

We describe DSPL in **Algorithm 2**. DSPL can be deployed in a multi-agent network where all the workers are connected to a master server. We assume the workers access random samples $\xi$ and compute the function value $c(x,\xi)$ and gradient $\nabla c(x,\xi)$. At the $k$-th iteration, the master node receives a delayed value $c(x^{k-\tau_k}, \xi^{k-\tau_k})$ and gradient $\nabla c(x^{k-\tau_k}, \xi^{k-\tau_k})$ of an earlier point $x^{k-\tau_k}$. Next, it performs a proximal update (4) to obtain the next iterate $x^{k+1}$. **Figure 1** illustrates the operational dynamics of DSPL and contrast it with DSGD. Throughout the rest of this paper, we assume that the proximal update is easy to compute. To derive the convergence results of DSPL, we now formally state the assumptions on $h$ and $c$.

**C1:** $h(x)$ is convex and $L_h$-Lipschitz continuous over its domain; $c(x,\xi)$ has $C(\xi)$-Lipschitz continuous gradient and is $L_c(\xi)$-Lipschitz continuous over $\mathrm{dom}\,\omega$. Moreover, we assume that $\mathbb{E}_\xi[C(\xi)^2] \leq C^2, \mathbb{E}_\xi[L_c(\xi)^2] \leq L_c^2$.

*Remark* 4. The assumption of Lipschitz continuity for $c$ is quite prevalent in the literature of weakly convex optimization [8, 12, 13]. However, it may not hold if $\mathrm{dom}\,\omega$ is unbounded and $c$ is nonlinear, such as in the case of quadratic functions. To circumvent this problem, we introduce the relative Lipschitzian property and extend the method to the Bregman proximal iteration. This involves replacing the Euclidean regularization term $\tfrac{1}{2}\|x - x_k\|^2$ in (4) with the divergence $V_d(x, x^k)$, which is generated by a strongly convex function $d(\cdot)$. The Bregman DSPL encompasses Algorithm 2 as a special case; the details are provided in the appendix for a thorough examination.

Under the above assumptions, $f_z(x,\xi)$ satisfies the following desirable properties.

**Proposition 1** (Properties of $f_z(x,\xi)$)**.**

  **P1:** $f_z(x,\xi)$ is convex for all $x, z \in \mathrm{dom}\,\omega$, and every $\xi \sim \Xi$.

  **P2:** $|f_z(x,\xi) - f(x,\xi)| \leq \frac{L_h C(\xi)}{2}\|x - z\|^2$ for all $x, z \in \mathrm{dom}\,\omega$ and $\xi \sim \Xi$.

  **P3:** $f_z(x,\xi) - f_z(y,\xi) \leq L_h L_c(\xi)\|y - x\|$ for all $x, y, z \in \mathrm{dom}\,\omega$ and all $\xi \sim \Xi$

From **Proposition 1**, we see $f(x, \xi)$ is $L_h C(\xi)$-weakly convex since the error between $f(x, \xi)$ and a convex function is bounded by a quadratic function. Furthermore, we know that $f(x)$ is $L_h L_c$ Lipschitz-continuous and $L_h C$-weakly convex after taking expectation. For a unified analysis, we take $L_f = L_h L_c, \lambda = L_h C$ and use these constants to present the results. The following lemma characterizes the descent property for our potential function in DSPL.

**Lemma 2.** *Suppose* **A1**, **A2**, **A3** *and* **C1** *hold, if* $\rho > 2\lambda + \kappa, \gamma_k \geq \rho$, *then*

$$\frac{\rho(\rho - 2\lambda - \kappa)}{2(\gamma_k - 2\lambda - \kappa)} \|\hat{x}^k - x^k\|^2 \leq \psi_{1/\rho}(x^k) - \mathbb{E}_k[\psi_{1/\rho}(x^{k+1})] - \frac{\rho(\gamma_k - \rho)}{2(\gamma_k - 2\lambda - \kappa)} \mathbb{E}_k[\|x^{k+1} - x^k\|^2]$$
$$+ \frac{2\rho L_f^2}{(\gamma_k - \kappa)(\gamma_k - 2\lambda - \kappa)} + \frac{3\rho\lambda}{2(\gamma_k - 2\lambda - \kappa)} \mathbb{E}_k[\|x^{k+1} - x^{k-\tau_k}\|^2].$$

Similar to the analysis of DSGD, we bound the delays to give the convergence result.

**Theorem 2.** *Under the same conditions as* **Lemma 2** *as well as* **A4**, *taking* $\gamma_k \equiv \gamma = \rho + 6\lambda + \kappa + \sqrt{K}/\alpha$ *for some* $\alpha > 0$, *letting* $k^*$ *be uniformly chosen between* 1 *and* $K$, *then*

$$\mathbb{E}[\|\nabla\psi_{1/\rho}(x^{k^*})\|^2] \leq \frac{2\rho}{\rho - 2\lambda - \kappa}\left[\frac{(\rho + 4\lambda)D}{K} + \frac{D}{\sqrt{K}\alpha} + \frac{2\rho L_f^2 \alpha}{\sqrt{K}} + \frac{6\rho\lambda(L_f + L_\omega)^2\alpha^2}{K}\Delta_2\right],$$

*where* $D = \psi_{1/\rho}(x^1) - \inf_x \psi(x)$.

If we take $\mathbb{E}[\Delta_2] = \tau_{\sigma^2}$, then Theorem 2 implies that

$$\mathbb{E}[\|\nabla\psi_{1/\rho}(x^{k^*})\|^2] = \mathcal{O}(\frac{1}{\sqrt{K}} + \frac{\lambda\Delta_2}{K}) \ \text{ or } \ \mathbb{E}[\|\nabla\psi_{1/\rho}(x^{k^*})\|^2] = \mathcal{O}(\frac{1}{\sqrt{K}} + \frac{\lambda\tau_{\sigma^2}}{K}).$$

Compared to DSGD, the delays for DSPL appear only in a higher-order term with respect to $K$. Different from DSGD where $\lambda$ only characterizes weak convexity, $\lambda$ for DSPL also represents the quadratic upper-bound on $f(x, \xi)$ in **P2**, which is not 0 even if the problem is convex.

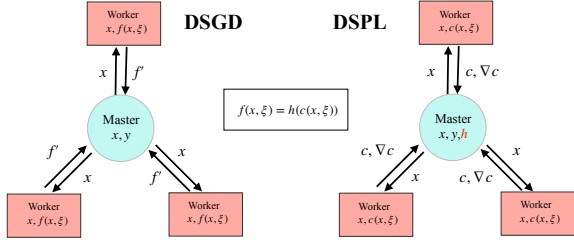

Figure 1: DSGD and DSPL in a master-worker architecture.

**DSPL vs. DSGD** We give some further insights on the behavior of DSPL and DSGD for the composition function (2). Intuitively, as the algorithm converges, we have $\lim_{k\to0}\|x^k - x^{k+1}\| = 0$ a.s. When $x^k \approx x^{k-\tau_k}$, DSPL enjoys an increasingly stable estimation of the proximal mapping (4), as the influence of delay is diminishing and the error is mainly driven by stochastic sampling. The same conclusion holds on smooth DSGD due to the Lipschitz continuity of $\nabla f$. On the other hand, when DSGD is applied for a nonsmooth problem, the master node receives an out-of-date subgradient $f'$ from the worker and solves (3). Since $f(x, \xi)$ is nonsmooth, the subgradient $f'(x^{k-\tau_k}, \xi)$ may differ significantly from $f'(x^k, \xi)$ even when the sequence $\{x^k\}$ converges. Hence, DSGD will constantly suffer from delay during all the updates (3).

**DSPL with momentum** We also remark that the momentum technique from DSGD can be extended to DSPL, which gives us the same $\mathcal{O}(\tau_{\max}/\sqrt{K} + \tau_{\max}^2/K)$ convergence rate as in [34]. We refer the interested readers to the Appendix F.

When $f$ is smooth, the analysis of DSPL can be adapted to yield a comparable convergence rate for the proximal stochastic gradient method for minimizing $\psi(\cdot)$.

**Theorem 3.** *Suppose all the assumptions in* **Lemma 1** *and* **A4** *hold, and that $f$ is $\lambda$-smooth. Let* $\gamma_k \equiv \gamma = \rho + 6\lambda + \kappa + \sqrt{K}/\alpha$ *for some $\alpha > 0$, then* $\mathbb{E}[\|\nabla\psi_{1/\rho}(x^{k^*})\|^2] = \mathcal{O}(\frac{1}{\sqrt{K}} + \frac{\lambda\Delta_2}{K})$.

So far we have spent two sections analyzing the two algorithms so that enough intuition can be established. All these serve for our ultimate goal: making DSGD and DSPL robust to arbitrary delays.

# 5 Weakly convex optimization robust to arbitrary delays

This section proposes robust variants of DSGD and DSPL, for which the explicit delays are eliminated from the convergence rate. What we will do is reduce the impact of delay to *the number of agents in the network*. Moving forward, we substitute **A4** with **D1**.

**D1:** The distributed environment has $m$ workers.

The previously established results have provided sufficient intuition on how delays impact our algorithms. In view of **Theorem 1** and **Theorem 2**, we have isolated delay-dependent $\mathcal{O}(\Delta_1/\sqrt{K})$ and $\mathcal{O}(\Delta_2/K)$ in the proof. Although $\Delta_1/\sqrt{K}$ seems larger, it turns out that $\Delta_2$ stands in our way. The following lemma shows $\Delta_1 = \frac{1}{K}\sum_{k=1}^{K}\tau_k$ is bounded by $m$.

**Lemma 3.** *In a distributed environment of $m$ workers* $\Delta_1 = \frac{1}{K}\sum_{k=1}^{K}\tau_k \leq m - 1 \leq m$.

Given **Lemma 3**, we know $\Delta_1/\sqrt{K} = \mathcal{O}(m/\sqrt{K})$ and we can replace dependency on $\Delta_1$ by $m$. Now we consider $\Delta_2 = \frac{1}{K}\sum_{k=1}^{K}\tau_k^2$. Even if we have a larger denominator $K$ neutralizing the effect of $\Delta_2$, the following example shows that in the worst case, $\Delta_2$ can be of $\mathcal{O}(K)$ and result in $\mathcal{O}(1)$ error.

**Example 1** (Why $\Delta_2$ hurts performance). *Given sequence of delays $\{\tau_k\}$ such that $\tau_k = 0, k \leq K - 1, \tau_K = K$. Then $\Delta_2 = K$ and $\Delta_2/K = 1$.*

The example tells that delays of $\mathcal{O}(K)$ ruin our convergence. In other words, to recover an overall $\mathcal{O}(1/\sqrt{K})$ convergence rate, we need $\Delta_2/K = \mathcal{O}(1/\sqrt{K}) \Rightarrow \sum_{k=1}^{K}\tau_k^2 = \mathcal{O}(K^{3/2})$. The next lemma provides a hint for our algorithm design.

**Lemma 4.** *If a sequence of nonnegative integers $\{\tau_k\}$ satisfy $\sum_{k=1}^{K}\tau_k \leq mK$, then given $T \geq 0$,*

1. *if $\tau_k \leq T$ for all $k$, then $\sum_{k=1}^{K}\tau_k^2 \leq mKT$;*

2. $\sum_{k=1}^{K}\mathbb{I}\{\tau_k \leq T\} \geq K - mKT^{-1}$.

**Lemma 4** tells us two facts about a nonnegative integer sequence of length $K$ with sum bounded by $\mathcal{O}(K)$: **1)**. if we restrict the elements to be less than $\mathcal{O}(T)$, then $\sum_{k=1}^{K}\tau_k^2 \leq \mathcal{O}(KT)$. **2)**. there will be $\Omega(K - mKT^{-1})$ elements bounded by $T$. Back to our context, this implies **1)**. To reduce $\Delta_2/K$ to $\mathcal{O}(1/\sqrt{K})$, we can discard the iterations of delays greater than $T = \mathcal{O}(\sqrt{K})$. **2)**. We skip no more than $\mathcal{O}(\sqrt{K})$ iterations and optimization works with $\mathcal{O}(K - \sqrt{K}) = \mathcal{O}(K)$ iterations left.

Having established the foundational understanding, we outline the main steps in **Algorithm 3**. It is based on the aforementioned intuition and incorporates a "safeguarding" step to discard inaccurate information which could potentially hurt the convergence performance. As the above argument on the accumulated delay is independent of any specific algorithm, the safeguarding strategy can be applied to both DSGD and DSPL. To make our parameter setting more general, in our analysis we consider $T = r^{-1}mK^\beta, r > 0, \beta \geq 0$.

---

**Algorithm 3:** Safeguarded DSGD/DSPL

**Input:** $x^1, T = r^{-1}mK^\beta$;
**for** $k = 1, 2,...$ **do**
    **if** $\tau_k \leq T$ **then**
        Update with
            (3) for DSGD or (4) for DSPL
    **else**
        $x^{k+1} = x^k$
    **end**
**end**

---

Intuitively, under worst-case scenarios, **Algorithm 3** will perform after $K$ iterations in a manner similar to its non-safeguarded counterpart. However, the maximum delay will be capped at $r^{-1}mK^\beta$, and the iteration count will be $K(1 - rK^{-\beta})$.

**Theorem 4** (Safeguarded DSGD). *Under the same conditions as **Lemma 1** as well as **D1**, taking **1)** $\beta > 0, K > r^{1/\beta}$ or **2)** $\beta = 0, r < 1$, then letting $\gamma_k \equiv \gamma = \frac{\sqrt{K}}{\alpha\sqrt{\eta}} + \rho + \kappa + 4\lambda, \eta = 1 + \frac{r}{K^\beta - r}$ for*

*some* $\alpha > 0$ *and* $k^*$ *be uniformly chosen between iterations where* $\tau_k \leq T = r^{-1}mK^\beta$,

$$\mathbb{E}[\|\nabla\psi_{1/\rho}(x^{k^*})\|^2] \leq \frac{2\rho}{\rho-2\lambda-\kappa}\left[\frac{\eta(\rho+2\lambda)D}{K} + \frac{\sqrt{\eta}D}{\sqrt{K}\alpha} + \frac{4\eta^{3/2}\rho L_f(L_f+L_\omega)m\alpha}{\sqrt{K}} + \frac{8\eta^2\rho\lambda(L_f+L_\omega)^2m^2\alpha^2}{rK^{1-\beta}}\right],$$

*where* $D = \psi_{1/\rho}(x^1) - \inf_x \psi(x)$.

**Theorem 5** (Safeguarded DSPL). *Under the same conditions as* **Lemma 2** *as well as* **D1**, *taking 1)* $\beta > 0, K > r^{1/\beta}$ *or 2)* $\beta = 0, r < 1$, *then letting* $\gamma_k \equiv \gamma = \frac{\sqrt{K}}{\alpha\sqrt{\eta}} + \rho + \kappa + 6\lambda, \eta = 1 + \frac{r}{K^\beta - r}$ *for some* $\alpha > 0$ *and* $k^*$ *be uniformly chosen between iterations where* $\tau_k \leq T = r^{-1}mK^\beta$,

$$\mathbb{E}[\|\nabla\psi_{1/\rho}(x^{k^*})\|^2] \leq \frac{2\rho}{\rho-2\lambda-\kappa}\left[\frac{\eta(\rho+4\lambda)D}{K} + \frac{\sqrt{\eta}D}{\sqrt{K}\alpha} + \frac{2\sqrt{\eta}\rho L_f\alpha}{\sqrt{K}} + \frac{6\eta^2\rho\lambda(L_f+L_\omega)^2m^2\alpha^2}{rK^{1-\beta}}\right].$$

*where* $D = \psi_{1/\rho}(x^1) - \inf_x \psi(x)$.

*Remark* 5. **Theorem 4** and **5** show that by employing a safeguarding step, both DSGD and DSPL can achieve delay-independent rates. It is also interesting to see how the choice of $\beta$ and $r$ affects performance. To recover a convergence rate of $\mathcal{O}(K^{-1/2})$, it is sufficient to have $\beta \leq 1/2$. If we set $\beta > 0$, then we skip $rK^{-\beta}$ of all the iterations and incur a penalty of up to $\eta^{3/2}$ in DSGD and $\eta^{1/2}$ in DSPL. However, this loss becomes negligible for large $K$, as $\eta = 1 + \frac{r}{K^\beta - r} \to 1$. Alternatively, if we set $\beta = 0$, DSGD achieves $\mathcal{O}(\sqrt{m}/\sqrt{K} + m/K)$ rate with $\alpha = 1/\sqrt{m}$ while DSPL yields a rate of $\mathcal{O}(1/\sqrt{K} + m^2/K)$ with $\alpha = 1$. This setting aligns with [7, 18] and achieves optimal rate on $m$. However, the penalty from $\eta > 1$ is non-negligible and adversely affects the overall convergence rate by a constant factor. Therefore, we can strike a balance in practice by choosing $\beta$ in $(0, 1/2]$ and allow delays of up to $\mathcal{O}(K^\beta)$.

# 6 Experiments

This section performs numerical experiments on the robust phase retrieval problem to demonstrate the efficiency of our methods. Given a measuring matrix $A \in \mathbb{R}^{m \times n}$ and a set of observations $b_i \approx |\langle a_i, \hat{x}\rangle|^2, 1 \leq i \leq m$ ($m$ in this section represents the number of samples), robust phase retrieval aims to recover the true signal $\hat{x}$ from

$$\min_{x \in \mathbb{R}^n} \frac{1}{m}\sum_{i=1}^m |\langle a_i, x\rangle^2 - b_i| + \delta_{\{x:\|x\|\leq M\}},$$

where $\delta_S$ denotes the set indicator function and the $\ell_1$ loss function improves robustness. Our experiment contains three parts. The first part profiles algorithms in an asynchronous environment simulated via MPI; our second experiment runs sequentially with simulated delays from common distributions; our last experiment also runs in the simulated environment and demonstrates the effectiveness of safeguarding step under adversarially chosen delays.

## 6.1 Experiment setup

**Synthetic data**. For the synthetic data, we take $m = 300, n = 100$ in the experiments of simulated delay and $m = 1500, n = 500$ in the asynchronous environment. Data generation follows the setup of [11], where, given some $\kappa \geq 1$, we compute $A = QD, Q \in \mathbb{R}^{m \times n}, q_{ij} \sim \mathcal{N}(0, 1)$ and $D = \text{diag}(d), d \in \mathbb{R}^n, d_i \in [1/\kappa, 1]$ for all $i$. Then we generate a true signal $\hat{x} \sim \mathcal{N}(0, I)$ and obtain the measurements $b$ using formula $b_i = \langle a_i, \hat{x}\rangle^2$. Last we randomly choose $p_{\text{fail}}$-fraction of the measurements and add $\mathcal{N}(0, 25)$ to them to simulate data corruption.

**Real-life data**. The real-life data is generated from `zipcode` dataset, where we vectorize a $16\times16$ hand-written digit from [16] and use it as the signal. The measuring matrix $A$ comes from a normalized Hadamard matrix $H \in \mathbb{R}^{256 \times 256}$: we generate three diagonal matrices $S_j = \text{diag}(s_j), j = 1, 2, 3$; each element of $s_j \in \mathbb{R}^{256}$ is taken from $\{-1, 1\}$ randomly and we let $A = H[S_1, S_2, S_3]^{\text{T}} \in \mathbb{R}^{768 \times 256}$. Finally $p_{\text{fail}}$-fraction of the observations are set 0.

1) **Dataset**. In the asynchronous environment, we keep up with [34] setting $\kappa = 1, p_{\text{fail}} = 0$ and in the simulated environment, we follow [11] setting $\kappa \in \{1, 10\}$ and $p_{\text{fail}} \in \{0.2, 0.3\}$.

2) **Initial point and radius**. Synthetic data: we generate $x' \sim \mathcal{N}(0, I_n)$ and start from $x^1 = \frac{x'}{\|x'\|}$; `zipcode` data: we generate $x' \sim \mathcal{N}(\hat{x}, I_n)$ and take $x^1 = 10x'$. $M = 1000\|x^1\|$.

3) **Stopping criterion**. We run algorithms for 400 epochs ($K = 400m$). In the asynchronous environment, algorithms run until reaching the maximum iteration. In the simulated environment, algorithms stop if $f \leq 1.5f(\hat{x})$. When $f$ contains corrupted measurements, $f(\hat{x}) > 0$.

4) **Stepsize**. We set $\gamma = \sqrt{K/\alpha}$, where $\alpha \in \{0.1, 0.5, 1.0\}$ in the asynchronous environment, $\alpha \in [10^{-2}, 10^1]$ for synthetic data and $\alpha \in [10^1, 10^2]$ for the `zipcode` dataset.

5) **Simulated delay**. In the simulated environment, we generate $\tau_k$ from two common distributions from literature, which are geometric $\mathcal{G}(p)$ and Poisson $\mathcal{P}(\lambda)$ [37]. After the delay is generated, it is truncated by twice the mean of the distribution.

6) **Adversarial delay**. We let delay happen at the last iteration of each epoch and use the information of $x^1$ to update. Safeguarding parameter is set to $T = 0.1\sqrt{K}$.

7) **Trade-off between computation and communication**. In the asynchronous environment, the numerical linear algebra on the worker uses a raw implementation (not importing package) to balance the cost of gradient computation and communication.

## 6.2 Asynchronous environment

Our first experiment runs in an asynchronous environment implemented by MPI `Python` interface and is profiled on an `Intel(R) Xeon(R) CPU E5-2640 v4 @ 2.40GHz` machine with 10 cores and 20 threads. This experiment runs on a single machine to verify our theoretical finding rather than to test the algorithm's real performance on a specific distributed architecture.

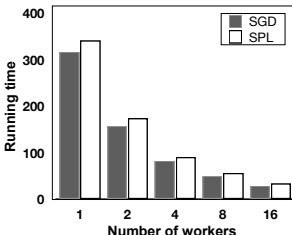 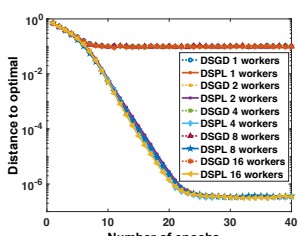 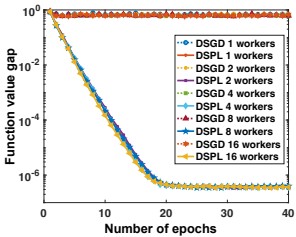

Figure 2: First: speedup in time and the number of workers. Second: progress of $\|x^k - \hat{x}\|$ in the first 40 epochs given $\alpha = 0.1$. Third: $f(x^k) - f(\hat{x})$ in the first 40 epochs given $\alpha = 0.5$. For more details about the two figures on the right, please refer to Figure 12 in the appendix.

The first figure plots the wall-clock time (in seconds) for `DSGD` and `DSPL` to complete 400 epochs when the number of workers increases. It is observed that both algorithms exhibit speed-up with more workers and note that `DSPL` takes more time than `DSGD` due to the need to pass the function value to the master and slightly more expensive updates. But as the second and the third figure suggest, this extra cost is justified by the superior convergence: in the first several epochs `DSPL` reaches a high accuracy of $10^{-6}$ in both function value and distance to the optimal solution, while `DSGD` stagnates at a relatively low-accuracy solution of $10^{-2}$. These observations suggest that `DSPL` offers better convergence behavior than the methods only based on subgradient. Finally, our experiments suggest both `DSGD` and `DSPL` are not sensitive to the increase in the number of workers when there are relatively few workers.

## 6.3 Simulated environment

The second part of our experiment compares the performance between `DSGD` and `DSPL` and is based on the simulated delay, where the algorithm runs sequentially but the gradient information is computed from the previous iterates.

Figure 3 plots the number of iterations for each algorithm to converge under different datasets and delay parameters. We see that in spite of delays, `DSPL` admits a wider range of stepsize parameters ensuring convergence than `DSGD`, and the performance slightly deteriorates as delay increases.

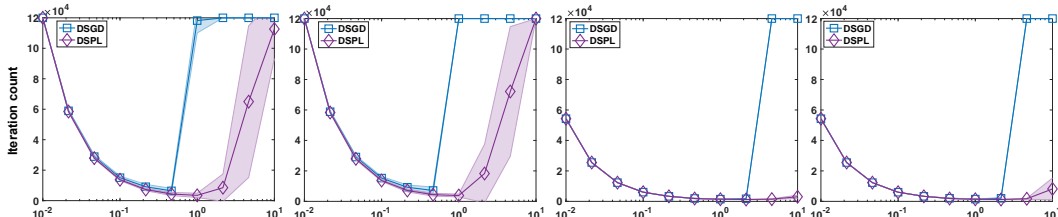

Figure 3: Left two: Geometric delay $(\kappa, p_{\text{fail}}) = (1, 0.3), 2p^{-1} \in \{28, 47\}$; right two: Poisson delay $(\kappa, p_{\text{fail}}) = (10, 0.2), 2\lambda \in \{28, 47\}$. x-axis: parameter $\alpha$; y-axis: number of iterations. Definition of delay is given in experiment setup 5).

## 6.4 Adversarial delay

The final experiment verifies the efficacy of our safeguarding step when delays are introduced in an adversarial setting. We evaluate the performance of DSGD, DSPL, both with and without the safeguarding step, under the delay patterns we have generated.

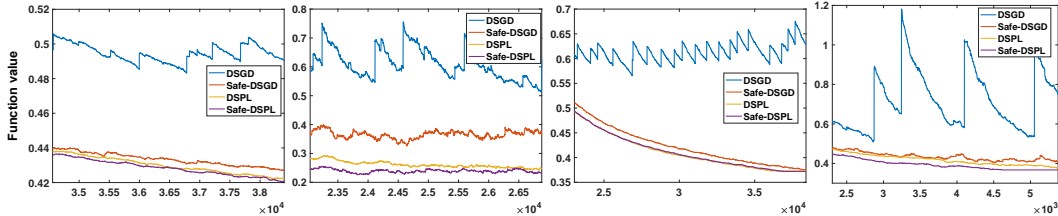

Figure 4: From left to right: zipcode data, $p_{\text{fail}} = 0.2, \alpha \in \{10, 100\}$, $p_{\text{fail}} = 0.3, \alpha \in \{10, 100\}$. x-axis: iteration count; y-axis: $f(x^k)$.

Figure 4 clearly illustrates the superior performance of our method incorporating the safeguarding step. As suggested by our theory, DSGD is notably sensitive to delays of $\mathcal{O}(K)$. However, upon discarding outdated information, DSGD exhibits much greater robustness. Interestingly, even under our adversarial setup, the performance of DSPL remains acceptable, albeit slightly inferior to its safeguarded version, which aligns well with our theoretical findings.

## 7 Conclusions

We offer a sharp analysis of delayed stochastic algorithms for weakly convex optimization, discussing the widely utilized DSGD method and introducing the novel DSPL method for problems with a composition structure. Through careful examination of delay factors, we propose a straightforward safeguarding approach to eliminate the effect of delays, and instead derive bounds depending on the number of agents in the distributed environment. This makes our algorithms resilient to arbitrary delays. A promising future direction is the application of these prox-linear methods in more diverse distributed settings, such as decentralized networks.

## 8 Acknowledgement

The authors thank the anonymous reviewers for their constructive suggestions. This research is partially supported by National Natural Science Foundation of China (NSFC-72150001, 11831002).

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

# Appendix

## Table of Contents

**Structure of the appendix** In this paper, we strive to provide a comprehensive treatment of delayed stochastic algorithms for stochastic weakly convex optimization in both theoretical and practical aspects.

**1) Theoretically**, we choose to employ a more general Bregman divergence context in our proof. This broader context covers the primary results for the Euclidean setting discussed in the main section. Specifically, this context will be invoked in our analysis for `DSPL` method to accommodate real-life applications of prox-linear method and by relaxing the Lipschitz condition **C1**.

**2) Practically**, we provide a detailed discussion on how to implement the DSPL algorithm, both in Euclidean and Bregman context. We specifically address the selection of the Bregman divergence kernel and propose efficient subroutines to solve proximal subproblems. Additionally, in view of the wide usage of momentum techniques in stochastic optimization, we analyze DSEPL, the momentum variant of DSPL. While this is not the central contribution of our paper, we believe it represents an important step for making DSPL applicable to real-world problems.

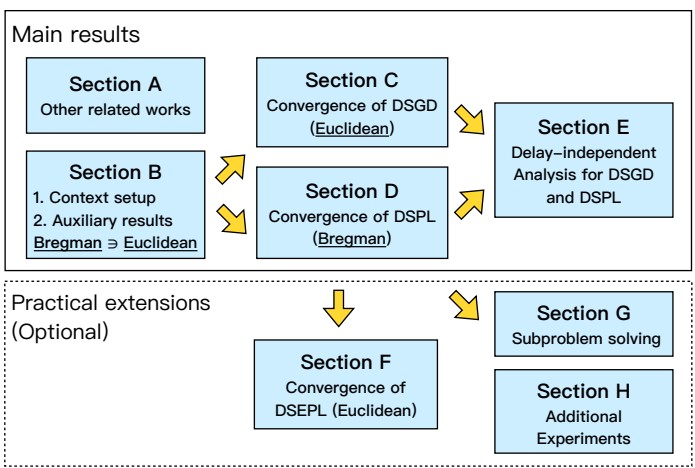

Figure 5: Structure of the appendix

The appendix is organized as follows.

The first five sections address the theoretical aspects of delayed stochastic algorithms.

In Section A, we first conduct an extended review of other related works. In Section B, we introduce the Bregman context with some auxiliary results. In Section C, we perform convergence analysis for the DSGD method, and Section D presents a more general Bregman DSPL that relaxes the Lipschitzness assumption. Section E presents the safeguarded variant of our algorithms that are robust to arbitrary delays.

**(Optional)** The last three sections are devoted to more practical aspects.

In Section F, we present the convergence analysis of DSEPL, namely, the momentum variant of DSPL using variable extrapolation. Section G discusses how to construct the kernel and solve the Bregman proximal subproblems. Finally, Section H displays additional experiment results.

## A    More on the related works

First, we review the literature on stochastic weakly-convex optimization. The Prox-Linear (PL) method [19, 12, 15] has received more attention lately. The seminal work [8] conducts a novel complexity analysis using the Moreau envelope as the potential function. They show that SGD (and many other stochastic algorithms) achieves an $\mathcal{O}(\frac{1}{\varepsilon^4})$ complexity in terms of convergence to the proximity of approximate stationary points. [20] analyzes an incremental subgradient method for finite-sum problems. [36] extends the prox-linear algorithm to the finite-sum setting and provides analysis based on variance-reduction. SGD with momentum is studied in [24] and [11] reveals that SPL can be accelerated by minibatching even for nonsmooth objective functions. [11] also employs heavy-ball momentum to further improve the model-based optimization.

There is substantial literature on distributed and asynchronous optimization. We refer to the seminal work [4]. In addition to the aforementioned asynchronous issues, another important research direction concerns reducing the network communication cost by gradient compression. For example, see [17]. Besides the centralized setting, there is also growing interest in decentralized algorithms. Recently, [6] presents a decentralized SGD in the network where nodes exchange parameters and subgradients locally. However, their method requires all the nodes to update synchronously and only

proves asymptotic convergence of SGD. An extensive study of decentralized optimization is beyond our work. Therefore, we refer interested readers to [22, 23, 35] for more recent advances.

# B  Bregman context and auxiliary results

We initiate the theoretical discussion by introducing the Bregman context, which includes the concepts of Bregman divergence, relative Lipschitzian property, and Bregman Moreau envelope. Following the provision of requisite definitions, we present some auxiliary results that will be used frequently in our analysis.

## B.1  Context setup

**Bregman divergence and relative Lipschitzness**  Given a Legendre function $d(\cdot)$ [7] that is proper, closed, strictly convex and essentially smooth, it induces the Bregman divergence defined by

$$V_d(x, y) := d(x) - d(y) - \langle \nabla d(y), x - y \rangle$$

and $d$ is called the kernel of $V_d$. Since $d$ is strictly convex, we know $V_d(x, y) \geq 0$ and the equality holds if and only if $x = y$. The notion of Bregman divergence greatly enhances the coverage of the algorithms and has recently been adopted in the context of weakly convex optimization [8, 3] to tackle a broader class of nonsmooth nonconvex optimization problems. In this paper, we mainly leverage the concept of relative Lipschitzian property to extend the analysis of delayed stochastic algorithms. First, we give its formal definition.

**Definition 1** (Relative Lipschitzian property). *[3] A function $f$ satisfies $L$-relative Lipschitzian property to kernel $d$ if for all $x, y \in \mathrm{dom}\, d$,*

$$f(x) - f(y) \leq L\sqrt{2V_d(y, x)}. \tag{5}$$

**Bregman proximal mapping and Bregman envelope**

As in conventional weakly convex optimization, we need an approximate measure of stationarity in the non-Euclidean setting. A naturally choice is the Bregman envelope [3]. Assume that $d$ is 1-strongly convex, then for any $\rho > 0$, the Bregman envelope of $f$ with respect to kernel $d$ is

$$\psi_{1/\rho}^d(x) := \min_y \{f(y) + \rho V_d(y, x)\}$$

and the Bregman proximal mapping is represented using

$$\mathrm{prox}_{\psi/\rho}^d(x) := \arg\min_y \{f(y) + \rho V_d(y, x)\}.$$

We sometimes refer to $\mathrm{prox}_{\psi/\rho}^d(x)$ as $\hat{x}$ and proximal distance $\mathbb{E}[V_d(\hat{x}^k, x^k)]$ has been proven a proper measure of the convergence of stochastic algorithms [3] under specific choice of $d$ and $\rho$.

**Connection with Euclidean geometry**  One advantage of using Bregman context is that it subsumes the Euclidean case. If we take $d(x) = \frac{1}{2}\|x\|^2$, then $V_d(x, y) = \frac{1}{2}\|x - y\|^2$ and relative Lipschitzian property becomes standard Lipschitz condition. Particularly we have the following relation between $V_d(\hat{x}, x)$ and Moreau envelope.

$$V_d(\hat{x}^k, x^k) = \frac{1}{2}\|\hat{x}^k - x^k\|^2 = \frac{1}{2\rho^2}\|\nabla \psi_{1/\rho}^d(x)\|^2.$$

When the context is clear, we will use $\psi = \psi^d$ directly if $d(x) = \frac{1}{2}\|x\|^2$. In the next section, we present several useful auxiliary results that appear frequently in the proof.

## B.2  Auxiliary lemmas

In this section, we present all the auxiliary lemmas that will be used in the proof of our main results. While some of these lemmas are widely recognized in the field, we reproduce them here to ensure that our work remains self-contained. To begin, we introduce the well-known three-point lemma.

**Lemma 5** (Three point lemma). *Let $f$ be a closed convex function and define*

$$z = \arg\min_x \{f(x) + \gamma V_d(x, y)\}.$$

*for some $y \in \text{dom}\, d$. Then we have*

$$f(z) + \gamma V_d(z, y) \leq f(x) + \gamma V_d(x, y) - \gamma V_d(x, z), \forall x \in \text{dom}\, d.$$

The following lemma bounds, for a deterministic function, the proximal step by the regularization term and relative Lipschitzian property.

**Lemma 6** (Bounding the proximal step). *Given the divergence $V_d$ generated by some 1-strongly convex function $d$. Let the function $f(x)$ satisfy the relative Lipschitzian property (5). For any $\gamma > 0$ such that $f + \gamma d$ is strongly convex, if we define*

$$x^+ = \arg\min_y \{f(y) + \gamma V_d(y, x)\},$$

*then*

$$\sqrt{V_d(x^+, x)} \leq \sqrt{2}\gamma^{-1}L \tag{6}$$

*and*

$$\|x^+ - x\| \leq 2\gamma^{-1}L. \tag{7}$$

*Proof.* By the optimality of $x^+$ and relative Lipschizian property, we have

$$f(x^+) + \gamma V_d(x^+, x) \leq f(x)$$

and

$$\gamma V_d(x^+, x) \leq f(x) - f(x^+) \leq L\sqrt{2V_d(x^+, x)}. \tag{8}$$

Dividing both sides of (8) by $\sqrt{V_d(x^+, x)}$ gives (9); (10) uses $V_d(x^+, x) \geq \frac{1}{2}\|x^+ - x\|^2$. $\qquad\square$

**Lemma 7** (Bounding the proximal step in expectation). *Given the divergence $V_d$ generated by some 1-strongly convex function $d$. Let convex stochastic function $f(x, \xi)$ satisfy the relative Lipschitzian property (5) with $L = L(\xi)$ for $\xi \sim \Xi$. For any $\gamma > 0$, if we define*

$$x^+ = \arg\min_y \{f(y, \xi) + \gamma V_d(y, x)\},$$

*then*

$$\frac{1}{\sqrt{2}}\mathbb{E}_{\xi \sim \Xi}[\|x^+ - x\|] \leq \mathbb{E}_{\xi \sim \Xi}[\sqrt{V_d(x^+, x)}] \leq \sqrt{2}\gamma^{-1}\mathbb{E}_{\xi \sim \Xi}[L(\xi)] \tag{9}$$

*and*

$$\mathbb{E}_{\xi \sim \Xi}[\|x^+ - x\|^2] \leq 4\gamma^{-2}\mathbb{E}_{\xi \sim \Xi}[L(\xi)^2]. \tag{10}$$

*Proof.* We can apply **Lemma 6**. Then we have, for each $\xi \sim \Xi$, that

$$\sqrt{V_d(x^+, x)} \leq \sqrt{2}\gamma^{-1}L(\xi) \quad \text{and} \quad V_d(x^+, x) \leq 2\gamma^{-2}L^2(\xi).$$

Taking expectation with respect to $\xi$, the fact that $V_d(x^+, x) \geq \frac{1}{2}\|x^+ - x\|^2$ completes the proof $\quad\square$

## C  Convergence Analysis of DSGD

In this section, we present the convergence analysis of DSGD. The proof follows a standard inexact potential reduction scheme and is done in the Euclidean setup, where **Lemma 5** and **Lemma 7** hold for $d = \frac{1}{2}\|\cdot\|^2$.

## C.1 Proof of Lemma 1

Since $\rho > \lambda + \kappa$ and $\gamma_k \geq \rho$, by three-point lemma we get the following two relations

$$\langle g^{k-\tau_k}, x^{k+1} \rangle + \omega(x^{k+1}) + \frac{\gamma_k}{2}\|x^{k+1} - x^k\|^2$$
$$\leq \langle g^{k-\tau_k}, \hat{x}^k \rangle + \omega(\hat{x}^k) + \frac{\gamma_k}{2}\|\hat{x}^k - x^k\|^2 - \frac{\gamma_k - \kappa}{2}\|\hat{x}^k - x^{k+1}\|^2 \tag{11}$$

$$f(\hat{x}^k) + \omega(\hat{x}^k) + \frac{\rho}{2}\|\hat{x}^k - x^k\|^2 \leq f(x^{k+1}) + \omega(x^{k+1}) + \frac{\rho}{2}\|x^{k+1} - x^k\|^2 \tag{12}$$

Summing up (11) and (12) and taking expectation, we have

$$\frac{\gamma_k - \rho}{2}\mathbb{E}_k[\|x^{k+1} - x^k\|^2] - \frac{\gamma_k - \rho}{2}\|\hat{x}^k - x^k\|^2 + \frac{\gamma_k - \kappa}{2}\mathbb{E}_k[\|\hat{x}^k - x^{k+1}\|^2]$$
$$\leq \mathbb{E}_{\xi^{k-\tau_k}}[\langle g^{k-\tau_k}, \hat{x}^k - x^{k-\tau_k} \rangle] - f(\hat{x}^k) + \mathbb{E}_k[f(x^{k+1})] - \mathbb{E}_k[\langle g^{k-\tau_k}, x^{k+1} - x^{k-\tau_k} \rangle]$$
$$= \mathbb{E}_{\xi^{k-\tau_k}}[\langle g^{k-\tau_k}, \hat{x}^k - x^{k-\tau_k} \rangle] + f(x^{k-\tau_k}) - f(\hat{x}^k)$$
$$\quad + \mathbb{E}_k[f(x^{k+1})] - f(x^{k-\tau_k}) - \mathbb{E}_k[\langle g^{k-\tau_k}, x^{k+1} - x^{k-\tau_k} \rangle]$$
$$\leq \frac{\lambda}{2}\|\hat{x}^k - x^{k-\tau_k}\|^2 + 2L_f\mathbb{E}_k[\|x^{k+1} - x^{k-\tau_k}\|] \tag{13}$$

where (13) follows from $\lambda$-weak convexity of $f(x, \xi)$ and **B1** (using [1]). Re-arranging the terms, we have

$$\frac{\gamma_k - \kappa}{2}\mathbb{E}_k[\|\hat{x}^k - x^{k+1}\|^2]$$
$$\leq \frac{\gamma_k - \rho}{2}\|\hat{x}^k - x^k\|^2 + \frac{\lambda}{2}\|\hat{x}^k - x^{k-\tau_k}\|^2$$
$$\quad + 2L_f\mathbb{E}_k[\|x^{k+1} - x^{k-\tau_k}\|] - \frac{\gamma_k - \rho}{2}\mathbb{E}_k[\|x^{k+1} - x^k\|^2]$$
$$\leq \frac{\gamma_k - \rho}{2}\|\hat{x}^k - x^k\|^2 + \lambda\mathbb{E}_k[\|x^{k+1} - x^{k-\tau_k}\|^2] + \lambda\mathbb{E}_k[\|\hat{x}^k - x^{k+1}\|^2]$$
$$\quad + 2L_f\mathbb{E}_k[\|x^{k+1} - x^{k-\tau_k}\| - \frac{\gamma_k - \rho}{2}\mathbb{E}_k[\|x^{k+1} - x^k\|^2] \tag{14}$$

where (14) follows by Cauchy's inequality $\|a + b\|^2 \leq 2\|a\|^2 + 2\|b\|^2$. Now we re-arrange the terms and divide both sides by $\gamma_k - 2\lambda - \kappa$ to derive

$$\mathbb{E}_k[\|\hat{x}^k - x^{k+1}\|^2]$$
$$\leq \frac{\gamma_k - \rho}{\gamma_k - 2\lambda - \kappa}\|\hat{x}^k - x^k\|^2 + \frac{2\lambda}{\gamma_k - 2\lambda - \kappa}\mathbb{E}_k[\|x^{k+1} - x^{k-\tau_k}\|^2]$$
$$\quad + \frac{4L_f}{\gamma_k - 2\lambda - \kappa}\mathbb{E}_k[\|x^{k+1} - x^{k-\tau_k}\|] - \frac{\gamma_k - \rho}{\gamma_k - 2\lambda - \kappa}\mathbb{E}_k[\|x^{k+1} - x^k\|^2]$$
$$= \|\hat{x}^k - x^k\|^2 - \frac{\rho - 2\lambda - \kappa}{\gamma_k - 2\lambda - \kappa}\|\hat{x}^k - x^k\|^2 + \frac{2\lambda}{\gamma_k - 2\lambda - \kappa}\mathbb{E}_k[\|x^{k+1} - x^{k-\tau_k}\|^2]$$
$$\quad + \frac{4L_f}{\gamma_k - 2\lambda - \kappa}\mathbb{E}_k[\|x^{k+1} - x^{k-\tau_k}\|] - \frac{\gamma_k - \rho}{\gamma_k - 2\lambda - \kappa}\mathbb{E}_k[\|x^{k+1} - x^k\|^2]. \tag{15}$$

Finally, we measure the reduction in potential function $\psi_{1/\rho}(x^k)$ and successively deduce that

$$
\begin{aligned}
&\mathbb{E}_k[\psi_{1/\rho}(x^{k+1})] \\
&= \mathbb{E}_k[f(\hat{x}^{k+1}) + \omega(\hat{x}^{k+1}) + \frac{\rho}{2}\|\hat{x}^{k+1} - x^{k+1}\|^2] \\
&\leq \mathbb{E}_k[f(\hat{x}^k) + \omega(\hat{x}^k) + \frac{\rho}{2}\|\hat{x}^k - x^{k+1}\|^2] \\
&\leq \mathbb{E}_k[f(\hat{x}^k) + \omega(\hat{x}^k) + \frac{\rho}{2}\|\hat{x}^k - x^k\|^2] - \frac{\rho(\rho - 2\lambda - \kappa)}{2(\gamma_k - 2\lambda - \kappa)}\|\hat{x}^k - x^k\|^2 \\
&\quad + \frac{\rho\lambda}{\gamma_k - 2\lambda - \kappa}\mathbb{E}_k[\|x^{k+1} - x^{k-\tau_k}\|^2] + \frac{2\rho L_f}{\gamma_k - 2\lambda - \kappa}\mathbb{E}_k[\|x^{k+1} - x^{k-\tau_k}\|] \qquad (16) \\
&\quad - \frac{\rho(\gamma_k - \rho)}{2(\gamma_k - 2\lambda - \kappa)}\mathbb{E}_k[\|x^{k+1} - x^k\|^2] \\
&= \psi_{1/\rho}(x^k) - \frac{\rho(\rho - 2\lambda - \kappa)}{2(\gamma_k - 2\lambda - \kappa)}\|\hat{x}^k - x^k\|^2 + \frac{\rho\lambda}{\gamma_k - 2\lambda - \kappa}\mathbb{E}_k[\|x^{k+1} - x^{k-\tau_k}\|^2] \\
&\quad + \frac{2\rho L_f}{\gamma_k - 2\lambda - \kappa}\mathbb{E}_k[\|x^{k+1} - x^{k-\tau_k}\|] - \frac{\rho(\gamma_k - \rho)}{2(\gamma_k - 2\lambda - \kappa)}\mathbb{E}_k[\|x^{k+1} - x^k\|^2].
\end{aligned}
$$

where relation (16) plugs (15) in. A simple re-arrangement completes the proof.

## C.2 Proof of Theorem 1

By the descent property revealed in **Lemma 1**, we multiply both sides of the inequality by $\gamma_k - 2\lambda - \kappa$, telescope over $k = 1, \ldots, K$ and deduce that

$$
\begin{aligned}
&\frac{\rho(\rho - 2\lambda - \kappa)}{2}\mathbb{E}[\|\hat{x}^{k^*} - x^{k^*}\|^2] \\
&= \frac{\rho(\rho - 2\lambda - \kappa)}{2K}\sum_{k=1}^{K}\mathbb{E}[\|\hat{x}^k - x^k\|^2] \\
&\leq \frac{\gamma - 2\lambda - \kappa}{K}\{\psi_{1/\rho}(x^1) - \mathbb{E}[\psi_{1/\rho}(x^{K+1})]\} - \frac{\rho(\gamma - \rho)}{2K}\sum_{k=1}^{K}\mathbb{E}[\|x^{k+1} - x^k\|^2] \\
&\quad + \frac{\rho}{K}\sum_{k=1}^{K}\mathbb{E}[\lambda\|x^{k+1} - x^{k-\tau_k}\|^2 + 2L_f\|x^{k+1} - x^{k-\tau_k}\|] \\
&\leq \frac{(\gamma - 2\lambda - \kappa)D}{K} - \frac{\rho(\gamma - \rho)}{2K}\sum_{k=1}^{K}\mathbb{E}[\|x^{k+1} - x^k\|^2] \\
&\quad + \frac{\rho}{K}\sum_{k=1}^{K}\mathbb{E}[-\frac{\gamma - \rho}{2}\|x^{k+1} - x^k\|^2 + \lambda\|x^{k+1} - x^{k-\tau_k}\|^2 + 2L_f\|x^{k+1} - x^{k-\tau_k}\|], \qquad (17)
\end{aligned}
$$

where (17) is due to $\psi(x^*) \leq \mathbb{E}_k[\psi_{1/\rho}(x^{K+1})]$. Now it remains to bound the error from the stochastic delays. Let's first consider applying $\|a + b\|^2 \leq 2\|a\|^2 + 2\|b\|^2$ to get

$$
\sum_{k=1}^{K}\mathbb{E}[\|x^{k+1} - x^{k-\tau_k}\|^2] \leq 2\sum_{k=1}^{K}\mathbb{E}[\|x^k - x^{k-\tau_k}\|^2] + 2\sum_{k=1}^{K}\mathbb{E}[\|x^{k+1} - x^k\|^2] \qquad (18)
$$

and we can bound the first term using

$$\sum_{k=1}^{K} \mathbb{E}[\|x^k - x^{k-\tau_k}\|^2] = \sum_{k=1}^{K} \mathbb{E}\left[\left\|\sum_{l=1}^{\tau_k} x^{k+1-l} - x^{k-l}\right\|^2\right]$$

$$\leq \sum_{k=1}^{K} \tau_k \sum_{l=1}^{\tau_k} \mathbb{E}[\|x^{k+1-l} - x^{k-l}\|^2] \tag{19}$$

$$\leq \frac{4(L_f + L_\omega)^2}{\gamma^2} \sum_{k=1}^{K} \tau_k^2, \tag{20}$$

where (19) uses the fact $\|\sum_{i=1}^{k} a_i\|^2 \leq k \sum_{i=1}^{k} \|a_i\|^2$, (20) invokes **Lemma 7** since $\langle g, x \rangle + \omega$ is $(\|g\| + L_\omega)$ Lipschitz continuous with $\mathbb{E}_\xi[\|g\|^2] \leq L_f^2$, and we let $x^{k-j} = x^1$ if $k \geq j$. On the other hand, we bound the first-order terms by

$$\sum_{k=1}^{K} \mathbb{E}[\|x^{k+1} - x^{k-\tau_k}\|] \leq \sum_{k=1}^{K} \sum_{l=0}^{\tau_k} \mathbb{E}[\|x^{k-l+1} - x^{k-l}\|]$$

$$\leq \sum_{k=1}^{K} 2\gamma^{-1}(L_f + L_\omega)(\tau_k + 1) \tag{21}$$

$$= 2\gamma^{-1}(L_f + L_\omega)\left(K + \sum_{k=1}^{K} \tau_k\right),$$

where (21) again invokes **Lemma 7**. Plugging the bound back, we deduce that

$$\frac{\rho(\rho - 2\lambda - \kappa)}{2} \mathbb{E}[\|\hat{x}^{k^*} - x^{k^*}\|^2]$$

$$\leq \frac{(\gamma - 2\lambda - \kappa)D}{K} + \frac{\rho}{K} \sum_{k=1}^{K} \mathbb{E}\left[-\frac{\gamma - \rho}{2}\|x^{k+1} - x^k\|^2 + \lambda\|x^{k+1} - x^{k-\tau_k}\|^2 + 2L_f\|x^{k+1} - x^{k-\tau_k}\|\right]$$

$$\leq \frac{(\gamma - 2\lambda - \kappa)D}{K} + \frac{4\rho L_f(L_f + L_\omega)}{\gamma K} \sum_{k=1}^{K}(\tau_k + 1) + \frac{8\rho\lambda(L_f + L_\omega)^2}{\gamma^2 K} \sum_{k=1}^{K} \tau_k^2 \tag{22}$$

$$\leq \frac{(\rho + 2\lambda)D}{K} + \frac{D}{\sqrt{K}\alpha} + \frac{4\rho L_f(L_f + L_\omega)\alpha}{\sqrt{K}}\left(\frac{1}{K}\sum_{k=1}^{K}\tau_k + 1\right) + \frac{8\rho\lambda(L_f + L_\omega)^2\alpha^2}{K}\left(\frac{1}{K}\sum_{k=1}^{K}\tau_k^2\right)$$

where (22) is by $\gamma = \sqrt{K}/\alpha + \rho + 4\lambda + \kappa$ and we cancel the first summation term in (18) with $-\sum_{k=1}^{K} \frac{\gamma - \rho}{2}\|x^{k+1} - x^k\|^2$ since $\gamma \geq \rho + 4\lambda$. The proof is complete after dividing both sides by $\rho(\rho - 2\lambda - \kappa)$ and using the fact that $\frac{1}{2}\|\hat{x}^k - x^k\|^2 = \frac{1}{2\rho^2}\|\nabla\psi_{1/\rho}(x)\|^2$.

# D  Convergence analysis of DSPL

In this section, we introduce a more general DSPL associated with Bregman divergence and establish its convergence analysis. After proving the result in our Bregman context, the Euclidean results in Section 4 follows immediately as a special case.

**Why use Bregman divergence?**  Recall that in **C1** we assume Lipschitz continuity of both $c$ and $\nabla c$, which does not hold without assuming a bounded set. The main issue here is to expect a Lipschitz smooth function to also be Lipschitz continuous. Using the Bregman context, we can relax such Lipschitz continuity to relative Lipschitzian property, so that we do not necessarily need the bounded set assumption.

## D.1  Preliminaries and analysis of Bregman DSPL

In this section, we present the convergence results of DSPL. As we mentioned in **Remark 4**, we replace $\frac{1}{2}\|\cdot\|^2$ regularization by divergence $V_d$ and summarize the update in **Algorithm 4**.

---

**Algorithm 4:** A delayed stochastic prox-linear method with Bregman proximal updates

---

**Input:** $x^1$;

**for** $k = 1, 2,...$ **do**

    Let $c(x^{k-\tau_k}, \xi^{k-\tau_k})$ and $\nabla c(x^{k-\tau_k}, \xi^{k-\tau_k})$ be computed by a worker with delay $\tau_k$;

    In the master node update

$$x^{k+1} = \arg\min_x \left\{ h\left(c(x^{k-\tau_k}, \xi^{k-\tau_k}) + \langle \nabla c(x^{k-\tau_k}, \xi^{k-\tau_k}), x - x^k \rangle\right) + \omega(x) + \gamma_k V_d(x, x_k) \right\}$$

(23)

**end**

---

Now we overload the assumptions used in Section 4.

**A1′:** (i.i.d. sample) It is possible to draw i.i.d. samples $\{\xi^k\}$ from $\Xi$.

**A2′:** (Relative Lipschitzian property) $\omega(x)$ satisfies relatively Lipschitzian property (5)

$$\omega(x) - \omega(y) \le L_\omega \sqrt{2V_d(y, x)}$$

**A3′:** (Weak convexity) $\omega(x)$ is $\kappa$-weakly convex.

**A4′:** (Strongly-convex kernel) Kernel function $d$ is 1-strongly convex.

**A5′:** (Bounded moment) The distribution of the independent stochastic delays $\{\tau_k\}$ has bounded first and second moments. i.e., $\mathbb{E}[\tau_k] \le \bar{\tau} < \infty, \mathbb{E}[\tau_k^2] \le \tau_{\sigma^2} < \infty$, for all $k$.

**C1′:** $h(x)$ is convex and $L_h$-Lipschitz continuous; $c(x, \xi)$ has $C(\xi)$-Lipschitz continuous gradient, for all $\xi \sim \Xi$. Moreover, during the algorithm the stochastic function $f_z(x, \xi), z \in \mathrm{dom}\, d$ is $L_f(\xi)$-relative Lipschitzian to $d$ for all $\xi \sim \Xi$. Moreover, we assume $\mathbb{E}_\xi[C(\xi)^2] \le C^2, \mathbb{E}_\xi[L_c(\xi)^2] \le L_c^2$.

*Remark* 6. It can be seen that we added extra assumptions on the kernel and relaxed the Lipschitz continuity of $c$ in **C1′** and $\omega$ in **A2′**. Specially if $c$ is $L_c(\xi)$-Lipschitz continuous, we have $f_z(x, \xi)$ is $L_h L_c(\xi)$-Lipschitz continuous since

$$\begin{aligned}
&f_z(x, \xi) - f_z(y, \xi) \\
&= h(c(z, \xi) + \langle \nabla c(z, \xi), x - z \rangle) - h(c(z, \xi) + \langle \nabla c(z, \xi), y - z \rangle) \\
&\le L_h |\langle \nabla c(z, \xi), x - y \rangle| \\
&\le L_h L_c(\xi) \|x - y\|
\end{aligned}$$

With the above assumptions, we can further derive a more general version of **Proposition 1**.

**Proposition 2 (Proposition 1** in the Bregman setting**).** *Let $\{f_z(\cdot, \xi)\}$ be the sequence of stochastic functions queried during the* `DSPL` *algorithm, then*

**P1′:** *(Convexity) $f_z(x, \xi)$ is convex, for all $x \in \mathrm{dom}\, d$ and $\xi \sim \Xi$.*

**P2′:** *(Two-sided approximation) $|f_z(x, \xi) - f(x, \xi)| \le \frac{L_h C(\xi)}{2} \|x - z\|^2$, for all $x \in \mathrm{dom}\, d$ and all $\xi \sim \Xi$.*

**P3′:** *(Relative Lipschitzian property) $f_z(x, \xi) - f_z(y, \xi) \le L_f(\xi)\sqrt{2V_d(y, x)}$ for all $x, y \in \mathrm{dom}\, d$ and all $\xi \sim \Xi$.*

As we did in Section 4, we take $\lambda = L_h C$ and use these constants to present the results. Our analysis adopts the conventional potential reduction framework but a more careful treatment of stochastic noise is needed to improve the convergence result. The next lemma bounds the stochastic noise of the algorithm and is key to our analysis of DSPL.

**Lemma 8** (Stability of the stochastic iteration)**.** *Assume that the assumptions **A1′** to **A4′** hold. If $f_x(\cdot, \xi)$ is convex and satisfies **Proposition 2**, then the proximal iteration*

$$x^{k+1} = \arg\min_x \{ f_{x^{k-\tau_k}}(x, \xi^{k-\tau_k}) + \omega(x) + \gamma_k V_d(x, z^k) \}$$

*satisfies*

$$\left| \mathbb{E}_k \left\{ \mathbb{E}_\xi \left[ f_{x^{k-\tau_k}}(x^{k+1}, \xi) \right] - f_{x^{k-\tau_k}}(x^{k+1}, \xi^{k-\tau_k}) \right\} \right| \leq \frac{2L_f^2}{\gamma - \kappa},$$

*for any $\gamma_k > \kappa$.*

Now we can get the lemmas that match our main results in a Bregman context.

**Lemma 9** (**Lemma 2** in the Bregman setting)**.** *Suppose* **A1$'$**,**A2$'$**,**A3$'$**,**A4$'$** *and* **C1$'$** *hold, if $\rho > 2\lambda + \kappa, \gamma_k \geq \rho$, then*

$$\frac{\rho(\rho - 2\lambda - \kappa)}{\gamma_k - 2\lambda - \kappa} V_d(\hat{x}^k, x^k) \leq \psi_{1/\rho}^d(x^k) - \mathbb{E}_k[\psi_{1/\rho}^d(x^{k+1})] - \frac{\rho(\gamma_k - \rho)}{2(\gamma_k - 2\lambda - \kappa)} \mathbb{E}_k[\|x^{k+1} - x^k\|^2]$$

$$+ \frac{3\rho\lambda}{2(\gamma_k - 2\lambda - \kappa)} \mathbb{E}_k[\|x^{k+1} - x^{k-\tau_k}\|^2] + \frac{2\rho L_f^2}{(\gamma_k - \kappa)(\gamma_k - 2\lambda - \kappa)}$$

**Theorem 6** (**Theorem 2** in the Bregman setting)**.** *Under the same conditions as **Lemma 9**, as well as* **A5$'$**, *taking $\gamma_k \equiv \gamma = \rho + 6\lambda + \kappa + \sqrt{K}/\alpha$ for some $\alpha > 0$, letting $k^*$ be uniformly chosen between 1 and $K$, then*

$$\mathbb{E}[V_d(\hat{x}^{k^*}, x^{k^*})] \leq \frac{1}{\rho(\rho - 2\lambda - \kappa)} \left[ \frac{(\rho + 4\lambda)D}{K} + \frac{D}{\sqrt{K}\alpha} + \frac{2\rho L_f^2 \alpha}{\sqrt{K}} + \frac{6\rho\lambda(L_f + L_\omega)^2 \alpha^2}{K} \Delta_2 \right],$$

*where $D = \psi_{1/\rho}^d(x^1) - \inf_x \psi(x)$.*

## Proof Proposition 2

$f_z(x, \xi)$ inherits convexity from $h$. The other properties hold by

$$\begin{aligned}
&|f(x, \xi) - f_y(x, \xi)| \\
&= |h(c(x, \xi)) - h(c(y, \xi) + \langle \nabla c(y, \xi), x - y \rangle)| \\
&\leq L_h |c(x, \xi) - c(y, \xi) - \langle \nabla c(y, \xi), x - y \rangle| \\
&\leq \frac{L_h C}{2} \|x - y\|^2
\end{aligned}$$

and **P3$'$** is by **C1$'$**.

## Proof of Lemma 8

Without loss of generality, we consider the following proximal mapping

$$\mathcal{A}(z, x, \xi) := \arg\min_w \{ f_z(w, \xi) + \omega(w) + \gamma V_d(w, x) \},$$

where $\mathcal{A}$ denotes the proximal mapping from two past iterates $z, x$ and a given sample $\xi \sim \Xi$. Then we invoke the three-point lemma to get, for any $u \in \text{dom}\, d$ that

$$\begin{aligned}
&f_z(\mathcal{A}(z, x, \xi), \xi) + \omega(\mathcal{A}(z, x, \xi)) + \gamma V_d(\mathcal{A}(z, x, \xi), x) \\
&\leq f_z(u, \xi) + \omega(u) + \gamma V_d(u, x) - (\gamma - \kappa) V_d(u, \mathcal{A}(z, x, \xi))
\end{aligned} \tag{24}$$

Similarly, given some $\xi' \sim \Xi, v \in \text{dom}\, d$, we have

$$\begin{aligned}
&f_z(\mathcal{A}(z, x, \xi'), \xi') + \omega(\mathcal{A}(z, x, \xi')) + \gamma V_d(\mathcal{A}(z, x, \xi'), x) \\
&\leq f_z(v, \xi') + \omega(v) + \gamma V_d(v, x) - (\gamma - \kappa) V_d(v, \mathcal{A}(z, x, \xi')).
\end{aligned} \tag{25}$$

Letting $u = \mathcal{A}(z, x, \xi')$ in (24) and $v = \mathcal{A}(z, x, \xi)$ in (25), we sum the two relations up and get

$$\begin{aligned}
&(\gamma - \kappa)[V_d(\mathcal{A}(z, x, \xi'), \mathcal{A}(z, x, \xi)) + V_d(\mathcal{A}(z, x, \xi), \mathcal{A}(z, x, \xi'))] \\
&\leq f_z(\mathcal{A}(z, x, \xi), \xi') - f_z(\mathcal{A}(z, x, \xi'), \xi') + f_z(\mathcal{A}(z, x, \xi'), \xi) - f_z(\mathcal{A}(z, x, \xi), \xi) \\
&\leq L_f[\sqrt{2V_d(\mathcal{A}(z, x, \xi'), \mathcal{A}(z, x, \xi))} + \sqrt{2V_d(\mathcal{A}(z, x, \xi), \mathcal{A}(z, x, \xi'))}] \tag{26} \\
&\leq 2L_f[\sqrt{V_d(\mathcal{A}(z, x, \xi'), \mathcal{A}(z, x, \xi))} + V_d(\mathcal{A}(z, x, \xi), \mathcal{A}(z, x, \xi'))] \tag{27}
\end{aligned}$$

where (26) is by Lipschitzness from **A2′**, **P3′** and (27) uses the relation $\sqrt{a} + \sqrt{b} \leq \sqrt{2}\sqrt{a + b}$. Then this implies

$$
\max\{\sqrt{V_d(\mathcal{A}(z,x,\xi'), \mathcal{A}(z,x,\xi))}, \sqrt{V_d(\mathcal{A}(z,x,\xi), \mathcal{A}(z,x,\xi'))}\}
$$
$$
\leq \sqrt{V_d(\mathcal{A}(z,x,\xi'), \mathcal{A}(z,x,\xi)) + V_d(\mathcal{A}(z,x,\xi), \mathcal{A}(z,x,\xi'))} \leq \frac{2L_f}{\gamma - \kappa}. \tag{28}
$$

Last we successively deduce that

$$
|\mathbb{E}_{\xi'}[\mathbb{E}_{\xi}[f_z(\mathcal{A}(z,x,\xi'), \xi) - f_z(\mathcal{A}(z,x,\xi'), \xi')]]|
$$
$$
= \left| \int_{\xi' \sim \Xi} \int_{\xi \sim \Xi} f_z(\mathcal{A}(z,x,\xi'), \xi) - f_z(\mathcal{A}(z,x,\xi'), \xi') d\mu_\xi d\mu_{\xi'} \right|
$$
$$
= \left| \int_{\xi' \sim \Xi} \int_{\xi \sim \Xi} f_z(\mathcal{A}(z,x,\xi'), \xi) d\mu_\xi d\mu_{\xi'} - \int_{\xi' \sim \Xi} f_z(\mathcal{A}(z,x,\xi'), \xi') d\mu_{\xi'} \right|
$$
$$
= \left| \int_{\xi' \sim \Xi} \int_{\xi \sim \Xi} f_z(\mathcal{A}(z,x,\xi'), \xi) d\mu_\xi d\mu_{\xi'} - \int_{\xi \sim \Xi} f_z(\mathcal{A}(z,x,\xi), \xi) d\mu_\xi \right|
$$
$$
= \left| \int_{\xi' \sim \Xi} \int_{\xi \sim \Xi} f_z(\mathcal{A}(z,x,\xi'), \xi) - f_z(\mathcal{A}(z,x,\xi), \xi) d\mu_\xi d\mu_{\xi'} \right|
$$
$$
\leq \int_{\xi' \sim \Xi} \int_{\xi \sim \Xi} |f_z(\mathcal{A}(z,x,\xi'), \xi) - f_z(\mathcal{A}(z,x,\xi), \xi)| d\mu_\xi d\mu_{\xi'} \tag{29}
$$
$$
\leq \int_{\xi' \sim \Xi} \int_{\xi \sim \Xi} L_f \cdot \max \left\{ \sqrt{V_d(\mathcal{A}(z,x,\xi), \mathcal{A}(z,x,\xi'))}, \right. \tag{30}
$$
$$
\left. \sqrt{V_d(\mathcal{A}(z,x,\xi'), \mathcal{A}(z,x,\xi))} \right\} d\mu_\xi d\mu_{\xi'} \tag{31}
$$
$$
\leq \int_{\xi' \sim \Xi} \int_{\xi \sim \Xi} \frac{2L_f^2}{\gamma - \kappa} d\mu_\xi d\mu_{\xi'} = \frac{2L_f^2}{\gamma - \kappa}, \tag{32}
$$

where the third equality holds since $\xi$ and $\xi'$ are from the same distribution; (29) follows from Jensen's inequality and (32) uses (28). Plugging in $z = x^{k-\tau_k}, x = z^k, \xi' = \xi^{k-\tau_k}$ and $\gamma = \gamma_k$ into (32) completes the proof.

### Proof of Lemma 9

First, we have, by the three-point lemma and optimality of $\hat{x}^k$, that

$$
f_{x^{k-\tau_k}}(x^{k+1}, \xi^{k-\tau_k}) + \omega(x^{k+1}) + \gamma_k V_d(x^{k+1}, x^k)
$$
$$
\leq f_{x^{k-\tau_k}}(\hat{x}^k, \xi^{k-\tau_k}) + \omega(\hat{x}^k) + \gamma_k V_d(\hat{x}^k, x^k) - (\gamma_k - \kappa)V_d(\hat{x}^k, x^{k+1})
$$

and that

$$
f(\hat{x}^k) + \omega(\hat{x}^k) + \rho V_d(\hat{x}^k, x^k) \leq f(x^{k+1}) + \omega(x^{k+1}) + \rho V_d(x^{k+1}, x^k)
$$

Summing the above two relations and taking expectations, we have

$$
(\gamma_k - \rho)\mathbb{E}_k[V_d(x^{k+1}, x^k)] - (\gamma_k - \rho)V_d(\hat{x}^k, x^k) + (\gamma_k - \kappa)\mathbb{E}_k[V_d(\hat{x}^k, x^{k+1})]
$$
$$
\leq \mathbb{E}_{\xi^{k-\tau_k}}[f_{x^{k-\tau_k}}(\hat{x}^k, \xi^{k-\tau_k})] - f(\hat{x}^k) + \mathbb{E}_k[f(x^{k+1})] - \mathbb{E}_k[f_{x^{k-\tau_k}}(x^{k+1}, \xi^{k-\tau_k})]
$$
$$
= \mathbb{E}_{\xi^{k-\tau_k}}[f_{x^{k-\tau_k}}(\hat{x}^k, \xi^{k-\tau_k})] - f(\hat{x}^k) + \mathbb{E}_k[f(x^{k+1})] - \mathbb{E}_k[\mathbb{E}_\xi[f_{x^{k-\tau_k}}(x^{k+1}, \xi)]]
$$
$$
+ \mathbb{E}_k[\mathbb{E}_\xi[f_{x^{k-\tau_k}}(x^{k+1}, \xi)]] - \mathbb{E}_k[f_{x^{k-\tau_k}}(x^{k+1}, \xi^{k-\tau_k})]
$$
$$
\leq \frac{\lambda}{2}\|x^{k-\tau_k} - \hat{x}^k\|^2 + \frac{\lambda}{2}\mathbb{E}_k[\|x^{k+1} - x^{k-\tau_k}\|^2] + \frac{2L_f^2}{\gamma_k - \kappa}, \tag{33}
$$

where (33) uses **P2** and **Lemma 8**. Next we lower-bound the left-hand side using the 1-strong convexity of kernel $d$

$$
\frac{\gamma_k - \rho}{2}\mathbb{E}_k[\|x^{k+1} - x^k\|^2] - (\gamma_k - \rho)V_d(\hat{x}^k, x^k) + (\gamma_k - \kappa)\mathbb{E}_k[V_d(\hat{x}^k, x^{k+1})]
$$
$$
\leq (\gamma_k - \rho)\mathbb{E}_k[V_d(x^{k+1}, x^k)] - (\gamma_k - \rho)V_d(\hat{x}^k, x^k) + (\gamma_k - \kappa)\mathbb{E}_k[V_d(\hat{x}^k, x^{k+1})]. \tag{34}
$$

Re-arranging the terms,

$$(\gamma_k - \kappa)\mathbb{E}_k[V_d(\hat{x}^k, x^{k+1})] + \frac{\gamma_k - \rho}{2}\mathbb{E}_k[\|x^{k+1} - x^k\|^2] \tag{35}$$

$$\leq (\gamma_k - \rho)V_d(\hat{x}^k, x^k) + \frac{\lambda}{2}\mathbb{E}_k[\|x^{k+1} - x^{k-\tau_k}\|^2] + \frac{\lambda}{2}\|x^{k-\tau_k} - \hat{x}^k\|^2 + \frac{2L_f^2}{\gamma_k - \kappa}$$

$$\leq (\gamma_k - \rho)V_d(\hat{x}^k, x^k) + \frac{3\lambda}{2}\mathbb{E}_k[\|x^{k+1} - x^{k-\tau_k}\|^2] + \lambda\mathbb{E}_k[\|x^{k+1} - \hat{x}^k\|^2] + \frac{2L_f^2}{\gamma_k - \kappa} \tag{36}$$

$$= (\gamma_k - \rho)V_d(\hat{x}^k, x^k) + \frac{3\lambda}{2}\mathbb{E}_k[\|x^{k+1} - x^{k-\tau_k}\|^2] + \frac{2L_f^2}{\gamma_k - \kappa} + 2\lambda\mathbb{E}_k[V_d(\hat{x}^k, x^{k+1})]$$

$$\quad + \lambda\mathbb{E}_k[\|x^{k+1} - \hat{x}^k\|^2 - 2V_d(\hat{x}^k, x^{k+1})]$$

$$\leq (\gamma_k - \rho)V_d(\hat{x}^k, x^k) + \frac{3\lambda}{2}\mathbb{E}_k[\|x^{k+1} - x^{k-\tau_k}\|^2] + \frac{2L_f^2}{\gamma_k - \kappa} + 2\lambda\mathbb{E}_k[V_d(\hat{x}^k, x^{k+1})] \tag{37}$$

where the (36) uses $\|a+b\|^2 \leq 2\|a\|^2 + 2\|b\|^2$ and (37) again follows by $V_d(\hat{x}^k, x^{k+1}) \geq \frac{1}{2}\|x^{k+1} - \hat{x}^k\|^2$. Now we re-arrange the terms and divide both sides by $\gamma_k - 2\lambda - \kappa$ to get

$$\mathbb{E}_k[V_d(\hat{x}^k, x^{k+1})]$$

$$\leq \frac{\gamma_k - \rho}{\gamma_k - 2\lambda - \kappa}V_d(\hat{x}^k, x^k) - \frac{\gamma_k - \rho}{2(\gamma_k - 2\lambda - \kappa)}\mathbb{E}_k[\|x^{k+1} - x^k\|^2]$$

$$\quad + \frac{2L_f^2}{(\gamma_k - \kappa)(\gamma_k - 2\lambda - \kappa)} + \frac{3\lambda}{2(\gamma_k - 2\lambda - \kappa)}\mathbb{E}_k[\|x^{k+1} - x^{k-\tau_k}\|^2]$$

$$= V_d(\hat{x}^k, x^k) - \frac{\rho - 2\lambda - \kappa}{\gamma_k - 2\lambda - \kappa}V_d(\hat{x}^k, x^k) - \frac{\gamma_k - \rho}{2(\gamma_k - 2\lambda - \kappa)}\mathbb{E}_k[\|x^{k+1} - x^k\|^2]$$

$$\quad + \frac{2L_f^2}{(\gamma_k - \kappa)(\gamma_k - 2\lambda - \kappa)} + \frac{3\lambda}{2(\gamma_k - 2\lambda - \kappa)}\mathbb{E}_k[\|x^{k+1} - x^{k-\tau_k}\|^2] \tag{38}$$

Now we are ready to evaluate the decrease in the potential function.

$$\mathbb{E}_k[\psi_{1/\rho}^d(x^{k+1})]$$

$$= \mathbb{E}_k[f(\hat{x}^{k+1}) + \omega(\hat{x}^{k+1}) + \rho V_d(\hat{x}^{k+1}, x^{k+1})]$$

$$\leq \mathbb{E}_k[f(\hat{x}^k) + \omega(\hat{x}^k) + \rho V_d(\hat{x}^k, x^{k+1})]$$

$$\leq f(\hat{x}^k) + \omega(\hat{x}^k) + \rho V_d(\hat{x}^k, x^k) - \frac{\rho(\rho - 2\lambda - \kappa)}{\gamma_k - 2\lambda - \kappa}V_d(\hat{x}^k, x^k) \tag{39}$$

$$\quad + \frac{3\rho\lambda}{2(\gamma_k - 2\lambda - \kappa)}\mathbb{E}_k[\|x^{k+1} - x^{k-\tau_k}\|^2] + \frac{2\rho L_f^2}{(\gamma_k - \kappa)(\gamma_k - 2\lambda - \kappa)}$$

$$\quad - \frac{\rho(\gamma_k - \rho)}{2(\gamma_k - 2\lambda - \kappa)}\mathbb{E}_k[\|x^{k+1} - x^k\|^2]$$

$$= \psi_{1/\rho}^d(x^k) - \frac{\rho(\rho - 2\lambda - \kappa)}{\gamma_k - 2\lambda - \kappa}V_d(\hat{x}^k, x^k) - \frac{\rho(\gamma_k - \rho)}{2(\gamma_k - 2\lambda - \kappa)}\mathbb{E}_k[\|x^{k+1} - x^k\|^2]$$

$$\quad + \frac{3\rho\lambda}{2(\gamma_k - 2\lambda - \kappa)}\mathbb{E}_k[\|x^{k+1} - x^{k-\tau_k}\|^2] + \frac{2\rho L_f^2}{(\gamma_k - \kappa)(\gamma_k - 2\lambda - \kappa)},$$

where (39) plugs in the relation from (38). Another re-arrangement completes the proof.

## Proof of Theorem 6

Summing the relation from **Lemma 9** from $k = 1, \ldots, K$ and multiplying both sides by $\gamma - 2\lambda - \kappa$,

$$\rho(\rho - 2\lambda - \kappa)\mathbb{E}[V_d(\hat{x}^{k^*}, x^{k^*})]$$

$$= \frac{\rho(\rho - 2\lambda - \kappa)}{K} \sum_{k=1}^{K} \mathbb{E}[V_d(\hat{x}^k, x^k)] \tag{40}$$

$$\leq \frac{\gamma - 2\lambda - \kappa}{K} \left\{ \psi_{1/\rho}^d(x^1) - \mathbb{E}[\psi_{1/\rho}^d(x^{K+1})] \right\} + \frac{2\rho L_f^2}{\gamma - \kappa}$$

$$+ \frac{3\rho\lambda}{2K} \sum_{k=1}^{K} \mathbb{E}[\|x^{k+1} - x^{k-\tau_k}\|^2] - \frac{\rho(\gamma - \rho)}{2K} \sum_{k=1}^{K} \mathbb{E}[\|x^{k+1} - x^k\|^2]$$

$$\leq \frac{(\gamma - 2\lambda - \kappa)D}{K} + \frac{2\rho L_f^2}{\gamma - \kappa} + \frac{\rho}{K} \sum_{k=1}^{K} \mathbb{E}\left[ -\frac{\gamma - \rho}{2}\|x^{k+1} - x^k\|^2 + \frac{3\lambda}{2}\|x^{k+1} - x^{k-\tau_k}\|^2 \right], \tag{41}$$

where (41) uses the relation $\psi_{1/\rho}^d(x^{K+1}) \geq \inf_x \psi^d(x)$. Now it remains to bound the error of delays and recall that we have

$$\sum_{k=1}^{K} \|x^{k+1} - x^{k-\tau_k}\|^2 \leq 2 \sum_{k=1}^{K} \|x^{k+1} - x^k\|^2 + 4\gamma^{-2}(L_f + L_\omega)^2 \sum_{k=1}^{K} \tau_k^2 \tag{42}$$

and plugging the bounds back, we have

$$\rho(\rho - 2\lambda - \kappa)\mathbb{E}[V_d(\hat{x}^{k^*}, x^{k^*})]$$

$$\leq \frac{(\gamma - 2\lambda - \kappa)D}{K} + \frac{2\rho L_f^2}{\gamma - \kappa} + \frac{6\rho\lambda(L_f + L_\omega)^2}{\gamma^2 K} \sum_{k=1}^{K} \tau_k^2$$

$$\leq \frac{(\rho + 4\lambda)D}{K} + \frac{D}{\sqrt{K}\alpha} + \frac{2\rho L_f^2 \alpha}{\sqrt{K}} + \frac{6\rho\lambda(L_f + L_\omega)^2 \alpha^2}{K} \Delta_2, \tag{43}$$

where in (43) we cancel the first summation from (42) since $\gamma - \rho = \sqrt{K}/\alpha + 6\lambda + \kappa \geq 6\lambda$. Finally we divide both sides by $\rho(\rho - 2\lambda - \kappa)$ to complete the proof.

### D.2    Proof of Lemma 2 and Theorem 2

First $d(x) = \frac{1}{2}\|x\|^2$ satisfies **A4'**. Since **A1**, **A2**, **A3**, **A4** and **C1** are equivalent to **A1'**, **A2'**, **A3'**, **A5'** and **C1'**, noticing that **Lemma 2** is a special case of **Lemma 9** and **Theorem 2** is a special case of **Theorem 6**, we complete the proof.

### D.3    Proof of Theorem 3

Recall that in the proof of DSPL, we actually used **A4** and the properties from **Proposition 1** that are deduced from **A1**, **A2**, **A3** and **C1**. Indeed, it is straightforward to verify that $f_z(x, \xi) = \langle \nabla f(z, \xi), x - z \rangle$ satisfies **Proposition 1** given **A1**, **A2**, **A3** and $\lambda$-smoothness of $f$.

## E    Delay-independent analysis

In this section, we further improve our analysis and show that by adopting a simple safeguarding strategy during the iterations.

### E.1    Proof of Lemma 3

We show $\sum_{k=1}^{K} \tau_k \leq mK$ by noticing the following facts: **1)**. at each iteration there are at most $m - 1$ agents accumulating 1 unit of delay; **2)**. $\tau_k$ must come from a machine's previous delay accumulation. **3)**. once the gradient is used, delay on a machine starts count from 0. Assuming that at $k = 1$ there are no previously accumulated delays on each agent, then we have $\sum_{k=1}^{K} \tau_k \leq (m - 1)K \leq mK$. We use a loose bound $mK$ to simplify notation.

## E.2 Proof of Lemma 4

We know that $T \sum_{k=1}^{K} \mathbb{I}\{\tau_k \geq T\} \leq \sum_{k=1}^{K} \tau_k \leq mK$ and recall that $T = r^{-1}mK^\beta$:

$$\sum_{k=1}^{K} \mathbb{I}\{\tau_k \leq T\} \geq K - \sum_{k=1}^{K} \mathbb{I}\{\tau_k \geq T\} \geq K - \frac{mK}{T} = K - rK^{1-\beta}.$$

Also we have $\sum_{k=1}^{K} \tau_k^2 \leq T \sum_{k=1}^{K} \tau_k \leq mKT$.

## E.3 Proof of Theorem 4

First we rewrite **Lemma 1** by associating potential reduction with $I_k := \mathbb{I}\{\tau_k \leq T\}$.

$$\mathbb{E}_k[\psi_{1/\rho}(x^{k+1})] \leq \psi_{1/\rho}(x^k) - \frac{\rho(\rho - 2\lambda - \kappa)I_k}{2(\gamma_k - 2\lambda - \kappa)}\|\hat{x}^k - x^k\|^2 - \frac{\rho(\gamma_k - \rho)I_k}{2(\gamma_k - 2\lambda - \kappa)}\mathbb{E}_k[\|x^{k+1} - x^k\|^2]$$

$$+ \frac{\rho\lambda I_k}{\gamma_k - 2\lambda - \kappa}\mathbb{E}_k[\|x^{k+1} - x^{k-\tau_k}\|^2] + \frac{2\rho L_f I_k}{\gamma_k - 2\lambda - \kappa}\mathbb{E}_k[\|x^{k+1} - x^{k-\tau_k}\|],$$

where we have $x^{k+1} = x^k$, thus $\mathbb{E}_k[\psi_{1/\rho}(x^{k+1})] = \psi_{1/\rho}(x^k)$ if $I_k = 0$.

From **Lemma 4**, we know that $\sum_{k=1}^{K} I_k \geq K - rK^{1-\beta} = K(1 - rK^{-\beta}) = \eta^{-1}K$. Then similar telescopic sum over the *un-skipped* iterations will give

$$\frac{\rho(\rho - 2\lambda - \kappa)}{2}\mathbb{E}[\|\hat{x}^{k^*} - x^{k^*}\|^2]$$

$$= \frac{\rho(\rho - 2\lambda - \kappa)}{2\sum_{k=1}^{K} I_k}\sum_{k=1}^{K}\mathbb{E}[\|\hat{x}^k - x^k\|^2 I_k]$$

$$\leq \frac{\gamma - 2\lambda - \kappa}{\sum_{k=1}^{K} I_k}\{\psi_{1/\rho}(x^1) - \mathbb{E}[\psi_{1/\rho}(x^{K+1})]\} - \frac{\rho(\gamma - \rho)}{2\sum_{k=1}^{K} I_k}\sum_{k=1}^{K}\mathbb{E}[\|x^{k+1} - x^k\|^2]I_k \tag{44}$$

$$+ \frac{\rho\lambda}{\sum_{k=1}^{K} I_k}\sum_{k=1}^{K}\mathbb{E}[\|x^{k+1} - x^{k-\tau_k}\|^2]I_k + \frac{2\rho L_f}{\sum_{k=1}^{K} I_k}\sum_{k=1}^{K}\mathbb{E}[\|x^{k+1} - x^{k-\tau_k}\|]I_k$$

$$\leq \frac{\rho D}{\sum_{k=1}^{K} I_k} + \frac{\sqrt{K}D}{\alpha\sqrt{\eta}\sum_{k=1}^{K} I_k} \tag{45}$$

$$+ \frac{\rho}{\sum_{k=1}^{K} I_k}\sum_{k=1}^{K}\mathbb{E}\Big[-\frac{\gamma - \rho}{2}\|x^{k+1} - x^k\|^2 + \lambda\|x^{k+1} - x^{k-\tau_k}\|^2 + 2L_f\|x^{k+1} - x^{k-\tau_k}\|\Big]I_k,$$

where (44) uses the fact that $\gamma_k \equiv \gamma$ is the same for all the iterations; (45) uses the relation $\gamma = \frac{\sqrt{K}}{\alpha\sqrt{\eta}} + \rho + \kappa + 4\lambda \geq \max\{\frac{\sqrt{K}}{\alpha\sqrt{\eta}}, \rho + \kappa + 4\lambda\}$. Then we bound the above terms, respectively, by

$$\frac{\rho D}{\sum_{k=1}^{K} I_k} \leq \frac{\rho D}{\eta^{-1}K} = \frac{\rho\eta D}{K} \tag{46}$$

$$\frac{\sqrt{K}D}{\alpha\sqrt{\eta}\sum_{k=1}^{K} I_k} \leq \frac{\sqrt{K}D}{\alpha\eta^{-1/2}K} = \frac{\sqrt{\eta}D}{\sqrt{K}\alpha}, \tag{47}$$

where we used the fact that $\sum_{k=1}^{K} I_k \leq K$. Meanwhile,

$$\frac{2\rho L_f}{\sum_{k=1}^{K} I_k}\sum_{k=1}^{K}\|x^{k+1} - x^{k-\tau_k}\|I_k \leq \frac{2\rho L_f}{\sum_{k=1}^{K} I_k}\sum_{k=1}^{K}\|x^{k+1} - x^{k-\tau_k}\|$$

$$\leq \frac{4\sqrt{\eta}\rho L_f(L_f + L_\omega)\alpha}{\sqrt{K}\eta^{-1}}\sum_{k=1}^{K}(\tau_k + 1)$$

$$\leq \frac{4\eta^{3/2}\rho L_f(L_f + L_\omega)m\alpha}{\sqrt{K}}, \tag{48}$$

where (48) uses a tighter bound $\sum_{k=1}^{K} \tau_k \leq (m-1)K$. Finally, we bound the delays using

$$\frac{2\rho\lambda}{\sum_{k=1}^{K} I_k} \sum_{k=1}^{K} \mathbb{E}[\|x^k - x^{k-\tau_k}\|^2] I_k \leq \frac{8\eta\rho\lambda(L_f + L_\omega)^2\alpha^2}{K\sum_{k=1}^{K} I_k} \sum_{k=1}^{K} \tau_k^2 I_k \leq \frac{8\eta^2\rho\lambda(L_f + L_\omega)^2\alpha^2}{K^2} \sum_{k=1}^{K} \tau_k^2 I_k,$$

and that $\frac{1}{K^2} \sum_{k=1}^{K} \tau_k^2 I_k \leq \frac{mKT}{K^2} = \frac{m^2}{rK^{1-\beta}}$. Putting the above bounds back, using the same technique as in (18) to cancel the error $2\lambda\|x^{k+1} - x^k\|^2$ with $-\sum_{k=1}^{K} \frac{\gamma-\rho}{2}\|x^{k+1} - x^k\|^2, \gamma - \rho \geq 4\lambda$, we can re-arrange the terms to complete the proof.

### E.4 Proof of Theorem 5

We prove **Theorem 5** by showing its Bregman version.

**Theorem 7** (Safeguarded DSPL in Bregman setting). *Under the same conditions as* **Lemma 9** *as well as* **D1**, *taking 1)* $\beta > 0, K > r^{1/\beta}$ *or 2)* $\beta = 0, r < 1$, *then letting* $\gamma_k \equiv \gamma = \frac{\sqrt{K}}{\alpha\sqrt{\eta}} + \rho + \kappa + 6\lambda, \eta = 1 + \frac{r}{K^\beta - r}$ *for some* $\alpha > 0$ *and* $k^*$ *be uniformly chosen between iterations where* $\tau_k \leq \sqrt{K}$, *then*

$$\mathbb{E}[V_d(\hat{x}^{k^*}, x^{k^*})] \leq \frac{1}{\rho(\rho - 2\lambda - \kappa)} \left[ \frac{\eta(\rho + 4\lambda)D}{K} + \frac{\sqrt{\eta}D}{\sqrt{K}\alpha} + \frac{2\sqrt{\eta}\rho L_f\alpha}{\sqrt{K}} + \frac{6\eta^2\rho\lambda(L_f + L_\omega)^2 m^2\alpha^2}{K^{1/\beta}} \right],$$

*where* $D = \psi_{1/\rho}^d(x^1) - \inf_x \psi^d(x)$.

### Proof of Theorem 7

Similar to the proof of **Theorem 4**, we write the modified potential reduction by

$$\mathbb{E}_k[\psi_{1/\rho}^d(x^{k+1})] \leq \psi_{1/\rho}^d(x^k) - \frac{\rho(\rho - 2\lambda - \kappa)I_k}{2(\gamma_k - 2\lambda - \kappa)} V_d(\hat{x}^k, x^k) - \frac{\rho(\gamma_k - \rho)I_k}{2(\gamma_k - 2\lambda - \kappa)} \mathbb{E}_k[\|x^{k+1} - x^k\|^2]$$

$$+ \frac{3\rho\lambda I_k}{\gamma_k - 2\lambda - \kappa} \mathbb{E}_k[\|x^{k+1} - x^{k-\tau_k}\|^2] + \frac{2\rho L_f I_k}{(\gamma_k - \kappa)(\gamma_k - 2\lambda - \kappa)}$$

Then telescoping gives

$$\frac{\rho(\rho - 2\lambda - \kappa)}{2} \mathbb{E}[V_d(\hat{x}^{k^*}, x^{k^*})]$$

$$= \frac{\rho(\rho - 2\lambda - \kappa)}{2\sum_{k=1}^{K} I_k} \sum_{k=1}^{K} \mathbb{E}[V_d(\hat{x}^k, x^k)I_k]$$

$$\leq \frac{\gamma - 2\lambda - \kappa}{\sum_{k=1}^{K} I_k} \{\psi_{1/\rho}(x^1) - \mathbb{E}[\psi_{1/\rho}(x^{K+1})]\} - \frac{\rho(\gamma - \rho)}{2\sum_{k=1}^{K} I_k} \sum_{k=1}^{K} \mathbb{E}[\|x^{k+1} - x^k\|^2] I_k$$

$$+ \frac{3\rho\lambda}{\sum_{k=1}^{K} I_k} \sum_{k=1}^{K} \mathbb{E}[\|x^{k+1} - x^{k-\tau_k}\|^2] I_k + \frac{2\rho L_f}{\gamma - \kappa}$$

$$\leq \frac{(\rho + 4\lambda)D}{\sum_{k=1}^{K} I_k} + \frac{\sqrt{K}D}{\alpha\sqrt{\eta}\sum_{k=1}^{K} I_k} + \frac{2\rho L_f}{\gamma - \kappa} \tag{49}$$

$$+ \frac{\rho}{\sum_{k=1}^{K} I_k} \sum_{k=1}^{K} \mathbb{E}\left[ -\frac{\gamma - \rho}{2}\|x^{k+1} - x^k\|^2 + \frac{3\lambda}{2}\|x^{k+1} - x^{k-\tau_k}\|^2 \right] I_k$$

$$\leq \frac{\eta(\rho + 4\lambda)D}{K} + \frac{\sqrt{\eta}D}{\sqrt{K}\alpha} + \frac{2\sqrt{\eta}\rho L_f\alpha}{\sqrt{K}} + \frac{6\eta^2\rho\lambda(L_f + L_\omega)^2 m^2\alpha^2}{rK^{1-\beta}},$$

where (49) reuses (46), (47) and (48). A re-arrangement completes the proof of **Theorem 7**. Taking $d(x) = \frac{1}{2}\|x\|^2$ completes the proof of **Theorem 5**.

## F  DSPL **with momentum**

### F.1  Preliminaries of DSEPL

Momentum has been an important ingredient for a stochastic algorithm to be implemented in practice. In this section, we incorporate the extrapolation technique into the delayed prox-linear algo-

rithm. Before the master performs a proximal update, it uses two recent iterates to compute an extrapolated iterate $y^k$. Then a proximal update is done centered around $y^k$ with delayed information. We summarize the procedure in **Algorithm 5**. Throughout this section we take $\gamma_k \equiv \gamma$.

---

**Algorithm 5:** Extrapolated DSPL

---

**Input:** $x^0, x^1, \beta$;

**for** $k = 1, 2, \ldots$ **do**

Let $c(x^{k-\tau_k}, \xi^{k-\tau_k})$ and $\nabla c(x^{k-\tau_k}, \xi^{k-\tau_k})$ be computed by a worker with delay $\tau_k$;

In the master node update

$$y^k = x^k + \beta(x^k - x^{k-1}),$$

$$x^{k+1} = \arg\min_x \left\{ f_{x^{k-\tau_k}}(x, \xi^{k-\tau_k}) + \omega(x) + \frac{\gamma}{2}\|x - y^k\|^2 \right\}.$$

**end**

---

To analyze DSEPL, we extend the framework from [5] to the delayed case and the analysis is based on a virtual iterate

$$z^k := x^k + \beta\theta^{-1}(x^k - x^{k-1}),$$

where the extrapolation parameter, also known as momentum, is fixed at some constant $\beta \in [0, 1)$ and $\theta = 1 - \beta$. Using the virtual iterate, DSEPL uses a more complicated potential function.

$$\psi_{1/\rho}(z^k) + \frac{\rho\beta}{(\gamma - \kappa)\theta^2}\psi(x^k) + \frac{\rho(\gamma\beta + 2\rho\beta^2\theta^{-2})}{2(\gamma - \lambda)\theta}\|x^k - x^{k-1}\|^2 \tag{50}$$

As extrapolation increases the instability of the iterations, analysis of DSEPL furthers requires boundedness of the delays.

**E1:** (Bounded delay) Independent delays are bounded by $\tau < \infty$.

The following lemma presents a similar descent property for the potential function.

**Lemma 10.** *Suppose* **A1**, **A2**, **A3** *and* **C1** *hold. Given* $0 \le \beta < 1, \rho > 3\lambda + 2\kappa\beta + \kappa$ *and* $\gamma \ge \rho$,

$$\frac{(\rho - \kappa\theta)}{2\rho(\gamma - \kappa)\theta}\|\nabla\psi_{1/\rho}(z^k)\|^2$$

$$\le \psi_{1/\rho}(z^k) - \mathbb{E}_k[\psi_{1/\rho}(z^{k+1})] + \frac{\rho\beta}{(\gamma - \kappa)\theta^2}\{\psi(x^k) - \mathbb{E}_k[\psi(x^{k+1})]\}$$

$$+ \frac{\rho(2\rho\beta^2\theta^{-1} + \gamma\beta\theta)}{2(\gamma - \kappa)\theta^2}\{\|x^k - x^{k-1}\|^2 - \mathbb{E}_k[\|x^{k+1} - x^k\|^2]\}$$

$$+ \frac{2\rho L_f^2}{(\gamma - \kappa)^2\theta^2} - \frac{\rho(\gamma\theta^2 - 2\theta(\rho + \kappa\beta) - 2\rho\beta^2\theta^{-1})}{2(\gamma - \kappa)\theta^2}\mathbb{E}_k[\|x^{k+1} - x^k\|^2] + \frac{\rho\mathbb{E}_k[\varepsilon_k]}{(\gamma - \kappa)\theta^2},$$

*where* $\varepsilon_k = (\lambda\theta + \frac{\lambda}{2})\|x^{k+1} - x^{k-\tau_k}\|^2 + \frac{\lambda(1-\theta)}{2}\|x^k - x^{k-\tau_k}\|^2$ *characterizes the error of delay.*

We next bound the stochastic delay using **E1** and eliminate the delays.

**Lemma 11.** *Under the same conditions as* **Lemma 10** *as well as* **E1**, *given* $\gamma > \max\{\rho, 2(\rho + \kappa\beta)\theta^{-1} + 2\rho\beta^2\theta^{-3}\}$,

$$\mathbb{E}[\|\nabla\psi_{1/\rho}(z^{k^*})\|^2] \le \frac{2\rho\theta}{\rho - \kappa\theta}\left[\frac{(\gamma - \kappa + \rho\beta\theta^{-2})D}{K} + \frac{2\rho L_f^2}{(\gamma - \kappa)\theta^2} + \frac{\rho\lambda(3\tau^2 + 2\beta)}{2\theta^2 K}\sum_{k=1}^{K}\mathbb{E}[\|x^{k+1} - x^k\|^2]\right],$$

*where* $D = \max\{\psi_{1/\rho}(z^1) - \psi_{1/\rho}(z^*), \psi(x^1) - \psi(x^*)\}$.

*Remark* 7. **Lemma 11** reveals the effect of extrapolation. After simplification $\theta = 1 - \beta$ appears in the denominator of the error from delay and will enlarge such an error if $\beta \to 1$. This is intuitive since when extrapolating using two iterations generated from delayed information, the error is magnified.

By taking less aggressive steps, we bound the term $\sum_{k=1}^{K} \mathbb{E}[\|x^{k+1} - x^k\|^2]$ and arrive at the final convergence result for DSEPL.

**Theorem 8.** *Under the same conditions as* **Lemma 11**, *if we further choose* $\gamma \geq 2\theta^{-1}(\rho + \kappa\beta) + 2\rho\theta^{-3}\beta^2 + 2\lambda\theta^{-2}\beta + 3\lambda\theta^{-1}\tau^2$, *then*

$$\mathbb{E}[\|\nabla\psi_{1/\rho}(z^{k^*})\|^2] \leq \frac{2\rho\theta}{(\rho - \kappa\theta)\sqrt{K}}\Big(\frac{\lambda(3\tau^2 + 2\beta)}{\Gamma_\beta} + 1\Big)\left[\frac{(\gamma - \kappa + \rho\beta\theta^{-2})D}{\sqrt{K}} + \frac{2\rho L_f^2\sqrt{K}}{(\gamma - \kappa)\theta^2}\right],$$

*where* $\Gamma_\beta := \gamma\theta^2 - 2\theta(\rho + \kappa\beta) - 2\rho\beta^2\theta^{-1} - 3\lambda\tau^2 - 2\lambda\beta$.

*Remark* 8. If $\gamma = \mathcal{O}(\sqrt{K})$, then $\frac{(\gamma - \kappa + \rho\beta\theta^{-2})D}{K} + \frac{2\rho L_f^2}{(\gamma - \kappa)\theta^2} = \mathcal{O}(\frac{1}{\sqrt{K}})$, $\Gamma_\beta^{-1} = \mathcal{O}(\frac{1}{\sqrt{K}})$ and

$$\mathbb{E}[\|\nabla\psi_{1/\rho}(z^{k^*})\|^2] = \mathcal{O}\Big(\frac{1}{K} + \frac{1}{\sqrt{K}} + (\frac{1}{K} + \frac{1}{\sqrt{K}})\frac{\tau^2}{\sqrt{K}}\Big),$$

which implies that the delay is negligible if $\tau = o(K^{1/4})$.

*Remark* 9. Although our convergence result is based on the extrapolated sequence, we can leverage **Lemma 10** to show that $\mathbb{E}[\|x^{k^*+1} - x^{k^*}\|^2]$, and subsequently $\mathbb{E}[\|z^{k^*} - x^{k^*}\|^2]$ is $\mathcal{O}(\frac{1}{K})$. Using smoothness of the Moreau envelope, we finally have $\mathbb{E}[\|\nabla\psi_{1/\rho}(x^{k^*})\|^2]$ is $\mathcal{O}(\frac{1}{\sqrt{K}})$.

## F.2 Convergence analysis of DSEPL

In this section we present the convergence analysis for DSEPL.

### F.2.1 Auxiliary results

To show the convergence of DSEPL, we need to define an auxiliary sequence

$$z^k := x^k + \frac{\beta}{1 - \beta}(x^k - x^{k-1}).$$

Given $x^k$, define $\bar{x} = \beta x^k + (1 - \beta)x$ for $x \in \text{dom}(\omega)$ and $\theta = 1 - \beta$. The following identities hold

$$\bar{x} - x^k = \theta(x - x^k)$$
$$\bar{x} - y^k = \theta(x - z^k)$$
$$\bar{x} - x^{k+1} = \theta(x - z^{k+1}).$$

and will be used frequently in the analysis.

### Proof of Lemma 10

First by the $(\gamma - \kappa)$-strong convexity of the proximal sub-problem, we have

$$f_{x^{k-\tau_k}}(x^{k+1}, \xi^{k-\tau_k}) + \omega(x^{k+1}) + \frac{\gamma}{2}\|x^{k+1} - y^k\|^2$$

$$\leq f_{x^{k-\tau_k}}(\bar{x}, \xi^{k-\tau_k}) + \omega(\bar{x}) + \frac{\gamma}{2}\|\bar{x} - y^k\|^2 - \frac{\gamma - \kappa}{2}\|x^{k+1} - \bar{x}\|^2$$

$$= f_{x^{k-\tau_k}}(\bar{x}, \xi^{k-\tau_k}) + \omega(\bar{x}) + \frac{\gamma\theta^2}{2}\|x - z^k\|^2 - \frac{(\gamma - \kappa)\theta^2}{2}\|x - z^{k+1}\|^2. \quad (51)$$

Also, since $f_{x^{k-\tau_k}}(\cdot, \xi^{k-\tau_k}) + \omega(\cdot) + \frac{\kappa}{2}\| \cdot - x^k\|^2$ is convex, we plug $\bar{x}$ in and apply convexity to get that

$$f_{x^{k-\tau_k}}(\bar{x}, \xi^{k-\tau_k}) + \omega(\bar{x}) + \frac{\kappa}{2}\|\bar{x} - x^k\|^2$$

$$\leq (1 - \theta)\big[f_{x^{k-\tau_k}}(x^k, \xi^{k-\tau_k}) + \omega(x^k)\big] + \theta\big[f_{x^{k-\tau_k}}(x, \xi^{k-\tau_k}) + \omega(x) + \frac{\kappa}{2}\|x - x^k\|^2\big]$$

$$\leq (1 - \theta)\big[f_{x^{k-\tau_k}}(x^k, \xi^{k-\tau_k}) + \omega(x^k)\big] + \theta\big[f(x) + \omega(x) + \frac{\kappa}{2}\|x - x^k\|^2 + \frac{\lambda}{2}\|x - x^{k-\tau_k}\|^2\big],$$

where the second inequality leverages two-sided approximation since $\left| f_{x^{k-\tau_k}}(x, \xi^{k-\tau_k}) - f(x) \right| \leq \frac{\lambda}{2} \|x - x^{k-\tau_k}\|^2$. Then a simple re-arrangement gives

$$
\begin{aligned}
&f_{x^{k-\tau_k}}(\bar{x}, \xi^{k-\tau_k}) + \omega(\bar{x}) \\
&\leq (1-\theta)\left[ f_{x^{k-\tau_k}}(x^k, \xi^{k-\tau_k}) + \omega(x^k) \right] \\
&\quad + \theta\left[ f(x) + \omega(x) + \frac{\kappa}{2}\|x - x^k\|^2 + \frac{\lambda}{2}\|x - x^{k-\tau_k}\|^2 \right] - \frac{\kappa\theta^2}{2}\|x - x^k\|^2.
\end{aligned}
$$

Now combining the above inequality with (51), we get that

$$
\begin{aligned}
&f_{x^{k-\tau_k}}(x^{k+1}, \xi^{k-\tau_k}) + \omega(x^{k+1}) + \frac{\gamma}{2}\|x^{k+1} - y^k\|^2 \\
&\leq (1-\theta)\left[ f_{x^{k-\tau_k}}(x^k, \xi^{k-\tau_k}) + \omega(x^k) \right] + \theta\left[ f(x) + \omega(x) + \frac{\kappa}{2}\|x - x^k\|^2 + \frac{\lambda}{2}\|x - x^{k-\tau_k}\|^2 \right] \\
&\quad - \frac{\kappa\theta^2}{2}\|x - x^k\|^2 + \frac{\gamma\theta^2}{2}\|x - z^k\|^2 - \frac{(\gamma-\kappa)\theta^2}{2}\|x - z^{k+1}\|^2.
\end{aligned} \tag{52}
$$

From now on we let $x = \hat{z}^k$ in (52) and successively deduce that

$$
\begin{aligned}
&f_{x^{k-\tau_k}}(x^{k+1}, \xi^{k-\tau_k}) + \omega(x^{k+1}) + \frac{\gamma}{2}\|x^{k+1} - y^k\|^2 \\
&\leq (1-\theta)\left[ f_{x^{k-\tau_k}}(x^k, \xi^{k-\tau_k}) + \omega(x^k) \right] + \theta\left[ f(\hat{z}^k) + \omega(\hat{z}^k) + \frac{\kappa}{2}\|\hat{z}^k - x^k\|^2 + \frac{\lambda}{2}\|\hat{z}^k - x^{k-\tau_k}\|^2 \right] \\
&\quad - \frac{\kappa\theta^2}{2}\|\hat{z}^k - x^k\|^2 + \frac{\gamma\theta^2}{2}\|\hat{z}^k - z^k\|^2 - \frac{(\gamma-\kappa)\theta^2}{2}\|\hat{z}^k - z^{k+1}\|^2. \\
&= (1-\theta)\left[ f_{x^{k-\tau_k}}(x^k, \xi^{k-\tau_k}) + \omega(x^k) \right] + \theta[f(\hat{z}^k) + \omega(\hat{z}^k)] + \frac{\kappa\theta - \kappa\theta^2}{2}\|\hat{z}^k - x^k\|^2 + \frac{\lambda\theta}{2}\|\hat{z}^k - x^{k-\tau_k}\|^2 \\
&\quad + \frac{\gamma\theta^2}{2}\|\hat{z}^k - z^k\|^2 - \frac{(\gamma-\kappa)\theta^2}{2}\|\hat{z}^k - z^{k+1}\|^2 \\
&\leq (1-\theta)\left[ f_{x^{k-\tau_k}}(x^k, \xi^{k-\tau_k}) + \omega(x^k) \right] + \theta[f(\hat{z}^k) + \omega(\hat{z}^k)] + \frac{\lambda\theta}{2}\|\hat{z}^k - x^{k-\tau_k}\|^2 \\
&\quad + \frac{\gamma\theta^2}{2}\|\hat{z}^k - z^k\|^2 - \frac{(\gamma-\kappa)\theta^2}{2}\|\hat{z}^k - z^{k+1}\|^2 + \theta\kappa\beta[\|x^{k+1} - x^k\|^2 + \|\hat{z}^k - x^{k+1}\|^2] \tag{53} \\
&\leq (1-\theta)\left[ f_{x^{k-\tau_k}}(x^k, \xi^{k-\tau_k}) + \omega(x^k) \right] + \theta[f(\hat{z}^k) + \omega(\hat{z}^k)] + \theta(\kappa\beta + \lambda)\|\hat{z}^k - x^{k+1}\|^2 \\
&\quad + \lambda\theta\|x^{k+1} - x^{k-\tau_k}\|^2 + \frac{\gamma\theta^2}{2}\|\hat{z}^k - z^k\|^2 - \frac{(\gamma-\kappa)\theta^2}{2}\|\hat{z}^k - z^{k+1}\|^2 + \theta\kappa\beta\|x^{k+1} - x^k\|^2,
\end{aligned} \tag{54}
$$

where the second inequality (53) applies $\|\hat{z}^k - x^k\|^2 \leq 2[\|x^{k+1} - x^k\|^2 + \|\hat{z}^k - x^{k+1}\|^2]$ and the last inequality applies $\|\hat{z}^k - x^{k-\tau_k}\|^2 \leq 2\|\hat{z}^k - x^{k+1}\|^2 + 2\|x^{k+1} - x^{k-\tau_k}\|^2$.

By definition of $\hat{z}^k$, for $\rho > \lambda + \kappa$, we have $(\rho - \lambda - \kappa)$-strong convexity of $\psi(x) + \frac{\rho}{2}\|\cdot - x\|^2$ and

$$
\begin{aligned}
&f(\hat{z}^k) + \omega(\hat{z}^k) + \frac{\rho}{2}\|\hat{z}^k - z^k\|^2 \\
&\qquad \leq f(x^{k+1}) + \omega(x^{k+1}) + \frac{\rho}{2}\|x^{k+1} - z^k\|^2 - \frac{\rho - \lambda - \kappa}{2}\|x^{k+1} - \hat{z}^k\|^2.
\end{aligned} \tag{55}
$$

Multiplying (55) by $\theta$ and adding it to (54), we have

$$
\begin{aligned}
&\frac{\gamma}{2}\|x^{k+1} - y^k\|^2 \\
&\leq (1-\theta)\left[ f_{x^{k-\tau_k}}(x^k, \xi^{k-\tau_k}) + \omega(x^k) \right] + \theta f(x^{k+1}) - f_{x^{k-\tau_k}}(x^{k+1}, \xi^{k-\tau_k}) - (1-\theta)\omega(x^{k+1}) \\
&\quad + \frac{\rho\theta}{2}\|x^{k+1} - z^k\|^2 - \frac{\theta(\rho - 3\lambda - 2\kappa\beta - \kappa)}{2}\|x^{k+1} - \hat{z}^k\|^2 + \lambda\theta\|x^{k+1} - x^{k-\tau_k}\|^2 \\
&\quad + \frac{\gamma\theta^2 - \rho\theta}{2}\|\hat{z}^k - z^k\|^2 - \frac{(\gamma-\kappa)\theta^2}{2}\|\hat{z}^k - z^{k+1}\|^2 + \theta\kappa\beta\|x^{k+1} - x^k\|^2.
\end{aligned} \tag{56}
$$

Then we bound the first line of the right hand side in (56) by

$$(1-\theta)\left[f_{x^{k-\tau_k}}(x^k,\xi^{k-\tau_k})+\omega(x^k)\right]+\theta f(x^{k+1})-f_{x^{k-\tau_k}}(x^{k+1},\xi^{k-\tau_k})-(1-\theta)\omega(x^{k+1})$$

$$\leq (1-\theta)[f(x^k,\xi^{k-\tau_k})+\omega(x^k)-f(x^{k+1})-\omega(x^{k+1})]+f(x^{k+1})-\mathbb{E}_\xi\left[f_{x^{k-\tau_k}}(x^{k+1},\xi)\right]$$

$$+\mathbb{E}_\xi\left[f_{x^{k-\tau_k}}(x^{k+1},\xi)\right]-f_{x^{k-\tau_k}}(x^{k+1},\xi^{k-\tau_k})+\frac{(1-\theta)\lambda}{2}\|x^k-x^{k-\tau_k}\|^2$$

$$\leq (1-\theta)[f(x^k,\xi^{k-\tau_k})+\omega(x^k)-f(x^{k+1})-\omega(x^{k+1})]+\frac{2L_f^2}{\gamma-\kappa}$$

$$+\frac{\lambda}{2}[\|x^{k+1}-x^{k-\tau_k}\|^2+(1-\theta)\|x^k-x^{k-\tau_k}\|^2], \tag{57}$$

where the inequalities invoke two-sided approximation $\left|f_{x^{k-\tau_k}}(x^k,\xi^{k-\tau_k})-f(x^k,\xi^{k-\tau_k})\right|\leq\frac{\lambda}{2}\|x^k-x^{k-\tau_k}\|^2$ and bound $\mathbb{E}_\xi\left[f_{x^{k-\tau_k}}(x^{k+1},\xi)\right]-f_{x^{k-\tau_k}}(x^{k+1},\xi^{k-\tau_k})$ with **Lemma 8**. Now plugging (57) back into (56) gives

$$\frac{\gamma}{2}\|x^{k+1}-y^k\|^2$$

$$\leq (1-\theta)[f(x^k,\xi^{k-\tau_k})+\omega(x^k)-f(x^{k+1})-\omega(x^{k+1})]+\frac{2L_f^2}{\gamma-\kappa}$$

$$+\frac{\rho\theta}{2}\|x^{k+1}-z^k\|^2-\frac{\theta(\rho-3\lambda-2\kappa\beta-\kappa)}{2}\|x^{k+1}-\hat{z}^k\|^2$$

$$+\frac{\gamma\theta^2-\rho\theta}{2}\|\hat{z}^k-z^k\|^2-\frac{(\gamma-\kappa)\theta^2}{2}\|\hat{z}^k-z^{k+1}\|^2+\theta\kappa\beta\|x^{k+1}-x^k\|^2.$$

$$+\left(\lambda\theta+\frac{\lambda}{2}\right)\|x^{k+1}-x^{k-\tau_k}\|^2+\frac{\lambda(1-\theta)}{2}\|x^k-x^{k-\tau_k}\|^2. \tag{58}$$

Isolating the delayed error by $\varepsilon_k=\left(\lambda\theta+\frac{\lambda}{2}\right)\|x^{k+1}-x^{k-\tau_k}\|^2+\frac{\lambda(1-\theta)}{2}\|x^k-x^{k-\tau_k}\|^2$ and we have, by algebraic manipulations of the momentum terms, that

$$\|x^{k+1}-y^k\|^2=\|x^{k+1}-x^k+x^k-y^k\|^2$$

$$\geq\|x^{k+1}-x^k\|^2+\beta^2\|x^k-x^{k-1}\|^2-\beta\|x^{k+1}-x^k\|^2-\beta\|x^k-x^{k-1}\|^2$$

$$=\theta\|x^{k+1}-x^k\|^2-\beta\theta\|x^k-x^{k-1}\|^2 \tag{59}$$

and that

$$\frac{\rho\theta}{2}\|x^{k+1}-z^k\|^2\leq\rho\theta\|x^{k+1}-x^k\|^2+\rho\beta^2\theta^{-1}\|x^k-x^{k-1}\|^2. \tag{60}$$

Combining (59), (60) with (58) and taking expectation, we have

$$\frac{\gamma\theta}{2}\mathbb{E}_k[\|x^{k+1}-x^k\|^2]-\frac{\gamma\beta\theta}{2}\|x^k-x^{k-1}\|^2$$

$$\leq (1-\theta)\{\psi(x^k)-\mathbb{E}_k[\psi(x^{k+1})]\}+\frac{2L_f^2}{\gamma-\kappa}$$

$$+(\rho+\kappa\beta)\theta\mathbb{E}_k[\|x^{k+1}-x^k\|^2]+\rho\beta^2\theta^{-1}\|x^k-x^{k-1}\|^2-\frac{\theta(\rho-3\lambda-2\kappa\beta-\kappa)}{2}\mathbb{E}_k[\|x^{k+1}-\hat{z}^k\|^2]$$

$$+\frac{\gamma\theta^2-\rho\theta}{2}\|\hat{z}^k-z^k\|^2-\frac{(\gamma-\kappa)\theta^2}{2}\mathbb{E}_k[\|\hat{z}^k-z^{k+1}\|^2]+\mathbb{E}_k[\varepsilon_k].$$

After re-arrangement, we arrive at

$$\frac{(\gamma-\kappa)\theta^2}{2}\mathbb{E}_k[\|\hat{z}^k-z^{k+1}\|^2]$$

$$\leq (1-\theta)\{\psi(x^k)-\mathbb{E}_k[\psi(x^{k+1})]\}+\frac{\gamma\theta^2-\rho\theta}{2}\|\hat{z}^k-z^k\|^2$$

$$+\left(\rho\beta^2\theta^{-1}+\frac{\gamma\beta\theta}{2}\right)\{\|x^k-x^{k-1}\|^2-\mathbb{E}_k[\|x^{k+1}-x^k\|^2]\}-\frac{\theta(\rho-3\lambda-2\kappa\beta-\kappa)}{2}\mathbb{E}_k[\|x^{k+1}-\hat{z}^k\|^2]$$

$$+\frac{2L_f^2}{\gamma-\kappa}-\frac{\gamma\theta^2-2\theta(\rho+\kappa\beta)-2\rho\beta^2\theta^{-1}}{2}\mathbb{E}_k[\|x^{k+1}-x^k\|^2]+\mathbb{E}_k[\varepsilon_k]. \tag{61}$$

Dividing both sides of (61) by $\frac{(\gamma-\kappa)\theta^2}{2}$, we obtain that

$$
\begin{aligned}
&\mathbb{E}_k[\|\hat{z}^k - z^{k+1}\|^2] \\
&\leq \frac{2(1-\theta)}{(\gamma-\kappa)\theta^2}\{\psi(x^k) - \mathbb{E}_k[\psi(x^{k+1})]\} + \frac{\gamma\theta - \rho}{(\gamma-\kappa)\theta}\|\hat{z}^k - z^k\|^2 \\
&\quad + \frac{2\rho\beta^2\theta^{-1} + \gamma\beta\theta}{(\gamma-\kappa)\theta^2}\{\|x^k - x^{k-1}\|^2 - \mathbb{E}_k[\|x^{k+1} - x^k\|^2]\} - \frac{\rho - 3\lambda - 2\kappa\beta - \kappa}{(\gamma-\kappa)\theta}\mathbb{E}_k[\|x^{k+1} - \hat{z}^k\|^2] \\
&\quad + \frac{4L_f^2}{(\gamma-\kappa)^2\theta^2} - \frac{\gamma\theta^2 - 2\theta(\rho+\kappa\beta) - 2\rho\beta^2\theta^{-1}}{(\gamma-\kappa)\theta^2}\mathbb{E}_k[\|x^{k+1} - x^k\|^2] + \frac{2\mathbb{E}_k[\varepsilon_k]}{(\gamma-\kappa)\theta^2} \\
&= \|\hat{z}^k - z^k\|^2 - \frac{\rho - \kappa\theta}{(\gamma-\kappa)\theta}\|\hat{z}^k - z^k\|^2 + \frac{2\beta}{(\gamma-\kappa)\theta}\{\psi(x^k) - \mathbb{E}_k[\psi(x^{k+1})]\} \\
&\quad + \frac{2\rho\beta^2\theta^{-1} + \gamma\beta\theta}{(\gamma-\kappa)\theta^2}\{\|x^k - x^{k-1}\|^2 - \mathbb{E}_k[\|x^{k+1} - x^k\|^2]\} - \frac{\rho - 3\lambda - 2\kappa\beta - \kappa}{(\gamma-\kappa)\theta}\mathbb{E}_k[\|x^{k+1} - \hat{z}^k\|^2] \\
&\quad + \frac{4L_f^2}{(\gamma-\kappa)^2\theta^2} - \frac{\gamma\theta^2 - 2\theta(\rho+\kappa\beta) - 2\rho\beta^2\theta^{-1}}{(\gamma-\kappa)\theta^2}\mathbb{E}_k[\|x^{k+1} - x^k\|^2] + \frac{2\mathbb{E}_k[\varepsilon_k]}{(\gamma-\kappa)\theta^2}. \quad (62)
\end{aligned}
$$

Next, we consider the Moreau envelope and

$$
\begin{aligned}
&\mathbb{E}_k[\psi_{1/\rho}(z^{k+1})] \\
&= \mathbb{E}_k\left[\psi(\hat{z}^{k+1}) + \frac{\rho}{2}\|\hat{z}^{k+1} - z^{k+1}\|^2\right] \\
&\leq \mathbb{E}_k\left[f(\hat{z}^k) + \omega(\hat{z}^k) + \frac{\rho}{2}\|\hat{z}^k - z^{k+1}\|^2\right] \\
&\leq f(\hat{z}^k) + \omega(\hat{z}^k) + \frac{\rho}{2}\|\hat{z}^k - z^k\|^2 + \frac{\rho\beta}{(\gamma-\kappa)\theta^2}\{\psi(x^k) - \mathbb{E}_k[\psi(x^{k+1})]\} \\
&\quad - \frac{\rho(\rho - \kappa\theta)}{2(\gamma-\kappa)\theta}\|\hat{z}^k - z^k\|^2 + \frac{\rho(2\rho\beta^2\theta^{-1} + \gamma\beta\theta)}{2(\gamma-\kappa)\theta^2}\{\|x^k - x^{k-1}\|^2 - \mathbb{E}_k[\|x^{k+1} - x^k\|^2]\} \\
&\quad + \frac{2\rho L_f^2}{(\gamma-\kappa)^2\theta^2} - \frac{\rho(\gamma\theta^2 - 2\theta(\rho+\kappa\beta) - 2\rho\beta^2\theta^{-1})}{2(\gamma-\kappa)\theta^2}\mathbb{E}_k[\|x^{k+1} - x^k\|^2] + \frac{\rho\mathbb{E}_k[\varepsilon_k]}{(\gamma-\kappa)\theta^2} \quad (63) \\
&= \psi_{1/\rho}(z^k) + \frac{\rho\beta}{(\gamma-\kappa)\theta^2}\{\psi(x^k) - \mathbb{E}_k[\psi(x^{k+1})]\} \\
&\quad - \frac{\rho(\rho - \kappa\theta)}{2(\gamma-\kappa)\theta}\|\hat{z}^k - z^k\|^2 + \frac{\rho(2\rho\beta^2\theta^{-1} + \gamma\beta\theta)}{2(\gamma-\kappa)\theta^2}\{\|x^k - x^{k-1}\|^2 - \mathbb{E}_k[\|x^{k+1} - x^k\|^2]\} \\
&\quad + \frac{2\rho L_f^2}{(\gamma-\kappa)^2\theta^2} - \frac{\rho(\gamma\theta^2 - 2\theta(\rho+\kappa\beta) - 2\rho\beta^2\theta^{-1})}{2(\gamma-\kappa)\theta^2}\mathbb{E}_k[\|x^{k+1} - x^k\|^2] + \frac{\rho\mathbb{E}_k[\varepsilon_k]}{(\gamma-\kappa)\theta^2}, \quad (64)
\end{aligned}
$$

where (63) uses the relation from (62). Last we re-arrangement the terms in (64) to get

$$
\begin{aligned}
&\frac{\rho(\rho - \kappa\theta)}{2(\gamma-\kappa)\theta}\|\hat{z}^k - z^k\|^2 \\
&\leq \psi_{1/\rho}(z^k) - \mathbb{E}_k[\psi_{1/\rho}(z^{k+1})] + \frac{\rho\beta}{(\gamma-\kappa)\theta^2}\{\psi(x^k) - \mathbb{E}_k[\psi(x^{k+1})]\} \\
&\quad + \frac{\rho(2\rho\beta^2\theta^{-1} + \gamma\beta\theta)}{2(\gamma-\kappa)\theta^2}\{\|x^k - x^{k-1}\|^2 - \mathbb{E}_k[\|x^{k+1} - x^k\|^2]\} \\
&\quad + \frac{2\rho L_f^2}{(\gamma-\kappa)^2\theta^2} - \frac{\rho(\gamma\theta^2 - 2\theta(\rho+\kappa\beta) - 2\rho\beta^2\theta^{-1})}{2(\gamma-\kappa)\theta^2}\mathbb{E}_k[\|x^{k+1} - x^k\|^2] + \frac{\rho\mathbb{E}_k[\varepsilon_k]}{(\gamma-\kappa)\theta^2}.
\end{aligned}
$$

Recalling that $\|\nabla\psi_{1/\rho}(\hat{z}^k)\|^2 = \rho^2\|\hat{z}^k - z^k\|^2$, we complete the proof.

**Proof of Lemma 11**

By **Lemma 10** we have that

$$\frac{\rho - \kappa\theta}{2\rho(\gamma - \kappa)\theta}\|\hat{z}^k - z^k\|^2$$

$$\leq \psi_{1/\rho}(z^k) - \mathbb{E}_k[\psi_{1/\rho}(z^{k+1})] + \frac{\rho\beta}{(\gamma - \kappa)\theta^2}\{\psi(x^k) - \mathbb{E}_k[\psi(x^{k+1})]\}$$

$$+ \frac{\rho(2\rho\beta^2\theta^{-1} + \gamma\beta\theta)}{2(\gamma - \kappa)\theta^2}\{\|x^k - x^{k-1}\|^2 - \mathbb{E}_k[\|x^{k+1} - x^k\|^2]\} \tag{65}$$

$$+ \frac{2\rho L_f^2}{(\gamma - \kappa)^2\theta^2} - \frac{\rho(\gamma\theta^2 - 2\theta(\rho + \kappa\beta) - 2\rho\beta^2\theta^{-1})}{2(\gamma - \kappa)\theta^2}\mathbb{E}_k[\|x^{k+1} - x^k\|^2] + \frac{\rho\mathbb{E}_k[\varepsilon_k]}{(\gamma - \kappa)\theta^2}.$$

Summing up the inequality (65) from $k = 1$ to $K$, multiplying both sides by $\gamma + \kappa > 0$ and taking expectation, we have

$$\frac{\rho - \kappa\theta}{2\rho\theta}\mathbb{E}[\|\nabla\psi_{1/\rho}(z^{k^*})\|^2]$$

$$\leq \frac{\gamma - \kappa}{K}\{\psi_{1/\rho}(z^1) - \mathbb{E}_k[\psi_{1/\rho}(z^{K+1})]\} + \frac{\rho\beta}{\theta^2 K}\{\psi(x^k) - \mathbb{E}_k[\psi(x^{k+1})]\}$$

$$+ \frac{\rho(2\rho\beta^2\theta^{-1} + \gamma\beta\theta)}{2\theta^2 K}\|x^1 - x^0\|^2 + \frac{2\rho L_f^2 K}{(\gamma - \kappa)\theta^2} + \frac{\rho}{\theta^2 K}\sum_{k=1}^K \mathbb{E}[\varepsilon_k]$$

$$- \frac{\rho(\gamma\theta^2 - 2\theta(\rho + \kappa\beta) - 2\rho\beta^2\theta^{-1})}{2\theta^2 K}\sum_{k=1}^K \mathbb{E}[\|x^{k+1} - x^k\|^2]$$

$$\leq \frac{(\gamma - \kappa + \rho\beta\theta^{-2})D}{K} + \frac{2\rho L_f^2}{(\gamma - \kappa)\theta^2} + \frac{\rho}{\theta^2 K}\sum_{k=1}^K \mathbb{E}[\varepsilon_k]$$

$$- \frac{\rho(\gamma\theta^2 - 2\theta(\rho + \kappa\beta) - 2\rho\beta^2\theta^{-1})}{2\theta^2 K}\sum_{k=1}^K \mathbb{E}[\|x^{k+1} - x^k\|^2] \tag{66}$$

where the last inequality uses $x^0 = x^1$ and $D = \max\{\psi_{1/\rho}(z^1) - \psi_{1/\rho}(z^*), \psi(x^1) - \psi(x^*)\}$. Now we divide both sides of the inequality by $\frac{\rho - \kappa\theta}{2\rho\theta}$ to get

$$\mathbb{E}[\|\nabla\psi_{1/\rho}(z^{k^*})\|^2] \leq \frac{2\rho\theta}{\rho - \kappa\theta}\left[\frac{(\gamma - \kappa + \rho\beta\theta^{-2})D}{K} + \frac{2\rho L_f^2}{(\gamma - \kappa)\theta^2} + \frac{\rho}{\theta^2 K}\sum_{k=1}^K \mathbb{E}_k[\varepsilon_k]\right].$$

Last we bound $\sum_{k=1}^K \mathbb{E}[\varepsilon_k]$. First, we have the following relations

$$\sum_{k=1}^K \|x^{k+1} - x^{k-\tau_k}\|^2 \leq \tau^2 \sum_{k=1}^K \|x^k - x^{k-1}\|^2 \leq \tau^2 \sum_{k=1}^K \|x^{k+1} - x^k\|^2,$$

where the first inequality is by **E1** and the second inequality uses the fact that $x^0 = x^1$. Similarly

$$\sum_{k=1}^K \|x^{k+1} - x^{k-\tau_k}\|^2 \leq \sum_{k=1}^K 2[\|x^k - x^{k-\tau_k}\|^2 + \|x^{k+1} - x^k\|^2]$$

$$\leq 2(\tau^2 + 1)\sum_{k=1}^K \|x^{k+1} - x^k\|^2,$$

where the first inequality uses $\|a + b\|^2 \leq 2\|a\|^2 + 2\|b\|^2$ and the second inequality re-uses the bound of the first term. Plugging the above bounds back into $\sum_{k=1}^K \mathbb{E}[\varepsilon_k]$, we successively deduce

that

$$\sum_{k=1}^{K} \mathbb{E}[\varepsilon_k] = \sum_{k=1}^{K} \frac{\lambda(1-\theta)}{2} \|x^k - x^{k-\tau_k}\|^2 + \sum_{k=1}^{K} (\theta + 1/2)\lambda \mathbb{E}[\|x^{k+1} - x^{k-\tau_k}\|^2]$$

$$\leq \lambda[(1-\theta)(\tau^2+1) + (\theta+1/2)\tau^2]\sum_{k=1}^{K} \mathbb{E}[\|x^{k+1} - x^k\|^2]$$

$$= \lambda(3\tau^2/2 + \beta)\sum_{k=1}^{K} \mathbb{E}[\|x^{k+1} - x^k\|^2],$$

which implies

$$\mathbb{E}[\|\nabla\psi_{1/\rho}(z^{k^*})\|^2]$$
$$\leq \frac{2\rho\theta}{\rho-\kappa\theta}\left[\frac{(\gamma-\kappa+\rho\beta\theta^{-2})D}{K} + \frac{2\rho L_f^2}{(\gamma-\kappa)\theta^2} + \frac{\rho\lambda(3\tau^2+2\beta)}{2\theta^2 K}\sum_{k=1}^{K}\mathbb{E}[\|x^{k+1}-x^k\|^2]\right], \quad (67)$$

and this completes the proof.

**Proof of Theorem 8**

By **Lemma 11**, it remains to bound the quantity $\sum_{k=1}^{K}\mathbb{E}[\|x^{k+1}-x^k\|^2]$. To this end we consider the relation in (66), where we have

$$\frac{\rho(\gamma\theta^2 - 2\theta(\rho+\kappa\beta) - 2\rho\beta^2\theta^{-1} - 3\lambda\tau^2 - 2\lambda\beta)}{2\theta^2 K}\sum_{k=1}^{K}\mathbb{E}[\|x^{k+1}-x^k\|^2]$$

$$\leq \frac{(\gamma-\kappa+\rho\beta\theta^{-2})D}{K} + \frac{2\rho L_f^2}{(\gamma-\kappa)\theta^2}.$$

Since we choose $\beta, \gamma$ such that $\gamma\theta^2 - 2\theta(\rho+\kappa\beta) - 2\rho\beta^2\theta^{-1} - 3\lambda\tau^2 - 2\lambda\beta > 0$, then

$$\frac{\rho\lambda(3\tau^2+2\beta)}{2\theta^2 K}\sum_{k=1}^{K}\mathbb{E}[\|x^{k+1}-x^k\|^2] \leq \frac{\lambda(3\tau^2+2\beta)\left\{\frac{(\gamma-\kappa+\rho\beta\theta^{-2})D}{K} + \frac{2\rho L_f^2}{(\gamma-\kappa)\theta^2}\right\}}{\gamma\theta^2 - 2\theta(\rho+\kappa\beta) - 2\rho\beta^2\theta^{-1} - 3\lambda\tau^2 - 2\lambda\beta}.$$

Plugging the bound back, we have

$$\mathbb{E}[\|\nabla\psi_{1/\rho}(z^{k^*})\|^2]$$
$$\leq \frac{2\rho\theta}{\rho-\kappa\theta}\left[\frac{(\gamma-\kappa+\rho\beta\theta^{-2})D}{K} + \frac{2\rho L_f^2}{(\gamma-\kappa)\theta^2} + \frac{\lambda(3\tau^2+2\beta)\left\{\frac{(\gamma-\kappa+\rho\beta\theta^{-2})D}{K} + \frac{2\rho L_f^2}{(\gamma-\kappa)\theta^2}\right\}}{\gamma\theta^2 - 2\theta(\rho+\kappa\beta) - 2\rho\beta^2\theta^{-1} - 3\lambda\tau^2 - 2\lambda\beta}\right].$$

This completes our proof.

# G  Kernel and subproblems

In this section, we delve into the practical considerations related to Bregman proximal methods. In particular, we discuss how to construct suitable divergence kernels for DSPL so that we can accommodate non-Lipschitzness of the objective function. We also propose efficient subroutines for solving Bregman proximal subproblems, which are applicable to a wide range of loss functions typically encountered in machine learning tasks.

## G.1  Choosing the divergence kernel

The selection of the kernel function is a critical aspect of the Bregman proximal method. The choice of kernel has been widely discussed in the literature [3, 6], and we incorporate these findings to construct suitable kernels for the DSPL method. Given the assumptions we have made throughout the paper, it suffices to consider kernels $d(x) = \mathcal{P}_n^p(\|x\|) := \sum_{k=0}^{n} p_k\|x\|^k$, which are degree-$n$ polynomials in $\|x\|$. We recall the following lemma and refer the readers to [3] for its proof.

**Lemma 12.** *Given $f(x)$ such that $\frac{f(x)-f(y)}{\|x-y\|} \leq \mathcal{P}_n^p(\|x\|) + \mathcal{P}_n^p(\|y\|) = \sum_{k=0}^n p_k(\|x\|^k + \|y\|^2), p \geq 0, \forall x \in \operatorname{dom} h$, then $f$ satisfies 1-relative Lipschitzian property with kernel $d(x) = \sum_{k=0}^n \frac{3k+7}{k+2} p_k \|x\|^{k+2}$.*

The above lemma implies that once we establish a polynomial upper bound on the Lipschitzness of $f_z(x,\xi) + \omega(x)$, a kernel $d$ is immediately available.

We note that the above kernel covers most of the applications [2] of the prox-linear method, including phase retrieval, blind deconvolution, matrix completion, covariance estimation, robust PCA, and so on. In many machine learning applications, $c$ has Lipschitz continuous gradient, meaning that $\nabla c$ is bounded by first-order growth. For example, in the phase retrieval problem, we have $\omega = 0, h(x) = |x|, c(x,\xi) = \langle a, x \rangle^2 - b$. Hence, we get

$$\|\nabla c(z,\xi)\| = 2|\langle a, z\rangle| \cdot \|a\| \leq 2\|a\|^2 \|z\| \text{ and } \frac{f_z(x,\xi) - f_z(y,\xi)}{\|x-y\|} \leq 2\|a\|^2 \|z\|.$$

Now assume that $z = x$, then for any $p_0 \geq 0$ and $p_1 = 2\alpha^2, \alpha = \sup_{a\sim\Xi} \|a\|$ we know that $2\|a\|^2 \|x\| \leq p_0 + p_1 \|x\|$. Hence, we can take $d(x) = \frac{7}{2}\|x\|^2 + \frac{10\alpha^2}{3}\|x\|^3$ as our kernel. Recall that in DSPL $z = x^{k-\tau_k}, x = x^k$, as long as there exist some $M, N$ such that $\|x^j\| \leq M\|x^k\| + N, k \geq j$ for all $k$, our assumptions are satisfied.

Now that we have specified ways to construct kernels for DSPL, we discuss how to solve the Bregman proximal subproblems in the next section. The subproblems' solution determines the practical applicability of our methods, and we show that these subproblems can be efficiently solved for a wide range of problems.

### G.2  Bregman proximal subproblem

In this section, we discuss the solution of the proximal subproblems for DSPL when $\omega = 0$, where the Bregman proximal subproblem can then be expressed as

$$\min_x \quad h(\langle a, x \rangle + b) + V_d(x,y). \tag{68}$$

Consider a piece-wise linear $h(x) = \max\{\alpha_1 x + \beta_1, \alpha_2 x + \beta_2\}$. This formulation characterizes common nonsmooth loss functions including $\ell_1$ loss $|x| = \max\{x, -x\}$ and hinge loss $\max\{0, x\}$. Substituting $h$ in and removing terms irrelevant to $x$, we arrive at the following DSPL subproblem.

$$\min_x \quad \max\{\langle a_1, x \rangle + b_1, \langle a_2, x \rangle + b_2\} + d(x) \tag{69}$$

The next proposition provides a solution to the above subproblem.

**Proposition 3** (Solving the proximal subproblem). *The proximal subproblem can be solved by evaluating solutions to the following three problems*

$$\min_x \langle a_1, x \rangle + d(x),$$
$$\min_x \langle a_2, x \rangle + d(x),$$

*and*

$$\min_x \quad \langle a_1, x \rangle + d(x)$$
$$\text{subject to} \quad \langle a_1 - a_2, x \rangle + b_1 - b_2 = 0,$$

*where the solution to the third problem satisfies $x = u + \alpha v$, where $u = -\frac{(a_1-a_2)(b_1-b_2)}{\|a_1-a_2\|^2}$, $v = \frac{(\|a_1\|^2 - \|a_2\|^2)}{2\|a_1-a_2\|^2}(a_1 - a_2) - \frac{a_1+a_2}{2}$ and $\alpha$ is a positive root of the following equation*

$$\sum_{k=0}^n p_k \alpha \|u + \alpha v\|^k - 1 = 0.$$

*Proof.* Recall that we are solving the following convex optimization problem

$$\min_x \quad \max\{\langle a_1, x \rangle + b_1, \langle a_2, x \rangle + b_2\} + d(x),$$

where $d$ is strongly convex and thus the problem admits a unique minimizer $x^*$. Then we do case analysis.

**Case 1**. $\langle a_1, x^* \rangle + b_1 > \langle a_2, x^* \rangle + b_2$. In this case solving $\min_x \langle a_1, x \rangle + d(x)$ produces $x^*$.

**Case 2**. $\langle a_1, x^* \rangle + b_1 < \langle a_2, x^* \rangle + b_2$. In this case solving $\min_x \langle a_2, x \rangle + d(x)$ produces $x^*$.

The above two cases degenerate into the subproblems from stochastic mirror descent and its solution has been discussed in [6]. The last case is a bit more tricky.

**Case 3**. $\langle a_1, x^* \rangle + b_1 = \langle a_2, x^* \rangle + b_2$.

In this case, the proximal subproblem becomes an equality constrained problem.

$$\min_x \quad \langle a_1, x \rangle + d(x)$$
$$\text{subject to} \quad \langle a_1 - a_2, x \rangle + b_1 - b_2 = 0$$

Letting $\lambda$ be the multiplier of the equality constraint and appealing to the optimality condition, we have

$$a_1 + \nabla d(x) + \lambda(a_1 - a_2) = 0 \tag{70}$$
$$a_2 + \nabla d(x) + \lambda(a_1 - a_2) = 0 \tag{71}$$
$$\langle a_1 - a_2, x \rangle + b_1 - b_2 = 0. \tag{72}$$

Summing (70) (71) and dividing both sides by 2, we have

$$\tfrac{a_1+a_2}{2} + \nabla d(x) + \lambda(a_1 - a_2) = 0. \tag{73}$$

Multiplying both sides of (73) by $a_1 - a_2$, we arrive at

$$\tfrac{\|a_1\|^2 - \|a_2\|^2}{2} + \langle a_1 - a_2, \nabla d(x) \rangle + \lambda \|a_1 - a_2\|^2 = 0. \tag{74}$$

Since $d(x)$ is a polynomial in $\|x\|$, $\nabla d(x) = \zeta x, \zeta > 0$ and using (72), we have

$$\lambda = \tfrac{b_1 - b_2}{\|a_1 - a_2\|^2} \zeta - \tfrac{\|a_1\|^2 - \|a_2\|^2}{2\|a_1 - a_2\|^2}. \tag{75}$$

Then substituting $\lambda$ into (74),

$$0 = \tfrac{a_1+a_2}{2} + \nabla d(x) + \lambda(a_1 - a_2)$$
$$= \tfrac{a_1+a_2}{2} + \zeta x + \zeta \tfrac{(a_1-a_2)(b_1-b_2)}{\|a_1-a_2\|^2} - \tfrac{\|a_1\|^2 - \|a_2\|^2}{2\|a_1-a_2\|^2}(a_1 - a_2)$$
$$= \zeta \left[ x + \tfrac{(a_1-a_2)(b_1-b_2)}{\|a_1-a_2\|^2} \right] - \tfrac{\|a_1\|^2 - \|a_2\|^2}{2\|a_1-a_2\|^2}(a_1 - a_2) + \tfrac{a_1+a_2}{2}$$

we get

$$x = \zeta^{-1} \left[ \tfrac{(\|a_1\|^2 - \|a_2\|^2)}{2\|a_1-a_2\|^2}(a_1 - a_2) - \tfrac{a_1+a_2}{2} \right] - \tfrac{(a_1-a_2)(b_1-b_2)}{\|a_1-a_2\|^2}$$

for some $\zeta > 0$. Without loss of generality, we express $x = u + \alpha v$ for $u = -\tfrac{(a_1-a_2)(b_1-b_2)}{\|a_1-a_2\|^2}$ and $v = \tfrac{(\|a_1\|^2 - \|a_2\|^2)}{2\|a_1-a_2\|^2}(a_1 - a_2) - \tfrac{a_1+a_2}{2}$ and $\alpha > 0$. Substituting back gives $\zeta = \sum_{k=0}^n p_k \|x\|^k$ and

$$\|\zeta(x - u) - v\| = \|\alpha \zeta v - v\| = |\alpha\zeta - 1| \cdot \|v\| = 0.$$

Therefore, we only need to verify the positive roots of $\sum_{k=0}^n p_k \alpha \|u + \alpha v\|^k - 1 = 0$, which can be done efficiently if $n$ is not large. This completes the proof. $\qquad \square$

### G.3 Euclidean subproblem with nonsmooth regularizer

In this section, we discuss the solution of the proximal subproblems for DSPL when $d(x) = \frac{1}{2}\|x\|^2$ and $\omega$ takes on one of two forms: **1)**. Indicator function of ball $\delta_{\{\|x\| \le R\}}$. **2)**. $\ell_1$ regularizer $\|x\|_1$. We still assume $h(x) = \max\{\alpha_1 x + \beta_1, \alpha_2 x + \beta_2\}$, which covers all the popular applications of SPL from [2, 4]. Following the same argument as in the previous section, each DSPL iteration is reduced to

$$\min_x \quad \max\{\langle a_1, x \rangle + b_1, \langle a_2, x \rangle + b_2\} + \omega(x) + \tfrac{1}{2}\|x - y\|^2$$

and after doing case analysis on $\langle a_1, x^* \rangle + b_1$ and $\langle a_2, x^* \rangle + b_2$ as in **Proposition 3**, it suffices to solve

$$\min_x \quad \omega(x) + \tfrac{1}{2}\|x - y\|^2$$
$$\text{subject to} \quad \langle a, x \rangle - b = 0$$

efficiently.

**Case 1.** $\omega(x) = \delta_{\{\|x\| \leq R\}}$. In this case, we need to solve

$$\min_x \quad \tfrac{1}{2}\|x - y\|^2$$
$$\text{subject to} \quad \langle a, x \rangle = b$$
$$\|x\|^2 \leq r = R^2,$$

whose KKT condition is given by

$$\langle a, x \rangle = b$$
$$\lambda \geq 0$$
$$\lambda(\|x\|^2 - r) = 0$$
$$x - y + \nu a + 2\lambda x = 0$$

After simplification, we arrive at

$$x = \frac{y}{1 + 2\lambda} - \frac{\langle a, y \rangle a - (1 + 2\lambda) b a}{(1 + 2\lambda)\|a\|^2}.$$

Then either $\lambda = 0$ or the solution can be obtained using bisection.

**Case 2.** $\omega(x) = \|x\|_1$. In this case, we need to solve

$$\min_x \quad \|x\|_1 + \tfrac{1}{2}\|x - y\|^2$$
$$\text{subject to} \quad \langle a, x \rangle - b = 0$$

Given the Lagrangian multiplier $\nu$ associated with the equality constraint, we have that $x(\nu) = \mathcal{S}(y - \nu a)$, which is given by the soft-thresholding operator. The residual of the equality constraint is given by:

$$f(\nu) = \langle a, x \rangle - b = \langle a, x(\nu) \rangle - b.$$

This function is univariate in $\nu$ and is semi-smooth. Therefore, semi-smooth Newton method can efficiently find its root. This makes the computation tractable.

# H Additional experiments

In this section, we present additional experiments incorporating momentum. Our experiments focus on both robust phase retrieval and blind deconvolution problems.

## H.1 Robust phase retrieval

We refer readers to Section 6 for the detailed formulation of phase retrieval. Most of the experiment settings in this section are consistent with Section 6.

### H.1.1 Experiment setup

**1) Initial point and radius**. For the synthetic data, we generate $x' \sim \mathcal{N}(0, I_n)$ and start from $x^0 = x^1 = \frac{x'}{\|x'\|}$ and for zipcode data, we generate $x' \sim \mathcal{N}(\hat{x}, I_n)$ and take $x^0 = x^1 = x'$. $M = 1000\|x^0\|$.

2) **Stepsize**. We tune the stepsize parameter setting $\gamma = \sqrt{K/\alpha}$, where $\alpha \in \{0.1, 0.5, 1.0\}$ in the asynchronous environment, $\alpha \in [10^{-2}, 10^1]$ for synthetic data in the simulated environment and $\alpha \in [10^0, 10^1]$ for the `zipcode` dataset.

3) **Momentum parameter**. In the asynchronous environment, we allow $\beta \in \{0, 0.1, 0.3, 0.6\}$ and in the simulated environment, we test $\beta \in \{0.6, 0.9\}$ for the synthetic and `zipcode` data respectively.

4) **Others**. Other settings are consistent with Section 6.

### H.1.2 Asynchronous environment

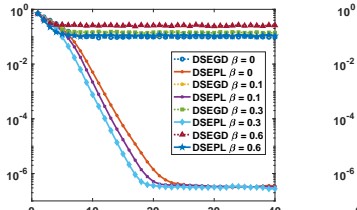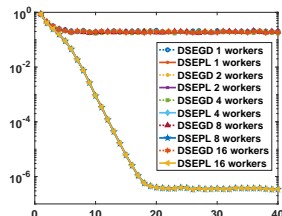

Figure 6: First: speedup in time and the number of workers. Second: progress of $\|x^k - \hat{x}\|$ in the first 40 epochs given 16 workers and $\alpha = 0.5$. Third: progress of $f(x^k) - f(\hat{x})$ in the first 40 epochs given $\beta = 0.3$ and different number of workers.

The first figure plots the effect of different momentum parameters in the distributed environment. It can be seen that both `DSEGD` and `DSEPL` perform better with momentum. Moreover, the performance of `DSEPL` dominate `DSEGD` as we observed in the previous experiments.

### H.1.3 Simulated Environment

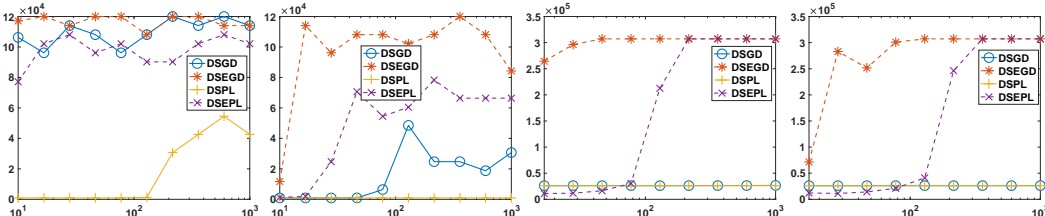

Figure 7: Left to right: $(\kappa, p_{\text{fail}}) = (10, 0.3), \alpha = 5.0$, Geometric and Poisson delays; `zipcode` data of $p_{\text{fail}} = 0.2, \alpha = 6.0$, Geometric and Poisson delays. x-axis represents $\tau_{\max}$ and y-axis shows average iteration number to reach the stopping criterion over 20 tests.

Figure 7 plots the impact of staleness on the number of iterations each algorithm takes to reach the desired accuracy. It can be seen that with other parameters fixed, `DSPL` tends to be more robust against delays than the pure subgradient-based methods, which is consistent with the theoretical results. Moreover, we observe that when extrapolation is used, the algorithm converges faster at the cost of less robustness as delay increases.

Our last experiment investigates the robustness of `DSPL` compared to `DSGD` and justifies the use of extrapolation in presence of delay. Figure 8 plots the number of iterations for each algorithm to converge with different datasets, extrapolation parameters, and delays. In spite of delays, `DSPL` and `DSEPL` still admit a wider range of stepsize parameters ensuring convergence than `DSGD`. Also, when the stepsize is not large, the use of extrapolation can effectively accelerate the convergence.

### H.2 Blind deconvolution problem

In this section, we present the additional experiments on blind deconvolution problem to further illustrate the efficiency of `DSPL`. The blind deconvolution problem, unlike robust phase retrieval,

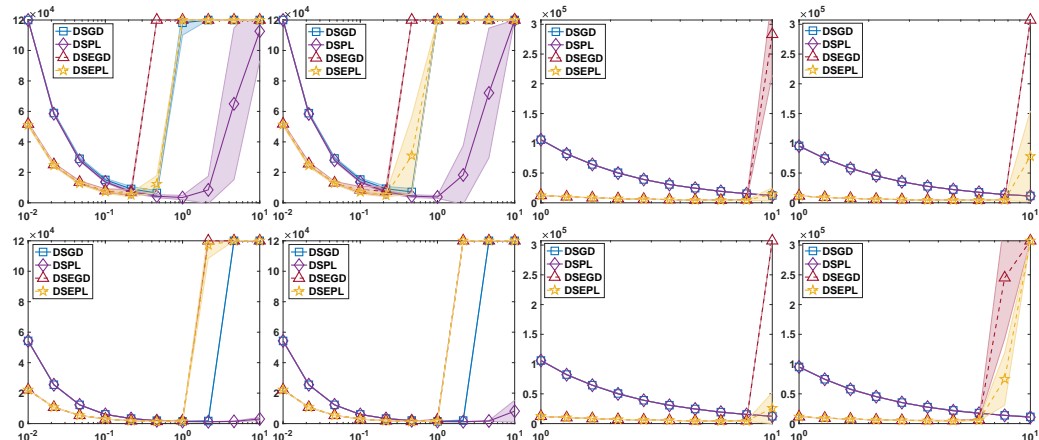

Figure 8: First row(Geometric) left: $(\kappa, p_{\text{fail}}) = (1, 0.3), \beta = 0.6, \tau \in \{28, 47\}$, right: `zipcode` $p_{\text{fail}} = 0.3, \beta = 0.9, \tau \in \{28, 600\}$. Second row(Poisson) left: $(\kappa, p_{\text{fail}}) = (10, 0.2), \beta = 0.6, \tau \in \{28, 47\}$. right: `zipcode` $p_{\text{fail}} = 0.3, \beta = 0.9, \tau \in \{28, 600\}$.

aims to recover two signals from their convolution and its mathematical formulation is given by

$$\min_{x,y \in \mathbb{R}^n} \frac{1}{m} \sum_{i=1}^{m} |\langle u_i, x \rangle \langle v_i, y \rangle - b_i| + \mathbb{I}\{\|x\|, \|y\| \leq \Delta\},$$

where $\{b_i\}$ are the measurements, $\{(u_i, v_i)\}$ are the measuring data and $x, y$ are the optimization variables corresponding to the signals. Similar to phase retrieval, we first present the detailed setup and then inspect the performance of DSPL in both real and simulated asynchronous environments.

### H.2.1 Experiment setup

**Data generation.** We use synthetic data for blind deconvolution problems.

**Synthetic data.** We take $m = 300, n = 100$ in the experiments of simulated delay and $m = 1500, n = 150$ in the asynchronous environment. The data is generated similar to phase retrieval, where, given some conditioning parameter $\kappa \geq 1$, we compute $U = Q_1 D_2, V = Q_2 D_2, Q \in \mathbb{R}^{m \times n}, q_{ij} \sim \mathcal{N}(0, 1)$ and $D = \text{diag}(d), d \in \mathbb{R}^n, d_i \in [1/\kappa, 1], \forall i$. Then two true signals are generated like $\hat{x}$ in phase retrieval and we random corruption is applied.

1) **Dataset.** In the asynchronous environment, we set $\kappa = 1, p_{\text{fail}} = 0$ and in the simulated environment, we set $\kappa = 1$ and $p_{\text{fail}} \in \{0.2, 0.3\}$.

2) **Initial point and radius.** We generate $x', y' \sim \mathcal{N}(0, I_n)$ and start from $x^0 = x^1 = \frac{x'}{\|x'\|}, y^0 = y^1 = \frac{y'}{\|y'\|}$. $\Delta = 1000\|(x^0, y^0)\|$.

3) **Stepsize.** We tune the stepsize parameter setting $\gamma = \sqrt{K/\alpha}$, where $\alpha \in \{0.1, 0.5, 1.0\}$ in the asynchronous environment, $\alpha \in [10^{-2}, 10^1]$ for synthetic data in the simulated environment.

The rest of the experiment setup are consistent with in phase retrieval.

### H.2.2 Asynchronous environment

The experiments for blind deconvolution again justifies the effect of DSPL in its convergence behavior and robustness to the stochastic delays. By retaining the smooth structure of the inner composite function, DSPL gives a more accurate approximation of the original objective function only at cost of transmitting one more scaler (i.e., the inner objective value). Hence DSPL serves as a competitive alternative to DSGD when the problem enjoys composite structure (2) in the distributed setting.

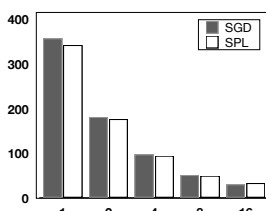 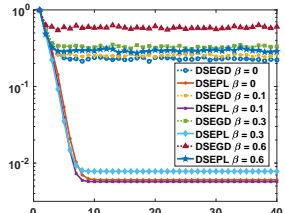 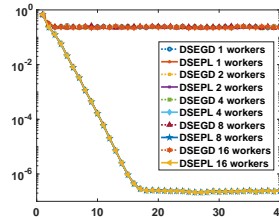

Figure 9: First: speedup in time and the number of workers. Second: progress of $\|(x^k, y^k) - (\hat{x}, \hat{y})\|$ in the first 40 epochs given 16 workers and $\alpha = 0.5$. Third: progress of $f(x^k, y^k) - f(\hat{x}, \hat{y})$ in the first 40 epochs given $\beta = 0.1$ and different number of workers.

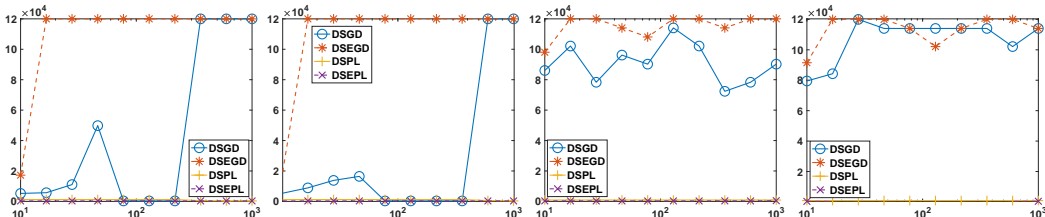

Figure 10: From left to right: $(\kappa, p_{\text{fail}}) = (1, 0.2), \alpha = 4.6$, Geometric and Poisson delays; $(\kappa, p_{\text{fail}}) = (1, 0.3), \alpha = 2.1$, Geometric and Poisson delays. The x-axis represents delay $\tau$ and the y-axis gives the number of iterations to reach the stopping criterion.

### H.2.3 Simulated environment

In the simulated experiments for blind deconvolution, we can see similar robustness of DSPL against delay and stepsize selection, which confirms our previous observations.

### H.3 Proximal sub-problems

This subsection shows how the sub-problems from DSPL and DSPL are solved for completeness. The results are a direct adaptation from [1] and interested readers can check [1] for the detailed derivations.

**Phase retrieval problem** In phase retrieval problem, we denote $x$ to be the point where the stochastic model function is constructed and $y$ to be the center of the proximal term. Then the DSPL proximal subproblem is given by

$$\min_x \ \left|\langle a, z\rangle^2 + 2\langle a, z\rangle\langle a, x - z\rangle - b\right| + \frac{\gamma}{2}\|x - y\|^2$$

and it admits closed-form solution when away from the boundary

$$x^+ = y + \text{Proj}_{[-1,1]}(-\frac{\delta}{\|\zeta\|^2})\zeta,$$

where $\delta = \gamma^{-1}(\langle a, z\rangle^2 + 2\langle a, z\rangle\langle a, x - z\rangle - b)$ and $\zeta = 2\gamma^{-1}\langle a, z\rangle a$ and $\text{Proj}_{[-1,1]}(\cdot)$ denotes projection onto the box $[-1, 1]$.

**Blind deconvolution problem** In the blind deconvolution problem, we use $z = (z_x, z_y)$ to denote the point of model construction and $w = (w_x, w_y)$ to denote the proximal center. Then the subproblem is given by

$$\min_{\|x\|, \|y\| \leq \Delta} \ \left|\langle u, z_x\rangle\langle v, z_y\rangle + \langle v, z_y\rangle\langle u, x - z_x\rangle + \langle u, z_x\rangle\langle v, y - z_y\rangle - b\right| + \frac{\gamma}{2}[\|x - w_x\|^2 + \|y - w_y\|^2].$$

First we assume that the constraints are inactive, then the solution is available in closed form

$$w^+ = w + \text{Proj}_{[-1,1]}(-\frac{\delta}{\|\zeta\|^2})\zeta,$$

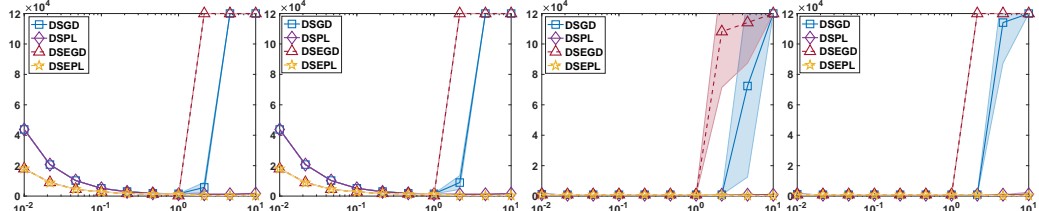

Figure 11: From left to right: $(\kappa, p_{\text{fail}}) = (1, 0.2)$, Geometric and Poisson delays with $\tau = 10$; $(\kappa, p_{\text{fail}}) = (1, 0.3)$, Geometric and Poisson delays with $\tau = 216$.

where

$$\delta = \gamma^{-1} \left[ \langle u, z_x \rangle \langle v, z_y \rangle + \langle v, z_y \rangle \langle u, w_x - z_x \rangle + \langle u, z_x \rangle \langle v, w_y - z_y \rangle - b \right]$$

$$\zeta = \gamma^{-1} (\langle v, z_y \rangle u, \langle u, z_x \rangle v).$$

## H.4 Separate figures of Section 6

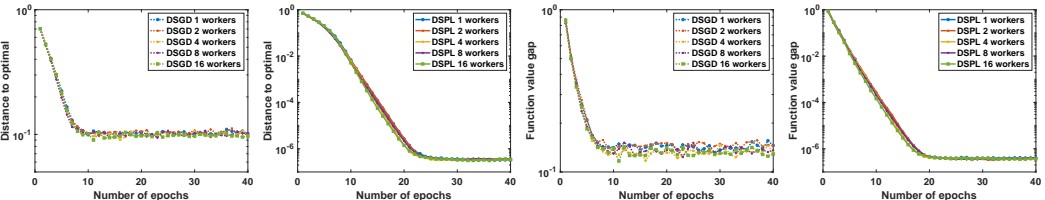

Figure 12: Figure 2 after splitting DSGD and DSPL. Two plots on the left: $\|x^k - \hat{x}\|$ in the first 40 epochs given $\alpha = 0.1$; two on the right: $f(x^k) - f(\hat{x})$ in the first 40 epochs given $\alpha = 0.5$.

## References in the appendix

[1] Damek Davis and Dmitriy Drusvyatskiy. Stochastic model-based minimization of weakly convex functions. *SIAM Journal on Optimization*, 29(1):207–239, 2019.

[2] Damek Davis and Dmitriy Drusvyatskiy. Graphical convergence of subgradients in nonconvex optimization and learning. *Mathematics of Operations Research*, 47(1):209–231, 2022.

[3] Damek Davis, Dmitriy Drusvyatskiy, and Kellie J MacPhee. Stochastic model-based minimization under high-order growth. *arXiv preprint arXiv:1807.00255*, 2018.

[4] Damek Davis and Benjamin Grimmer. Proximally guided stochastic subgradient method for nonsmooth, nonconvex problems. *SIAM Journal on Optimization*, 29(3):1908–1930, 2019.

[5] Qi Deng and Wenzhi Gao. Minibatch and momentum model-based methods for stochastic weakly convex optimization. *Advances in Neural Information Processing Systems*, 34, 2021.

[6] Haihao Lu. "relative continuity" for non-lipschitz nonsmooth convex optimization using stochastic (or deterministic) mirror descent. *INFORMS Journal on Optimization*, 1(4):288–303, 2019.

[7] R Tyrrell Rockafellar. Convex analysis princeton university press. *Princeton, NJ*, 1970.

[8] Siqi Zhang and Niao He. On the convergence rate of stochastic mirror descent for nonsmooth nonconvex optimization. *arXiv preprint arXiv:1806.04781*, 2018.

