# OpenReview forum: "Delayed Algorithms for Distributed Stochastic Weakly Convex Optimization"
_NeurIPS.cc/2023/Conference — NeurIPS 2023 poster_

### Official Review · Reviewer_93YJ · 2023-06-21

**Soundness:** 2 fair
**Presentation:** 3 good
**Contribution:** 2 fair
**Rating:** 4
**Confidence:** 3

**Summary:**

This paper investigates delayed stochastic optimization problems with composite objective functions. The authors demonstrate that, under weakly convex conditions, the convergence rate of the delayed stochastic subgradient method can be improved from $\mathcal{O}(\tau_{\max}/\sqrt{K})$ to $\mathcal{O}(\bar{\tau}/\sqrt{K})$. Additionally, when the stochastic part $f$ in the objective function follows the structure $f = h\circ c$, the authors propose a delayed stochastic prox-linear method to address the impact of delayed information. Lastly, by utilizing a hard-thresholding technique, the authors effectively manage the magnitude of delay, mitigating the issue of excessive delay compromising the robustness of the original algorithm.

**Strengths:**

1. The paper is written in a clear and easily understandable manner. The language and organization make it accessible to readers.

2. The majority of the proofs presented in the paper are accurate and sound. This demonstrates the authors' proficiency in the mathematical aspects of their work.

3. The authors provide valuable numerical experiments that effectively showcase the strengths and weaknesses of the DSGD and DSPL methods, specifically in terms of their step size selection. This empirical evidence enhances the practical relevance of the research.

**Weaknesses:**

Weaknesses:

1. The research direction pursued in the paper appears to be relatively narrow, as the authors focus solely on improving the convergence rate under a specific setting. Moreover, the paper lacks substantial evidence to demonstrate the significance of these improvements. It would benefit from a more comprehensive exploration of the broader implications and potential applications of the proposed methods.

2. The paper lacks in-depth explanations of the theoretical results. For instance, the authors fail to clarify why DSGD improves the maximum delay $\tau_{\max}$ to $\bar{\tau}$ compared to the reference [1]. Similarly, there is a lack of explanation regarding why DSPL can eliminate the impact of delay in objective functions that do not possess the $h\circ c$ structure. And if there is a minimax lower bound for the latter? It would be helpful to provide further insights and discussions to enhance the readers' understanding.

3. The assumptions made about the objective function predominantly revolve around the uniformity of the randomness $\xi$. For example, requiring $f(x,\xi)$ to be weakly convex for any $\xi$, compared to the less restrictive assumption of $ E_{\xi} f(x,\xi) $
being weakly convex in the reference [1]. This raises concerns about whether the improvements in convergence rate are solely attributed to the introduction of stronger assumptions. Additionally, the assumption $\sup_{g\in \partial f(x,\xi)}\|g\| \le L_f, \forall \xi$ may not hold for the robust phase retrieving problem considered in the authors' experiments, where the measuring matrix is sampled by a Gaussian ensemble. These limitations should be acknowledged and discussed to provide a more comprehensive analysis.

[1] Distributed stochastic inertial-accelerated methods with delayed derivatives for nonconvex problems.

**Questions:**

Most of the questions are raised in the Weaknesses section. Another question worth considering is whether an experiment can be conducted to demonstrate that the convergence of DSPL is unaffected by the magnitude of delay.

**Limitations:**

The main limitations of the paper center around its limited scope, incremental contributions, and insufficient exploration of key theoretical aspects, undermining its broader impact and significance in the field.

---

> ### Author Rebuttal · Authors · 2023-08-09
>
> **Response to Reviewer 93YJ**
>
> We thank the reviewer for the time spent on our paper and the constructive suggestions.
>
> **Weaknesses**
>
> 1. The research direction is narrow, only focusing on a specific setting
>
>    We politely disagree with the reviewer on "narrow, only focusing on a specific setting".
>
>    First of all, we believe that our target setting, weakly convex problem, is a quite a general class of problems covering many applications. Since [1] introduced the notion of model-based weakly convex optimization, a number of researches have been conducted to understand the theory of weakly convex problems in different fields. We believe that our contribution establishes a fundamental understanding of weakly convex optimization in the distributed setting. Besides, due to two recent work [6, 8], we believe that the understanding of DSGD in convex and smooth nonconvex setting has been rather mature, while there is still a gap between theory and practice in the weakly convex setting.
>
> 2. Lack of substantial evidence to demonstrate the significance of these improvements
>
>    We also beg to disagree with the reviewer on this point. Our major contribution is the improvement of dependency of our algorithms on delay, which is a very significant topic in distributed stochastic optimization. One clear evidence is that since [2], a lot of researches have been done on delayed stochastic optimization, and related works have been consistently published at decent venues and journals (see [5, 6, 7, 8, 9, 10, 11], where recent work [5, 8, 9] improve delays and are all published at ICML/NeurIPS). Given the vast amount literature on distributed delayed stochastic optimization, the significance of weakly convex optimization, together with our comprehensive empirical studies, we believe that our improvements are indeed significant.
>
> 3. Lack of in-depth explanation of DSGD/DSPL
>
>    We totally agree with the reviewer on the importance of intuition, and we have been trying to give the mathematical intuition of our improvements, especially on the delay-independent analysis. However, due to space limit we have to move some of the contents to the appendix. We kindly refer the reviewer to **Section D.4. DSPL vs DSGD** on the intuition why DSPL improves beyond DSGD.
>
>    As with why we manage to improve over the result of [3], it is indeed a good question. In our understanding, [3] puts emphasis on the momentum variant of DSGD, which we believe somehow complicates the analysis and makes it hard to improve on the dependency of delays. The intuition is that: if you extrapolate with inaccurate information, it would unlikely do you much favor.
>
> 4. Existence of min-max bound
>
>    To our knowledge there's not a min-max bound for our setting. The most relevant paper we can find [4] analyzes a lower-bound of DSGD, but their setting is different from ours, and does not take into account the real distributed environment.
>
> 5. Assumptions are strong
>
>    We understand the concern of the reviewer on our assumptions. In a word our analysis needs a few lines of modification to support non-uniform assumptions. For more details we refer the reviewer to our global reply.
>
> **Questions**
>
> 1. Experiment showing the convergence of DSPL is unaffected by the magnitude of delay
>
>    We kindly refer the reviewer to the third part of experiment (**Section 6.4**), which compares DSGD, DSPL and their robust variants under adversarial delays. We also have additional experiments in the appendix **Section H** that study the performance of DSGD/DSPL on blind deconvolution problem.
>
> We believe that our replies adequately address the concerns of the reviewer, and we again thank the reviewer for the efforts in reviewing our paper.
>
> **References**
>
> [1] Davis, Damek, and Dmitriy Drusvyatskiy. "Stochastic model-based minimization of weakly convex functions." *SIAM Journal on Optimization*
>
> [2] Agarwal, Alekh, and John C. Duchi. "Distributed delayed stochastic optimization." *Advances in neural information processing systems*
>
> [3] Xu, Yangyang, et al. "Distributed stochastic inertial-accelerated methods with delayed derivatives for nonconvex problems." *SIAM Journal on Imaging Sciences*
>
> [4] Arjevani, Yossi, Ohad Shamir, and Nathan Srebro. "A tight convergence analysis for stochastic gradient descent with delayed updates." *Algorithmic Learning Theory*. PMLR
>
> [5] Mishchenko, Konstantin, et al. "Asynchronous sgd beats minibatch sgd under arbitrary delays." *Advances in Neural Information Processing Systems*
>
> [6] Zheng, Shuxin, et al. "Asynchronous stochastic gradient descent with delay compensation." *International Conference on Machine Learning*
>
> [7] Stich, Sebastian U., and Sai Praneeth Karimireddy. "The error-feedback framework: Better rates for sgd with delayed gradients and compressed updates." *The Journal of Machine Learning Research*
>
> [8] Cohen, Alon, et al. "Asynchronous stochastic optimization robust to arbitrary delays." *Advances in Neural Information Processing Systems*
>
> [9] Koloskova, Anastasiia, Sebastian U. Stich, and Martin Jaggi. "Sharper convergence guarantees for asynchronous sgd for distributed and federated learning." *Advances in Neural Information Processing Systems*
>
> [10] Karimireddy, Sai Praneeth, et al. "Error feedback fixes signsgd and other gradient compression schemes." *International Conference on Machine Learning*
>
> [11] Lian, Xiangru, et al. "Asynchronous parallel stochastic gradient for nonconvex optimization." *Advances in neural information processing systems*

---

> > ### Author Response · Authors · 2023-08-20
> >
> > Dear Reviewer 93YJ,
> >
> > Thank you for your time spent on our paper. As the discussion period is coming to an end, we wish to follow up to ascertain if there are any additional questions or concerns that you may have, requiring our clarification. We sincerely appreciate the valuable comments you have provided on our paper thus far.

---

### Official Review · Reviewer_6vL5 · 2023-06-30

**Soundness:** 3 good
**Presentation:** 4 excellent
**Contribution:** 3 good
**Rating:** 8
**Confidence:** 3

**Summary:**

This paper studies distributed nonsmooth optimization with delayed updates. The authors study two kinds of objective functions, the first class is the sum of a weakly-convex nonsmooth function and a computation-friendly nonsmooth regularizer, while the second class is the sum of a composite function with a regularization, where the inner function of the composite function is smooth. For the first class of functions, the paper improves the results in Reference[1] by replacing the maximum delay with the expected delay and the second momentum of the delays. For the second class, they propose a prox-linear algorithm that can further get rid of the delay in the dominant term of the convergence rate. Convincing numerical experiments also demonstrate the effectiveness of the proposed algorithms.

**Strengths:**

* the delay in the dominant term is removed for the composite functions where the inner function is smooth. This is the first result to remove the dependence on the delay in the dominant term for nonsmooth and nonconvex functions.

* for weakly-convex nonsmooth functions, the maximum delay in the convergence rate is improved to the expected delay.

* the presentation and structure of the paper are quite clear.

**Weaknesses:**

* for prox-linear methods, the inner function is usually vector-valued. I would like to know whether it is possible to extend Algorithm 2 to the cases when the inner function is a mapping while the convergence rate remains invariant.

**Questions:**

* Is it possible to generalize the inner function of the composite function to a mapping while the convergence rate remains invariant.

* I would like to know about the maximum magnitude of numerical errors in the solutions to the subproblem in Algorithm 2 so as to keep the convergence rate invariant.

**Limitations:**

I am not aware of any potential negative social impact of this paper.

---

> ### Author Rebuttal · Authors · 2023-08-09
>
> **Response to Reviewer 6vL5**
>
> We appreciate the efforts of the reviewer in reviewing our paper.
>
> **Questions**.
>
> 1. Is it possible to generalize the inner function of the composite function to a mapping
>
>    Yes. Our analysis actually can be directly extended to the case, where function $c$ is replaced by $C$, almost without any modification. The only difference is that we have to replace the properties (e.g., Lipschitzness) of $c$  by the counterparts in mappings. We adopt function $c$ to make our notations consistent with DSGD.
>
> 2. The maximum magnitude of numerical errors in the solutions to the subproblem in Algorithm 2.
>
>    This is indeed a very good question concerning the inexact analysis of prox-linear method. To our knowledge  one close work [1] discusses the impact of inexactness on prox-linear method, where the result therein (See [1], **Theorem 5.2**) is summarized as follows
>
>    **Letting $\delta_k$ be the error (sub-optimality) of the $k$-th prox-linear update, then the impact of error will not hurt convergence if $\sum_k \delta_k$ is summable.**
>
>    Following [2], the above analysis can be further extended to the stochastic setting, where we need  $\frac{1}{\sqrt{K}}\sum_k\sqrt{\delta_k}$ to be summable. We have a formal proof of this inexact analysis and are willing to provide it using an anonymous link on reviewer's request.
>     We also remark that for the structures we are interested in, the subproblems 1) often have closed form solution, and 2) can commonly be solved to high accuracy using efficient subroutines. We kindly refer the reviewer to our appendix **Section G.3**, where we show how to efficiently solve the proximal subproblems.
>
>
> **References**
>
> [1] Drusvyatskiy, Dmitriy, and Courtney Paquette. "Efficiency of minimizing compositions of convex functions and smooth maps." *Mathematical Programming* 178 (2019): 503-558.
>
> [2] Wang, Jialei, Weiran Wang, and Nathan Srebro. "Memory and communication efficient distributed stochastic optimization with minibatch prox." *Conference on Learning Theory*. PMLR, 2017.

---

> ### Comment · Reviewer_6vL5 · 2023-08-16
> **Reply**
>
> I appreciate the authors' efforts to refine their paper and consider more general and practical scenarios. I have raised the score.

---

> > ### Author Response · Authors · 2023-08-20
> >
> > Thank you for your valuable comments and for raising the score

---

### Official Review · Reviewer_SDCr · 2023-07-07

**Soundness:** 3 good
**Presentation:** 3 good
**Contribution:** 3 good
**Rating:** 7
**Confidence:** 3

**Summary:**

This paper presents DSPL, which is an optimization for a certain structured weakly convex problems. This algorithm can work with arbitrarily delayed updates, and outperforms existing algorithms in both theoretical rates and numerical experiments.

**Strengths:**

1. The algorithm is robust to arbitrary delays. The robust rate doesn't rely on $\tau$.
2. The safeguard is a simple procedure, which makes it more practical.
3. The authors conduct numerical experiments which showcase the effectiveness of proposed algorithms.

**Weaknesses:**

1. The authors compare the proposed algorithm with some non-convex algorithms. Is this paper's setting inherently simpler than nonconvex optimization?

**Questions:**

N/A

---

> ### Author Rebuttal · Authors · 2023-08-09
>
> **Response to Reviewer SDCr**
>
> We thank the reviewer for the efforts put into the review process.
>
> **Question**. Is this paper's setting inherently simpler than nonconvex optimization.
>
> This is a good question, and we would like to answer the question for SGD and SPL respectively.
>
> For SGD, it is known that general weakly convex problems are a subset of locally Lipschitz continuous nonsmooth nonconvex optimization, and it is more diffIcult than the convex or smooth case, since smooth nonconvex optimization with Lipschitz continuous gradient is a subset of weakly convex functions.
>
> For SPL, it deals with weakly convex problems with composition structure. This setting is a subset of weakly convex optimization with broad applications from machine learning. And it's still harder than smooth nonconvex optimization with Lipschitz continuous gradient.
>
> We would like summarize the difficulty of the settings as follows.
>
> Smooth nonconvex (DSGD) < Structured Weakly convex (DSPL, ours) <  Weakly convex (DSGD, ours)  < Locally Lipschitz Nonsmooth Nonconvex (no result to our knowledge)

---

> > ### Comment · Reviewer_SDCr · 2023-08-12
> > **Response**
> >
> > Thanks for clarifying.

---

> > > ### Author Response · Authors · 2023-08-20
> > >
> > > Thank you for your constructive comments

---

### Official Review · Reviewer_Ld2g · 2023-07-27

**Soundness:** 4 excellent
**Presentation:** 4 excellent
**Contribution:** 3 good
**Rating:** 8
**Confidence:** 2

**Summary:**

This paper studies delayed stochastic algorithms for weakly convex optimization. For a general weakly convex problem, the authors provide a sharper bound that improves the max delay in the previous bounds to "average" delays. For a weakly convex problem with a composition structure, they propose a variant, DSPL, of the delayed SGD (DSGD) by only linearizing the inner function and provide a convergence analysis. Moreover, they propose variants of DSGD and DSPL that are robust to arbitrary delays by checking if the delay is too outdated. Finally, they provide some empirical results to support their theoretical findings.

**Strengths:**

1. I'm not an expert in this literature but the work seems to be technically solid and make a significant contribution.
2. The paper is very well written with enough intuition and discussions given after each main results. I enjoyed reading it.
3. The theoretical results are strong and supported with empirical evidence

**Weaknesses:**

The experiment section seems to be rushed. I left a few questions below.

**Questions:**

1. Could you add labels to the axes in your figures?
2. The right two figures in Figure 1 are impossible to parse given that many curves are very close to each other. I suggest you split each of them into subplots. For example, you could have a subplot showing DSGD 1 worker v.s. DSPL 1 worker, one showing DSGD with different workers, and one showing DSPL with different workers.
3. The two types of delays in Figure 2 are not defined.
4. In Section 6.4 it is claimed that "upon discarding outdated information, DSGD exhibits much greater robustness". How to see that from the plots?
5. The bound of DSPL also has the $\Delta_2$ term. Why is it robust to your adversarial delay? Does that mean your bound could be improved?

**Limitations:**

Yes

---

> ### Author Rebuttal · Authors · 2023-08-09
>
> **Response to Reviewer Ld2g**
>
> We appreciate the efforts the reviewer puts into the review process and thank the reviewer for acknowledging the novelty of our work.
>
> **Questions**.
>
> 1. Add labels to the axes of figures and split the figures.
>
>    We thank the reviewer for the suggestion and we are glad to add the axes to the figures.
>
>    As with splitting Figure 1, we would put this in the appendix due to page limit.
>
> 2. The two types of delays in Figure 2 are not defined
>
>    We kindly refer the reviewer to our experiment setup 5) (Line 284), where we briefly discussed the two types of delays. But we'll add more details on how these delays are generated.
>
> 3. "upon discarding outdated information, DSGD exhibits much greater robustness"
>
>    Sorry for the confusion, we intended to mean "SAFE-DSGD" exhibits greater robustness, as we can see from the convergence plot
>
> 4. The bound of DSPL also has the $\Delta_2$ term.
>
>    We interpret the reviewer's question as "DSPL, even without safe-guarding step, still performs well in practice". Indeed, in our experiments we observed that DSPL is only slightly inferior to SAFE-DSPL. One explanation is that our starting point is initialized within a ball, which itself is not that far from the optimal solution. Therefore updating with information of $x^0$ will not badly hurt convergence of DSPL.
>
>    In other words, while the delays are generated according to the worst case, the problem we optimize is not.
>
> We again thank the reviewer for the efforts in reviewing our paper.

---

> > ### Comment · Reviewer_Ld2g · 2023-08-17
> > **Response to Rebuttal**
> >
> > Thank you for your clarification. I have no more questions.

---

> > > ### Author Response · Authors · 2023-08-20
> > >
> > > Thank you for your constructive suggestions!

---

### Official Review · Reviewer_bGwb · 2023-07-27

**Soundness:** 2 fair
**Presentation:** 2 fair
**Contribution:** 2 fair
**Rating:** 5
**Confidence:** 5

**Summary:**

The authors study a class of weakly convex optimization over a distributed network under the presence of delayed derivatives. They first analyze an existing algorithm and show that they can improve its complexity bound in terms of the dependence on the properties of the delay. They further provide another algorithms and show even a more relaxed bound dependent on the delay.

**Strengths:**

The authors can improve the complexity bounds in terms of their dependence on the values of delay.

**Weaknesses:**

Assumptions B1 and C1 seems to be much stronger than the ones in the existing works as they have been made on the stochastic functions.

**Questions:**

It seems that one of the reasons to get a better dependency of the delay, is to have direct assumption on the stochastic functions and not the expectation ones. The authors need to clarify this.

**Limitations:**

.

---

> ### Author Rebuttal · Authors · 2023-08-09
>
> **Response to reviewer bGwb**
>
> We thank the reviewer for the time spent reviewing our paper.
>
> **Question**. Assumptions B1 and C1 are much stronger than the ones in the existing works; The reasons to get a better dependency of the delay is to have direct assumption on the stochastic functions and not the expectation ones.
>
> We understand the concern of the reviewer on the assumptions, and our original intention was to put emphasis on the delay itself in a clean and reader-friendly manner, so that enough intuition can be established. In a word we can make standard assumptions (as in [1]) by a few lines' modification of the proof, giving a slightly worse constant in the final results. And we kindly refer the reviewer to our global reply for more details.
>
> **References**
>
> [1] Xu, Yangyang, et al. "Distributed stochastic inertial-accelerated methods with delayed derivatives for nonconvex problems." *SIAM Journal on Imaging Sciences* 15.2 (2022): 550-590.

---

> > ### Comment · Reviewer_bGwb · 2023-08-17
> >
> > Thanks for the clarification. I have raised my rating.

---

> > > ### Author Response · Authors · 2023-08-20
> > >
> > > Thank you for your valuable comments and for raising the score

---

### Author Rebuttal · Authors · 2023-08-09

**Clarification of the assumptions**

First, we thank all the reviewers for their efforts in the review process, and for acknowledging the novelty of our work.

Both reviewer bGwb and 93YJ share the concern that the assumptions are made on stochastic functions and not the expectation ones, which could be a limitation and make it hard to tell the source of improvement.

We appreciate the insightful feedback from the reviewers. We concur that assumptions on the expected function, $\mathbb{E}[f(x,\xi)]$, are indeed more standard and weaker compared to those on the stochastic functions $f(x,\xi)$. However, transitioning to assumptions on $\mathbb{E}[f(x,\xi)]$ can be achieved in a standard manner. **While this change might lead to small adjustments to some constants in the rate, we assure that the dependency on both $K$ and $\tau$ are unaffected.**

To convince the reviewers, we highlighted the key difference in the updated analysis for DSGD/DSPL and are glad to provide a link of pdf with more details on reviewer's request. We will also attach the proofs directly on official comments (when it becomes visible). We believe this suffices to convince the reviewers that these adjustments are primarily related to the constants in a few inequalities, leaving the core conclusions of our work intact. Furthermore, we promise that the revisions will detail the adjustments made in our proof to incorporate these changes.

Our revised assumptions will be

**Rlx B1**. $f(x)$ is $\lambda$-weakly convex and $\mathbb{E}[\|\| f' (x,\xi)
\|\|^2]\leq L_f^2$. ($f'(x,\xi)=g\in \partial f(x,\xi)$) for all $x \in \text{dom} ~\omega$

**Rlx C1**. $c(x,\xi)$ is $C(\xi)$-smooth and $\mathbb{E}[\|\| \nabla c(x,\xi)\|\|^2]\leq L_c^2$,   $\mathbb{E}[\|\| C(\xi)\|\|^2]\leq C^2$
for all $x \in \text{dom} ~\omega$.

These assumptions are standard in stochastic optimization literature, and now we highlight the modifications of the proof, taking DSGD as an example (similar analysis is applicable to DSPL). Now we get down to technical details.

**Step 1.** Bounding $\|\|x^{k+1}-x^k\|\|$ on expectation.
The first consequence of **Rlx B1** is that our original bound
$$\|\|x^{k+1}-x^k\|\|\leq \frac{2(L_f+L_\omega)}{\gamma} \tag{1}$$
no longer holds uniformly. Now we instead show it on expectation. Recall that
$$\begin{align}
  x^{k+1} ={} & \arg \min\_x  \\{ \langle g^{k-\tau_k},x-x^k\rangle +
  \omega (x)+\frac{\gamma}{2}\|\|x-x^k\|\|^2 \\}
\end{align}$$
in our proximal update. And by three-point lemma,
$$\langle g^{k-\tau_k},x^{k+1}-x^k\rangle+\omega (x^{k+1}) +
   \frac{\gamma}{2}\|\|x^{k+1}-x^k\|\|^2\leq \omega (x^k)- \frac{\gamma}{2}\|\|x^{k+1}-x^k\|\|^2$$
Re-arranging the terms and applying Cauchy-Schwartz,
$$\gamma\|\|x^{k+1}-x^k\|\|^2\leq (L_{\omega} +\|\|g^{k-\tau_k}\|\|)
  \|\|x^{k+1}-x^k\|\|$$
Dividing both sides by $\|\|x^{k+1}-x^k\|\|$,
$$\|\|x^{k+1}-x^k\|\|\leq \frac{1}{\gamma} (L_{\omega}+g^{k-\tau_k})$$
Conditioned on the history, taking expectations on both sides,
$$\mathbb{E}\_k[\|\|x^{k+1}-x^k\|\|]\leq \frac{1}{\gamma} \mathbb{E}\_{\xi^{k-\tau_k}}[L_{\omega}+\|\|g^{k-\tau_k}\|\|]\leq \frac{L_f+L_\omega}{\gamma},$$
where the last relation is by Jensen's inequality. Now we get a counterpart of $(1)$. A more tricky part is about $\mathbb{E}\_k[\|\|x^{k+1}-x^k\|\|^2]$, and we note that
$$\begin{align}
  \gamma \mathbb{E}\_k[\|\|x^{k+1}-x^k\|\|^2] \leq{} & \mathbb{E}\_k
 [(L_{\omega}+\|\|g^{k-\tau_k}\|\|)\|\|x^{k+1}-x^k\|\|]\\\\
  ={} & \mathbb{E}\_k
 [\|\|g^{k-\tau_k}\|\|\cdot\|\|x^{k+1}-x^k\|\|]+\mathbb{E}\_k
 [L_{\omega}\|\|x^{k+1}-x^k\|\|]\\\\
  \leq{} & \mathbb{E}\_k[(2\gamma)^{- 1}\|\|g^{k-\tau_k}\|\|^2]+\frac{\gamma}{2} \mathbb{E}\_k[\|\|x^{k+1}-x^k\|\|^2]+\frac{2 L_{\omega} (L_{\omega}+L_f)}{\gamma}\\\\
  \leq{} & \frac{L_f^2+4L_{\omega}(L_f+L_{\omega})}{2\gamma}+\frac{\gamma}{2}\mathbb{E}\_k[\|\|x^{k+1}-x^k\|\|^2]
\end{align}$$
The last relation uses $2x y\leq a x^2+\frac{y^2}{a}$ to decouple correlated $\|\|g^{k-\tau_k}\|\|$ and $x^{k+1}$; $\mathbb{E}\_k
 [L_{\omega}\|\|x^{k+1}-x^k\|\|]$ is bounded using $(1)$. Finally, we re-arrange the terms to cancel $\frac{\gamma}{2} \mathbb{E}\_k[\|\|x^{k+1}-x^k\|\|^2]$ on both sides, giving
$$\mathbb{E}\_k[\|\|x^{k+1}-x^k\|\|^2]\leq \frac{L_f^2+4L_{\omega}(L_f+L_{\omega})}{\gamma^2}$$
Compared to our original result, the bound on $\mathbb{E}\_k[\|\|x^{k+1}-x^k\|\|^2]$ becomes slightly worse in the enumerator.

**Step 2. Relaxing uniform assumption on the stochastic function**

Our assumption on uniform weak convexity appears on Line 561 in the appendix, where we have
$$\mathbb{E}\_{\xi^{k-\tau_k}}[\langle g^{k-\tau_k}, \hat{x}^k - x^{k -
   \tau_k} \rangle]+f (x^{k-\tau_k}) - f (\hat{x}^k) \leq
   \frac{\lambda}{2}\|\| \hat{x}^k - x^{k-\tau_k}\|\|^2$$
using $\lambda$-weak convexity. However, since $\mathbb{E}\_{\xi^{k-\tau_k}}[g^{k-\tau_k}] \in \partial f (x^{k -
\tau_k})$, we only need $\lambda$-weak convexity of $f$.


**Step 3. Putting things together**

After the two steps above, we can smoothly go through, where our last piece of proof (Line 572) needs a modification of constant in bounding $\sum_{k=1}^K \mathbb{E}[\|\| x^k - x^{k-\tau_k}\|\|^2]$.
$$\begin{align}
 \sum_{k=1}^K\mathbb{E}[\|\|x^k-x^{k-\tau_k}\|\|^2]={}&\sum_{k=1}^K
\mathbb{E}[\|\|\sum_{l=1}^{\tau_k}x^{k+1-l}-x^{k-l}
\|\|^2]\\\\
 \leq{}&\sum_{k=1}^K\tau_k\sum_{l=1}^{\tau_k}\mathbb{E}[\|\|x^{k+1-
l}-x^{k-l}\|\|^2]\\\\
\leq{}&\frac{L_f^2+4L_{\omega}(L_{\omega}+L_f)}{\gamma^2}\sum_{k=
1}^K\tau_k^2,
\end{align}$$
where the first inequality uses the relation $\|\|\sum_{i=1}^ka_i\|\|^2\leq k \sum_{i=1}^k\|\|a_i\|\|^2$ and the second inequality applies our modified bound on $\mathbb{E}\_k[\|\|x^{k+1}-x^k\|\|^2]$. The bound on $\sum_{k=1}^K \mathbb{E}[\|\| x^k - x^{k-\tau_k}\|\|]$ remains unchanged.

At this stage, **our analysis is fully compatible with the standard assumptions**, at the cost of a slightly worse constant in the bound. Our convergence rate and main results are unaffected.

---

### Decision · Program_Chairs · 2023-09-21

**Decision:**

Accept (poster)

**Comment:**

The authors develop new theoretical guarantees for the delayed stochastic subgradient method for nonsmooth weakly convex optimization problems. Their work improves on previous bounds for nonsmooth distributed optimization by replacing the maximum delay by the first and second moments of the delay. They also present delay-independent rates for nonsmooth weakly convex optimization problems.